# Brazilian fossils reveal homoplasy in the oldest mammalian jaw joint

James R. G. Rawson[1✉], Agustín G. Martinelli[2✉], Pamela G. Gill[1,3], Marina B. Soares[4], Cesar L. Schultz[5] & Emily J. Rayfield[1✉]

The acquisition of the load-bearing dentary–squamosal jaw joint was a key step in mammalian evolution[1–5]. Although this innovation has received decades of study, questions remain over when and how frequently a mammalian-like skull–jaw contact evolved, hindered by a paucity of three-dimensional data spanning the non-mammaliaform cynodont–mammaliaform transition. New discoveries of derived non-mammaliaform probainognathian cynodonts from South America have much to offer to this discussion. Here, to address this issue, we used micro-computed-tomography scanning to reconstruct the jaw joint anatomy of three key probainognathian cynodonts: *Brasilodon quadrangularis*, the sister taxon to Mammaliaformes[6–8], the tritheledontid-related *Riograndia guaibensis*[9] and the tritylodontid *Oligokyphus major*. We find homoplastic evolution in the jaw joint in the approach to mammaliaforms, with ictidosaurs (*Riograndia* plus tritheledontids) independently evolving a dentary–squamosal contact approximately 17 million years before this character first appears in mammaliaforms of the Late Triassic period[10–12]. *Brasilodon*, contrary to previous descriptions[6–8], lacks an incipient dentary condyle and squamosal glenoid and the jaws articulate solely using a plesiomorphic quadrate–articular joint. We postulate that the jaw joint underwent marked evolutionary changes in probainognathian cynodonts. Some probainognathian clades independently acquired 'double' craniomandibular contacts, with mammaliaforms attaining a fully independent dentary–squamosal articulation with a conspicuous dentary condyle and squamosal glenoid in the Late Triassic. The dentary–squamosal contact, which is traditionally considered to be a typical mammalian feature, therefore evolved more than once and is more evolutionary labile than previously considered.

Mammals possess a unique secondarily evolved jaw joint between the dentary and squamosal bones[1,2]. The elements forming the ancestral jaw joint, the quadrate and articular, were separated from the lower jaw during the evolution of mammals from Mesozoic era fossil relatives, the non-mammaliaform cynodonts[1–5]. The evolution of the dentary–squamosal articulation is therefore key to understanding the origin of the unique mammalian body plan. Newly discovered non-mammaliaform cynodonts from the Late Triassic of South America are well placed, both in time and in the evolutionary tree, to answer when and how often the precursor conditions to this load-bearing jaw joint first evolved. Here we use micro-computed-tomography (μCT) scanning to analyse the jaw joint morphology of three key non-mammaliaform cynodont species and identify the oldest example of a dentary–squamosal contact in the fossil record in *R. guaibensis*[9], predating that seen in Mammaliaformes by approximately 17 million years[10–12]. Our updated phylogenetic analyses show that this contact evolved independently from that of mammal precursors, demonstrating that this classical feature is more evolutionarily labile than

previously thought. *B. quadrangularis*, the sister taxon to Mammaliaformes, does not possess a dentary–squamosal jaw articulation as was previously interpreted[6–8]. Evolutionary experimentation in jaw joint morphology occurred across the cynodont–mammaliaform transition, coinciding with the evolution of features such as thermoregulation, insectivory and miniaturization that would become key elements of the mammalian body plan for millions of years to come[13–18].

Among the most important diagnostic features of mammals are a unique middle ear containing three ossicles (malleus, incus, stapes) and a load-bearing jaw hinge comprising a dentary condyle and squamosal glenoid[3,4,19,20]. The definitive mammalian middle ear (also called the detached middle ear[21,22]) evolved within Mammaliaformes during the Mesozoic, with fossil and developmental evidence suggesting that this event occurred at least three times independently[20,23–25]. However, the major morphological changes needed to facilitate this innovation took place in non-mammaliaform cynodonts (hereafter, cynodonts), the precursors of mammals[5,26]. The fossil record documents the evolution of the jaw from early diverging cynodonts

[1]Palaeobiology Research Group, School of Earth Sciences, University of Bristol, Bristol, UK. [2]Museo Argentino Ciencias Naturales "Bernardino Rivadavia"—CONICET, Buenos Aires, Argentina. [3]Natural History Museum, London, UK. [4]Museu Nacional, Universidade Federal do Rio de Janeiro, Rio de Janeiro, Brazil. [5]Universidade Federal do Rio Grande do Sul, Porto Alegre, Brazil. ✉e-mail: jr17384@bristol.ac.uk; agustin_martinelli@yahoo.com.ar; e.rayfield@bristol.ac.uk

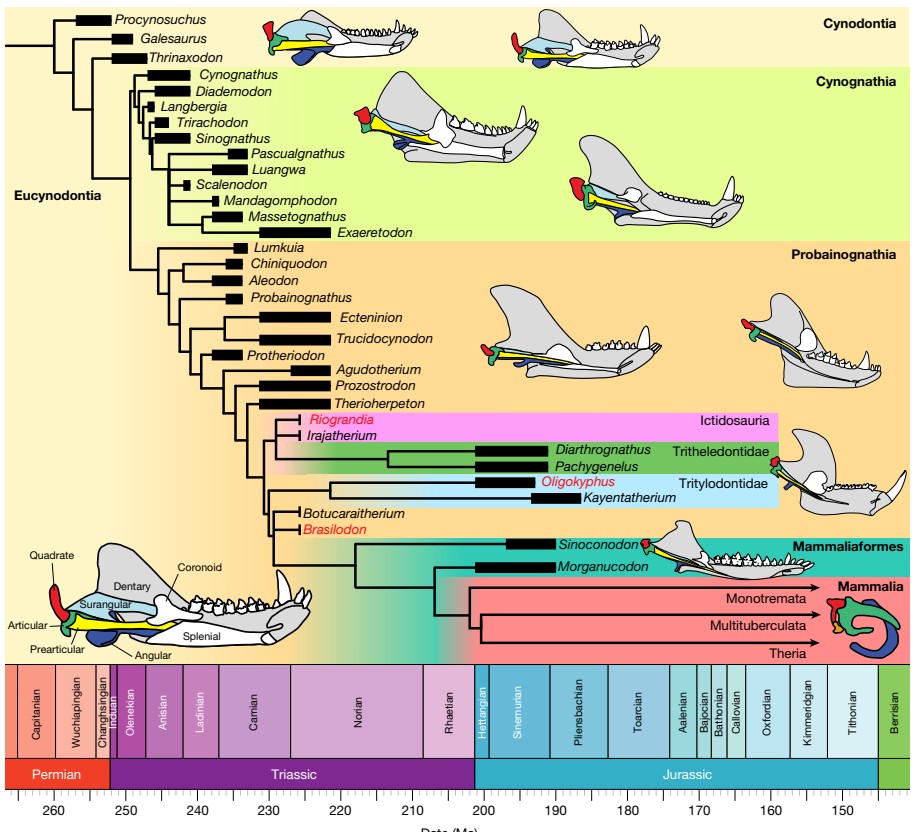

**Fig. 1 | Time-calibrated relationships of Cynodontia.** The topology is based on the phylogenetic analysis within this study (methodology and detailed results are shown in the Supplementary Information). The black bars at the tree tips show the stratigraphic range of each taxon, with the geological timescale below. Outlines of lower jaws and postdentaries for key cynodont taxa adapted from the literature[1–5,26,28,46,49,62,66,83,84] are shown in medial view (not to scale), with an example *Thrinaxodon* jaw in the bottom left corner denoting the colour of each bone. Taxa studied here (*Brasilodon*, *Riograndia*, *Oligokyphus*) are highlighted in red. Ma, million years ago. Outlines of lower jaws and postdentaries adapted with permission from refs. 4,26,84 (Springer Nature Limited); ref. 28 (*Cynognathus, Morganucodon*; Oxford Univ. Press); ref. 5 (*Thrinaxodon*; Annual Review of Ecology, Evolution and Systematics); refs. 3,46 (*Probainognathus*, *Diarthrognathus*; John Wiley and Sons); and ref. 83 (*Diademodon*; Palaeontologia Africana).

to mammaliaforms[1,2,4], in which the postdentary bones (articular, prearticular, surangular, angular) are reduced in size compared with the tooth-bearing dentary to become a collection of thin, rod-like elements as their role in jaw articulation is incrementally reduced (Fig. 1). Eucynodonts experimented with several modifications to the skull that improved sensorial and feeding habits, coupled with the formation of a secondary jaw articulation in between the surangular boss and the articular flange of the squamosal[2,26]. Further morphological changes led to the formation of the load-bearing dentary–squamosal joint, consisting of a distinct dentary condyle that articulates to the glenoid cavity of the squamosal bordered posteriorly by a glenoid ridge. In early-diverging mammaliaforms such as *Morganucodon*, *Sinoconodon* and others, the dentary–squamosal articulation lies alongside the plesiomorphic quadrate–articular joint[27–29]. The formation of the dentary–squamosal jaw articulation freed the postdentary bones from their ancestral role as load-bearing elements and allowed them to specialize for sound detection and finally separate from the lower jaw entirely in crown mammals (Fig. 1). Understanding the phylogenetic history and timing of evolution of the dentary–squamosal contact through the cynodont–mammaliaform transition is therefore crucial to interpretation of the origin of this unique mammalian structure.

Despite a long history of study[20], questions remain surrounding the evolution of the jaw joint in cynodonts. Probainognathian cynodonts show variation in jaw joint morphology suggesting evolutionary experimentation among some, but not all, groups[2,4,5]. Disagreement in anatomical interpretation and scarcity of fossils have hindered investigation into jaw anatomy and function in key clades. Furthermore,

the method by which cynodonts maintained a functioning jaw joint before the formation of the load-bearing dentary–squamosal articulation while simultaneously reducing the postdentary bones for improved sound detection remains unclear. Both reorganization of adductor musculature[1,3,30,31] and overall body miniaturization[32] have been considered as hypotheses, although these are not mutually exclusive. Comparative studies of cynodont jaw anatomy have to date largely focused on well-known taxa[1–3,32] that are exemplary in terms of preservation and usefulness in anatomical study, but gaps remain in our understanding of cynodont evolution. The reliably dated Candelária Sequence of the Santa Maria Supersequence of southern Brazil[12] has in recent years delivered a suite of derived Late Triassic probainognathian cynodonts[7,9,33–35], many of which are represented by multiple specimens including ontogenetic sequences. Of particular note is *B. quadrangularis*, which has been recovered by most phylogenetic analyses[7–9,26,33–43] as the sister taxon to mammaliaforms and is purported to possess a dentary–squamosal joint similar to that of *Morganucodon*[5,7,8,26,28]. The robust-jawed *R. guaibensis* is also abundant in the Candelária Sequence, being the key taxon of the *Riograndia* Assemblage Zone, which includes *Brasilodon*. This Assemblage Zone was dated using [206]Pb/[238]U, revealing a maximum deposition age of 225.42 ± 0.37 million years (early Norian)[44]. The phylogenetic placement of *Riograndia* has varied through time, being considered either a tritheledontid[9], a non-tritheledontid ictidosaur[37] or allied to other traditional ictidosaurs but not forming a monophyletic clade[35,38,43,45]. Regardless, it is unambiguously a derived probainognathian placed stemward of mammaliaforms. Previous descriptions

have indicated that Jurassic tritheledontids may possess a type of dentary–squamosal contact[1,4,5,9,46,47], although this contact lacks the robust dentary condyle as seen in mammaliaforms such as *Morganucodon*[5,28]. The phylogenetic position and preservational quality of *Brasilodon*, *Riograndia* and other recently described probainognathian cynodonts makes these taxa key to understanding the origin of the mammalian jaw joint and middle ear, but the anatomical information of these new fossils has yet to be integrated into the wider picture of cynodont morphological evolution alongside better-known species.

To better establish the evolution of jaw joint morphology at this pivotal stage in mammalian history, we reconstructed the three-dimensional anatomy of the jaw joint in *Brasilodon*, *Riograndia* and the tritylodontid *Oligokyphus* using a 3D dataset of cynodont fossils obtained by μCT scanning (Supplementary Table 1; available at data.Bris (Data availability)). We incorporate morphology revealed by our μCT data into an updated phylogenetic analysis (Fig. 4 and Supplementary Information), demonstrating substantial homoplastic evolution of the jaw hinge and middle ear characters in the phylogenetic predecessors of mammals. *Brasilodon* does not possess a dentary–squamosal articulation as previously described, whereas *Riograndia* possesses a clear dentary–squamosal contact similar to that of the closely related tritheledontids. *Riograndia* predates the earliest-known mammaliaform condyle–glenoid jaw joint by approximately 17 million years. This demonstrates that the ictidosaur–tritheledontid clade (Fig. 4) independently acquired the dentary–squamosal contact that bears resemblance to the jaw hinge of mammaliaforms. Evolutionary experimentation with the jaw hinge occurred in derived cynodonts, and some precursor characters evolved iteratively in probainognathians, and then in mammaliaforms.

## The jaw joint morphology of *Brasilodon*

μCT scans of nine *Brasilodon* specimens highlight notably different jaw joint anatomy from that described previously (Fig. 2). As in many derived probainognathians[1,2,26,48], the lateral and medial ridges of the dentary in *Brasilodon* create a mediolaterally thickened articular process that extends posteriorly beyond the postdentary bones. The articular process was described previously as an incipient condyle comparable to the condylar process seen in mammaliaforms[6–8], but the 3D data show that this thickening is much less developed than in *Morganucodon* and *Sinoconodon*, despite the pronounced posterior extension (Fig. 2d–f). The μCT data of the well-preserved specimens of *Brasilodon* (UFRGS-PV-929-T, UFRGS-PV-1030-T, UFRGS-PV-1043-T) also do not support the presence of an incipient squamosal glenoid, and there is no clear contact surface on the squamosal to accept the lateral ridge of the dentary (Fig. 2e). Digital manipulation of the 3D model of the well-preserved dentaries of *Brasilodon* UFRGS-PV-1043-T to articulate the jaws shows that the articular process does not reach the squamosal when the teeth are in occlusion, even in the largest specimens. This condition is also present in the minimally distorted left side of UFRGS-PV-1030-T (Fig. 2e), and other specimens such as UFRGS-PV-929-T and UFRGS-PV-760-T, although more distorted, show the same morphology. The lateral ridge of the dentary prohibits any interaction between the surangular and squamosal, meaning that *Brasilodon* also lacks the secondary surangular–squamosal articulation found in many other eucynodonts[2,26]. We find that *Brasilodon* possesses only the ancestral quadrate-articular jaw joint. The quadrate itself has a well-developed stapedial process and a pronounced neck around the articular surface resembling that of *Morganucodon*[7,47]. μCT scans show that the quadrate surface for contacting the squamosal is strongly concave, in a similar manner to *Pachygenelus* and especially *Morganucodon*[47]. Tightly attached to the quadrate is the quadratojugal, which is housed in its own notch in the squamosal separated from the more medial quadrate notch by a ventrally projecting lamina

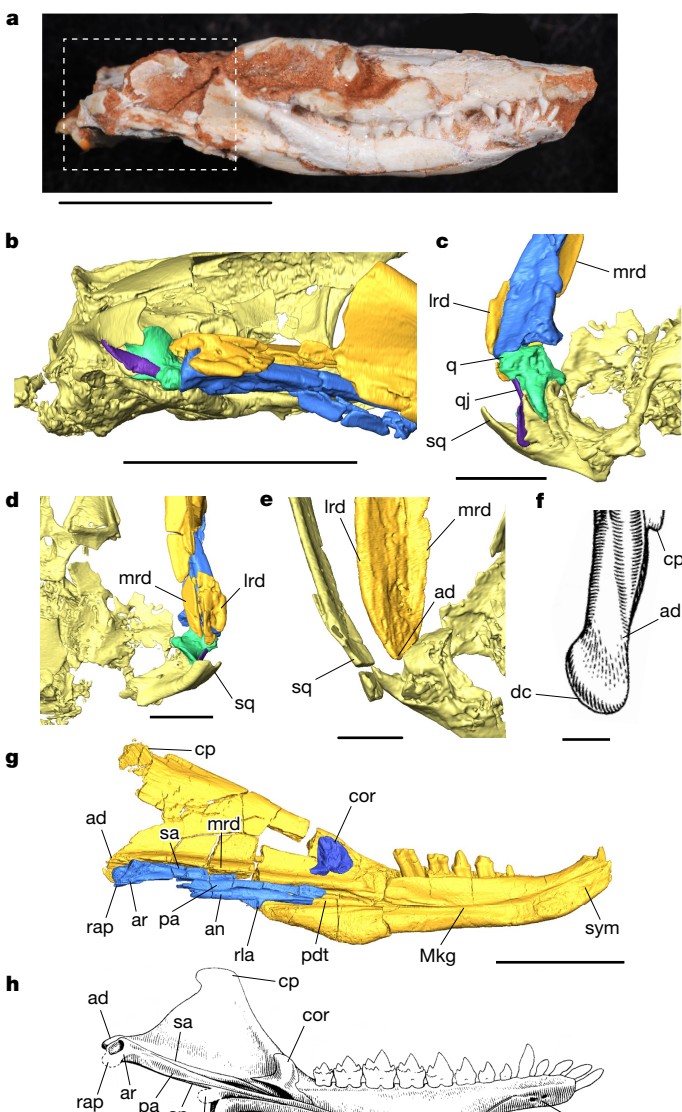

**Fig. 2 | Anatomy of *B. quadrangularis* informed by μCT scans. a**, Photograph of UFRGS-PV-929-T in the lateral view. The dashed area indicates the region shown in **b**. **b**, Lateral view of the right jaw joint region of UFRGS-PV-929-T. **c**, Ventral view of same region as in **b**. **d**, Dorsal view of same region as in **b**. **e**, Ventral view of surface model of right jaw of UFRGS-PV-1030-T, showing the dentary–squamosal relationship while the teeth are in occlusion. **f**, The right articular process of *Morganucodon* in the ventral view, for comparison. **g**, Medial view of left jaw of UFRGS-PV-628-T. **h**, Medial view of the left jaw of *Morganucodon*. ad, articular process of the dentary; an, angular; ar, articular; cor, coronoid bone; cp, coronoid process; dc, dentary condyle; lrd, lateral ridge of the dentary; Mkg, Meckelian groove; mrd, medial ridge of the dentary; pa, prearticular; pdt, postdentary trough; q, quadrate; qj, quadratojugal; rap, retroarticular process; rla, reflected lamina of the angular; sa, surangular; sq, squamosal; sym, mandibular symphysis. Scale bars, 10 mm (**a** and **g**), 5 mm (**b** and **h**), 2 mm (**c** and **d**) and 1 mm (**e** and **f**). Illustrations of *Morganucodon* are modified from ref. 28 (**f** and **h**; Oxford Univ. Press).

of bone. The arrangement of the quadrate, quadratojugal and squamosal in *Brasilodon* therefore represents the plesiomorphic condition shown in other non-mammaliaform probainognathians[42,47], contrary to previous descriptions[26]. Our scans show that the postdentary bones of *Brasilodon* (Fig. 2g) generally resemble those of cynodonts like *Probainognathus* and *Pachygenelus* in size and shape[5,48], rather than the comparatively much smaller and more slender postdentaries of

mammaliaforms such as *Morganucodon* (Fig. 2h), again contrary to previous descriptions[6,7]. The articular and prearticular are fused in all of the specimens and the surangular is fused to the dorsal surface of both in all but UFRGS-PV-1030-T, one of the smallest specimens. The angular remains unfused to the other postdentary bones in all of the specimens.

The jaw joint anatomy of *Brasilodon* presents a contradiction. The lack of a squamosal glenoid and spatial separation between squamosal and dentary indicates that the quadrate–articular joint is the main point of articulation, but the dentary possesses numerous features associated with the evolution of a dentary–squamosal joint, such as the expanded lateral and medial ridges and long articular process that extends posteriorly beyond the postdentary bones and prevents any squamosal–surangular contact (Fig. 2). Posterior extension of the articular process is a prerequisite condition for establishing a dentary–squamosal contact, but this by itself does not necessarily denote a dentary squamosal articulation. Tritylodontids possess an articular process that extends posteriorly to the postdentary bones, as in *Brasilodon*, but unambiguously show no evidence of a dentary–squamosal contact (Supplementary Information). The large medial ridge and posteriorly extended articular process of *Brasilodon* may have braced the postdentaries from above during jaw movement in the absence of a secondary joint (although such features are less developed in tritylodontids[49,50]), resulting in a precursor condition to the dentary–squamosal contact in which the articular process extends posteriorly but does not yet reach the squamosal. Alternatively, *Brasilodon* may have possessed some form of synovial or ligamentous connection between the dentary and squamosal that provided additional jaw joint support, producing a mismatch of plesiomorphic and derived features. Such a connection has been suggested for *Thrinaxodon* as a precursor to the secondary surangular-squamosal articulation[2]. There is some osteological evidence for this too: the posterior surface of the articular process of the dentary does appear to possess the type of rugosity commonly associated with cartilaginous covering.

## Dentary–squamosal contact in *Riograndia*

Our examination of *Riograndia* using μCT scanning has revealed the 3D morphology of its jaw joint region in fine detail (Fig. 3). We find a dentary–squamosal contact in *Riograndia* (Fig. 4), making this the oldest currently confirmed in the fossil record, although it differs in morphology to the developed articulation first seen in the Late Triassic (Norian–Rhaetian) mammaliaform *Haramiyavia*[10,51,52], the Late Triassic (Rhaetian) to Early Jurassic (Hettangian–Sinemurian) *Morganucodon*[11,16,53,54] and the Early Jurassic (Sinemurian) *Sinoconodon*[27,53]. The medial ridge of the dentary forms the roof of the postdentary trough and, together with the lateral ridge, it forms the mediolaterally broad articular process. As in *Brasilodon*, the articular process extends posteriorly beyond the postdentary bones and therefore prevents any articulation between the surangular and the squamosal (Fig. 3b,c,e,f). In contrast to *Brasilodon* and tritylodontids, the squamosal of *Riograndia* bears a clear indentation on the ventromedial surface that would have contacted the lateral ridge of the articular process of the dentary (Fig. 3c,d). This contact surface is well described in the Jurassic tritheledontids *Pachygenelus* and *Diarthrognathus*[1,2,27,46,55], which belong to the more inclusive ictidosaur clade including *Riograndia*[34,35] (Fig. 1). The 3D morphology shown by our μCT scans, particularly that of UFRGS-PV-596-T, enables us to firmly state that a similar contact between the dentary and squamosal was also present in *Riograndia*. The morphology of this contact differs in several important respects from a condylar dentary–squamosal jaw joint, as seen in mammaliaforms such as *Sinoconodon*, *Morganucodon* and *Haramiyavia*. In *Riograndia*, there is no distinct glenoid cavity on the squamosal posteriorly rimmed by a ridge, or a true condyle on the articular process

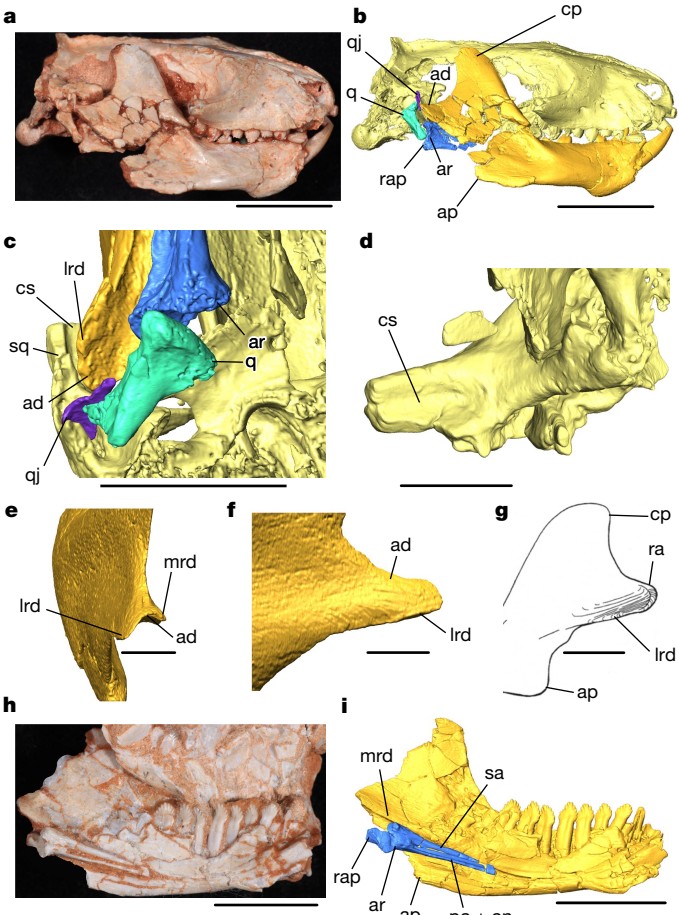

**Fig. 3 | Anatomy of the lower jaws and jaw joint of *R. guaibensis*.**
**a**, Photograph of UFRGS-PV-596-T in the lateral view. **b**, Surface model of UFRGS-PV-596-T in the lateral view. **c**, Ventral view of the right jaw joint of UFRGS-PV- 596-T. **d**, Medial view of the squamosal of UFRGS-PV-596-T, showing the contact surface. **e**, Surface model of the articular process of the left jaw of MPDC-1B1 in the posterior view. **f**, Lateral view of same region as in **e**. **g**, Line drawing of the articular process of *Pachygenelus* sp., modified from ref. 27. **h**, Photograph of the left jaw of UFRGS-PV-833-T in the medial view. **i**, Surface model of same region as in **h**, showing intact postdentary bones. ap, angular process of the dentary; cs, dentary contact surface; ra, thickened posterolateral rim of the articular process. Scale bars, 10 mm (**a**, **b**, **e** and **f**), 5 mm (**c**) and 2 mm (**d**).

of the dentary (Fig. 3c–f), and most of the articulating surface is on the dorsolateral portion of the articular process, rather than on the posterior. Overall, the squamosal configuration in *Riograndia* bears resemblance to the surangular-squamosal articulation present in some other eucynodonts such as *Probainognathus*[2]. However, the flaring lateral ridge of the dentary in *Riograndia* excludes the surangular from directly contacting the squamosal. Despite damage to the zygomatic arch, it is clear that the articular surface in *Riograndia* extended over a much larger area than the surangular–squamosal articulation of more basal eucynodonts[2] (Fig. 3d). This condition also differs from the morphology observed in the South African Jurassic tritheledontid *Diarthrognathus*, which possesses a thickened and rounded posterolateral rim on the articular process of the dentary, superficially resembling a condylar surface that is absent, or at least much less developed, in *Riograndia*[46]. *Pachygenelus*, another derived tritheledontid from South Africa, probably also possesses this condition[1,5,27,46,47] (Fig. 3g). It appears then that ictidosaurs (including *Riograndia* and tritheledontids)[37] show some structural variation of the dentary–squamosal contact: the early diverging *Riograndia* shows a simple convex articular

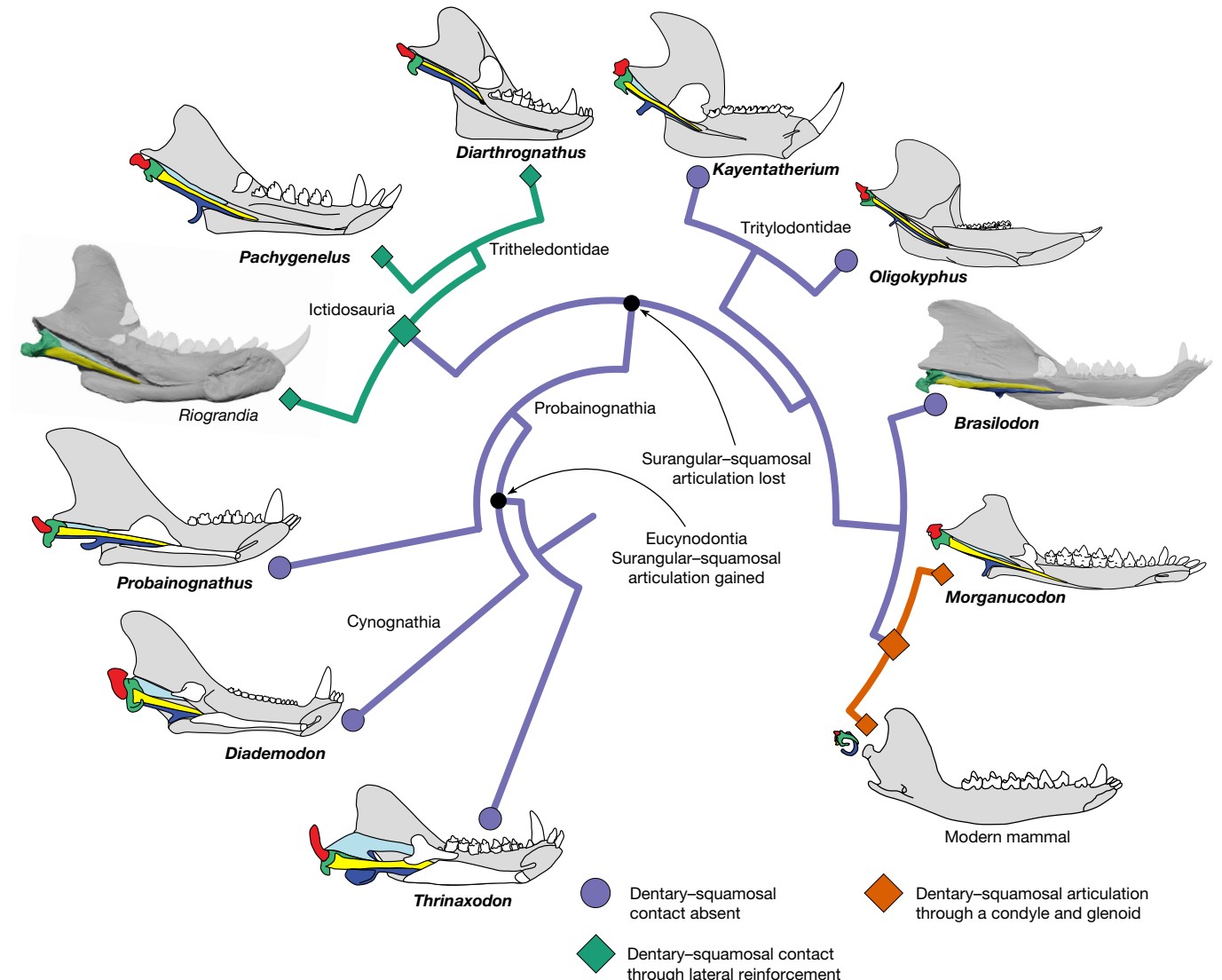

**Fig. 4 | Homoplastic evolution of the dentary–squamosal jaw joint in Cynodontia.** The branch colour denotes the inferred relationship between the dentary and squamosal across the tree. Reconstructions of *Brasilodon* and *Riograndia* jaws produced in this study are shown in the medial view. Topology is adapted from the most parsimonious trees obtained here, including the resolution of the Ictidosauria, and the additional medial jaw outlines are adapted from the literature[3–5,26,27,46,83–85] (Fig. 1). Details of phylogenetic analyses and results are provided in the Supplementary Information. Medial jaw outlines are adapted with permission from ref. 5 (*Thrinaxodon*; Annual Review of Ecology, Evolution and Systematics); refs. 3,46 (*Probainognathus*, *Diarthrognathus*, *Oligokyphus*; John Wiley and Sons); refs. 4,26,84 (Springer Nature Limited); ref. 83 (*Diademodon*; Palaeontologia Africana); and ref. 28 (*Morganocudon*; Oxford Univ. Press).

process forming the dentary–squamosal contact, whereas the more derived *Pachygenelus* and *Diarthrognathus* show more (albeit still incomplete) resemblance to the dentary condyle and squamosal glenoid in mammaliaforms. Tritheledontids never evolved the dentary condyle or posteriorly ridged glenoid of mammaliaforms, yet it does appear that the Jurassic forms were approaching a more derived condition with a more developed contact between the dentary and the squamosal.

The dentary–squamosal contact acts as reinforcement to the quadrate–articular joint in *Riograndia*. The postdentary bones are large, boxy and even more robust than those found in many basal probainognathians (such as *Probainognathus*) and tritheledontids such as *Pachygenelus* and *Diarthrognathus*[4,5,56] (Fig. 3b,c,h,i). Our μCT scans confirm the key features of postdentaries of *Riograndia* as described previously[9]: the fusion of the angular, articular and prearticular, and a posteriorly projected retroarticular process of the articular, and the reflected lamina on the angular is absent (or alternatively not ossified). *Riograndia* has a slender quadrate, and the new scans also confirm that

it is more similar to that of *Probainognathus* in the absence of stapedial process (a plesiomorphic trait) than to *Brasilodon* and *Morganuco-don*[9,47]. The μCT scans of UFRGS-PV-596-T also reveal the presence of a tiny quadratojugal in *Riograndia* (Fig. 3b,c), to our knowledge the first time that this bone has been observed in an ictidosaur. The quadrato-jugal is a narrow splint of bone of which the ventral portion is accepted by a shallow depression on the anterolateral surface of the quadrate, although the abutting contact has disarticulated in UFRGS-PV-596-T. Dorsally, the quadratojugal becomes flattened and plate-like and inserts into a notch in the squamosal separated by a thin ridge of bone from the more medial notch for the quadrate. The reported absence of a quadratojugal in tritheledontids such as *Pachygenelus*[5,47] may therefore be due to small size and a lack of preservation rather than a genuine loss, but this requires further investigation. The tiny quadratojugal contrasts with the large, robust quadrate, and the loose connection of the quadratojugal with the squamosal would have improved the ability of the quadrate/quadratojugal to rock forwards during jaw opening to prevent disarticulation[2].

The robust construction of the quadrate–articular joint alongside a secondary dentary–squamosal contact may imply adaptation to higher stresses in the jaw joint of *Riograndia*, and perhaps ictidosaurs in general. Characteristics such as enlarged and procumbent lower incisors, reduced canines, leaf-shaped postcanines and the general robustness of the skull and jaws suggest a herbivorous diet in *Riograndia*[9], and the boxy postdentaries may have been an additional adaptation to cope with increased loading at the jaw joint. The evolution of the jaw articulation across the cynodont–mammaliaform transition may therefore have been influenced by confounding ecological factors in specific taxa, rather than being a simple stepwise reduction in postdentary size. In tritheledontids like *Diarthrognathus* and *Pachygenelus*, the more derived jaw joint may have been able to distribute more stress across its thickened posterior contact surface than the simple convex morphology of *Riograndia* and so allowed those taxa the freedom to further reduce the size of their postdentary bones and quadrate-articular jaw joint. Although a true dentary–squamosal joint with a developed condyle and squamosal glenoid evolved only in mammaliaforms, the morphology of *Riograndia* demonstrates that another clade of probainognathian cynodonts independently evolved a dentary–squamosal contact some 17 million years earlier.

*Brasilodon* has emerged as the sister to mammaliaforms in almost every phylogenetic study since its discovery[6–8,26,33–43]. Both tritheledontids[27,57–60] and tritylodontids[13,19,61–65] have been considered as the sister clade to mammaliaforms in previous studies (summarized previously[66]); however, neither hypothesis invalidates our interpretation that the dentary–squamosal contact evolved at least twice among probainognathian cynodonts. The derived condition in *Diarthrognathus* compared to *Riograndia* demonstrates that a more developed contact surface between the dentary and squamosal must have evolved independently within tritheledontids and mammaliaforms (Fig. 4). Tritheledontids are united by a suite of shared characters not possessed by mammaliaforms[37,67–69], so it is unlikely that they would represent a grade of cynodonts leading towards mammaliaforms to the exclusion of *Brasilodon* and tritylodontids. Considering the current spread of phylogenetic results in the literature, our conclusions on homoplasy present within the jaw joints of derived probainognathians are well supported. To address what our revised morphological descriptions mean for the phylogenetic position of taxa, we coded the new data obtained in this study (Supplementary Information) and found little change to the topology of previously published trees. *Brasilodon* remains the sister taxon to Mammaliaformes alongside the poorly known *Botucaraitherium*, to the exclusion of tritylodontids and ictidosaurs successively. The strict consensus tree of eight most parsimonious trees (MPTs) shows the ictidosaur clade (*Riograndia*, *Irajatherium*, *Diarthrognathus* + *Pachygenelus*) as monophyletic and sister to the clade composed of tritylodontids, *Botucaraitherium/Brasilodon* plus Mammaliaformes (Supplementary Fig. 1). In four out of the eight MPTs, *Riograndia* was found to be the sister of *Irajatherium* in a clade that is itself the sister to *Diarthrognathus* + *Pachygenelus*, whereas, in the other four MPTs, *Riograndia* was found to be the sister of *Diarthrognathus* + *Pachygenelus*, with *Irajatherium* recovered as the most basal ictidosaur. We found similar results using Bayesian analysis, with Ictidosauria forming a monophyletic group that, alongside *Therioherpeton*, forms the sister to a clade containing tritylodontids, *Botucaraitherium/Brasilodon* plus Mammaliaformes (Supplementary Fig. 2). The more basal position of a monophyletic Ictidosauria compared to tritylodontids and *Brasilodon* in all of the generated trees cements our conclusion that the dentary–squamosal contact evolved twice within Cynodontia. The use of this character as a defining feature of mammals is therefore problematic as it arose not only within non-mammalian mammaliaforms but also separately within probainognathian cynodonts. The evolution of a distinct condyle and glenoid, specifically, separates the dentary–squamosal joint in Mammaliaformes from the more basal dentary–squamosal lateral contact in derived cynodonts[70].

## Jaw joint innovation in Cynodontia

The morphology of *Brasilodon*, ictidosaurs and tritylodontids demonstrates that considerable experimentation with jaw joint morphology occurred within derived probainognathians, both in the two independent acquisitions of the dentary–squamosal jaw contact and the loss of the surangular-squamosal jaw joint (Fig. 4). This is in contrast to both basal probainognathians and cynognathians, the other large radiation of eucynodonts (Fig. 1), which show little variation in their jaw joint configuration. Among cynognathians, derived forms like the traversodontid *Massetognathus* possess a more developed articular facet on the squamosal and a more lateral articular surface on the articular boss than basal taxa like *Diademodon*[2], but the general topology of the bones remains the same. Low disparity in the cynognathian jaw joint contrasts with the general pattern of their evolution, which has previously been characterized by higher diversity, morphological disparity and character evolutionary rates compared to probainognathians[61], alongside the evolution of highly complex food processing and tooth replacement systems[71]. The surangular-squamosal joint of basal probainognathians generally resembles that of derived cynognathians, with a laterally placed squamosal flange similar to taxa like *Massetognathus*[2]. Initial descriptions of the early Carnian cynodont *Probainognathus* described the secondary articulation with the squamosal as involving both the surangular and the dentary[72,73], a description later repeated by Allin[3]. If accurate, this would constitute a third, older independent evolution of dentary–squamosal contact in probainognathians, but subsequent works[4,5,26] state that the secondary articulation in *Probainognathus* involved only the surangular, as in most other eucynodonts.

This raises the question of why derived probainognathians show much more plasticity in jaw articulation than other non-mammaliaform cynodonts. It has been hypothesized that the reorganization of muscles to a more mammalian arrangement reduced the mechanical load on the primary jaw joint, allowing for a reduction in the postdentary bones and promoting the evolution of secondary articulations[3,30,46,74]. It is likely that the majority of muscle reorganization had already occurred at the base of Eucynodontia[31], with the exception being the shift of the pterygoideus from the postdentary rod to the dentary[3,30,31,75,76]. However, uncertainty surrounding osteological correlates for the pterygoideus has obscured the exact timing of this shift[3,31,56,77]. One possible correlate, the angular process of the dentary (that may or may not be homologous to the mammalian angular process[77]), evolved sporadically across eucynodonts and occurs in many cynognathians; it is extremely pronounced in taxa such as *Massetognathus* and *Exaeretodon*. Cynognathians may therefore have evolved a fully mammalian masticatory muscle arrangement but never evolved the diversity in jaw articulation that derived probainognathians did. Moreover, recent biomechanical studies have shown that the musculoskeletal system of the cynodont jaw can be evolutionarily labile without affecting jaw joint stress[78], and that changing the arrangement of masticatory muscles probably had little effect on the reaction forces around the quadrate–articular joint[32]. Rearrangement of muscles is therefore unlikely to be solely responsible for the increased innovation in derived probainognathians.

Our findings for *Brasilodon* and *Riograndia* are more consistent with the hypothesis that body-size reduction would have facilitated, and perhaps promoted, experimentation with jaw joint morphology in derived probainognathian cynodonts including homoplastic evolution of a dentary–squamosal contact[32]. Experimentation with jaw joint morphology in derived probainognathians, but not in other eucynodonts, is consistent with a reduction in absolute jaw joint stress accompanying smaller body sizes in these taxa. The jaw joint of *Brasilodon* was apparently capable of functioning without a reinforcing secondary contact, and articulation took place solely through the small quadrate–articular joint. Miniaturization has been correlated with the evolution of other key mammalian traits including metabolic

rate, thermoregulation and insectivorous ecology[13–18,79–81], and it seems likely that small body size was also involved in driving evolution of the mammalian jaw joint. However, the relatively larger size of some tritylodontids like *Kayentatherium* suggests further complexity, and other factors such as muscle reorganization, feeding ecology and masticatory behaviour may have contributed[30,49,74,82]. The relatively larger postdentaries of *Riograndia* alongside a secondary dentary–squamosal contact represent a different instance of evolutionary experimentation that occurred earlier than the acquisition of a secondary jaw joint for load bearing in mammaliaforms; one that appears to be characterized by greater jaw joint robustness, possibly to cope with a more mechanically demanding diet, even at small body sizes.

## Conclusions

3D μCT data of *Brasilodon* and *Riograndia* illustrate homoplastic evolution of the dentary–squamosal contact within probainognathian cynodonts and reveal the oldest occurrence of this feature in stem mammals. Although only Mammaliaformes evolved a fully developed dentary condyle and squamosal glenoid, the presence of an extensive lateral dentary–squamosal contact in *Riograndia* and the thickened, more condyle-like posterior rim of the articular process in more derived Jurassic tritheledontids demonstrates that at least two cynodont lineages were evolving features of the jaw articulation that are typically associated with mammals. Although traditionally considered a mammalian character, both non-mammalian mammaliaforms and non-mammaliaform probainognathian cynodonts evolved a dentary–squamosal contact, making the use of this character in defining the unique features of mammals problematic. Despite occurring in multiple lineages of derived probainognathians, innovation in jaw joint morphology occurs relatively infrequently among other groups of cynodonts, and the miniaturization of the cynodont body plan may have been vital to the development of the dentary–squamosal jaw joint, demonstrating the importance of studying the cynodont–mammaliaform transition for understanding mammalian evolution.

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

## Methods

### Sampling

Many cynodont specimens from the Triassic of Brazil are available for study, but we chose to focus on those with well-preserved lower jaw anatomy, especially those with postdentary bones. Numerous specimens used in this study were loaned for scanning from the Universidad Federal do Rio Grande do Sul (Porto Alegre, Brazil). Further information for each specimen is provided in Supplementary Table 1.

### Scanning and data processing

The 26 specimens used for this study were scanned either in the Nikon XTH225 ST Micro CT scanner at the University of Bristol, the Nikon Metrology HMX ST 225 or Zeiss Xradia Versa 520 at the Natural History Museum, London, the Phoenix High-Resolution X-ray x|s Nanofocus system at the University of Finland or the SkyScan 1173 High Energy Micro-CT system at the Pontifical Catholic University of Rio Grande do Sul. Transverse TIFF slice number in the scanned specimens ranged from around 1,500 to >3,000 and the scans produced had a voxel size range of 0.0057 mm to 0.026 mm. Scans were processed using the 3D visualization software Avizo 3D 2021.1 (Thermo Fisher Scientific) or Dragonfly 2022.2 (Object Research Systems) to transform scan data into 3D reconstructions. Density thresholding was used to distinguish fossil material from air and surrounding matrix. Scans were then segmented manually slice by slice, interpolating across no more than five slices at a time, to separate the relevant bones. The dentary was segmented in each case alongside any postdentary elements that could be recovered, including disarticulated or fragmented material. Where available, the quadrate, quadratojugal and squamosal were also segmented to allow description of the jaw joint. Reconstructions of complete postdentary anatomy were estimated for *Brasilodon* and *Riograndia* by manipulating individual bone surfaces and placing them into the hypothesized life position, informed by in situ position on other specimens. Further repairs for missing elements were carried out in Blender 3.4, details of which are provided below.

***B. quadrangularis.*** Specimens included in *Brasilodon* were initially considered to represent three separate species[10–12] but, more recently, have been suggested to represent an ontogenetic series (refs. 38,86,87 and additional work in preparation by the authors), which is how they are treated in this work. All nine *Brasilodon* specimens used in this study (Supplementary Table 1) were recovered from the *Riograndia* Assemblage Zone, Candelária Sequence, Santa Maria Supersequence in the Rio Grande do Sul State, Brazil[11]. To create the reconstruction of shown in Fig. 4, the dentary, postdentaries and splenials of UFRGS-PV-1043-T were first moved into the hypothesized life position and minor cracks were repaired using the segmentation editor in Avizo[88]. Some postdentary elements from the right jaw were mirrored over to the left jaw, which had a complete coronoid process. The coronoid was segmented from UFRGS-PV-628-T (Fig. 2g) and scaled using equivalent measurements of the jaws and skull. Once moved into the life position, the remaining missing portions were repaired in Blender v.3.4 using previously established protocols[88]. The initial segmented surfaces of UFRGS-PV-1043-T before restoration can be viewed in Extended Data Fig. 1.

***R. guaibensis.*** Nine specimens of *Riograndia* were used in this study, all of which originate from the *Riograndia* AZ of the Candelária Sequence[6]. The reconstruction of the jaw of *Riograndia* shown in Fig. 4 was amalgamated using elements from UFRGS-PV-596-T (dentary), UFRGS-PV-833-T (postdentaries and teeth) and UFRGS-PV-624-T (coronoid). The broken elements of the dentary of UFRGS-PV-596-T were moved into the life position relative to each other in Avizo and minor cracks were repaired in the segmentation editor[88]. The remaining elements were scaled from their respective specimens using

equivalent measurements of the skull and jaws and positioned in the reconstruction based on information from additional specimens. The postdentary bones of UFRGS-PV-624-T were also reconstructed in Avizo to compare to those of UFRGS-PV-833-T. Once positioned, all elements were exported to Blender v.3.4 as .stl files and restored using information from additional specimens to produce the final reconstruction[88]. The various stages of this process are shown in Extended Data Fig. 2.

***O. major.*** Seven specimens of *Oligokyphus* were scanned for use in this study, all of which are held at the Natural History Museum, London. These specimens were excavated by W. Künhe from the Windsor Hilll quarry in Somerset, a terrestrial fissure deposit that has been interpreted as Hettangian to early Sinemurian in age[8,50]. No complete jaw or skull exists for *O. major*, although between the specimens scanned in this study, the dentary at least can be reconstructed with some accuracy. Our µCT scans corroborated the findings of previous descriptions, including the absence of both a dentary–squamosal joint and a surangular–squamosal joint (Extended Data Fig. 3 and Supplementary Information for more details). Details of all specimens used are provided in Supplementary Table 1.

### Phylogenetic analysis

To ensure that our conclusions about character evolution within derived probainognathians are robust, we performed a phylogenetic analysis of non-mammaliaform cynodonts with updated characters based on our anatomical interpretations. The full character list and an outline of all changes made can be found in the Supplementary Information. The final data matrix includes 158 morphological characters and 42 terminal taxa of specific level. The data matrix was analysed in TNT (v.1.5)[89] under a heuristic search including random addition sequence + tree bisection reconnection, with 1,000 replicates of Wagner trees (with random seed = 0) and using tree bisection reconnection and branch swapping (holding 10 trees save per replication). All characters were considered as unordered and with the same weight. Time calibration of the strict consensus tree (Supplementary Fig. 1) was performed in R using the 'equal' method implemented in the paleotree package[90]. The first and last appearance dates (Supplementary Table 2) were sourced from the literature[8,42,61,91]. Moreover, a Bayesian phylogenetic analysis was performed in MrBayes (v.3.2.7)[92] with four Markov chain Monte Carlo chains, using a discrete morphological character model. The analysis was run over 1 million generations and was sampled trees every 100 generations which, using a burn-in of 25%, resulted in 7,500 retained trees (Supplementary Fig. 2).

### Reporting summary

Further information on research design is available in the Nature Portfolio Reporting Summary linked to this article.

## Data availability

The data matrix used in the phylogenetic analysis is provided in separate files in Mesquite and TNT formats. Videos showing rotating views of the 3D surface models of UFRGS-PV-628-T (*Brasilodon*), UFRGS-PV-929-T (*Brasilodon*), UFRGS-PV-1030-T (*Brasilodon*), UFRGS-PV-1043-T (*Brasilodon*), MPDC-1B1 (*Riograndia*), UFRGS-PV-596-T (*Riograndia*), UFRGS-PV-833-T (*Riograndia*) and UFRGS-PV-1319-T (*Riograndia*); STL files of figured specimens; and jaw reconstructions for *Brasilodon* and *Riograndia* shown in Fig. 4 (the method for reconstruction described above) are available at the University of Bristol data repository data. bris (https://doi.org/10.5523/bris.2ie8neiry701e23iayx9gdsytv).

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

**Acknowledgements** We thank E. Martin-Silverstone, T. Davies, M. Day, V. Fernandez and B. Clark, A. Kallonen and I. Corfe for assistance and use of CT facilities for scanning the specimens used in this study. Exploratory segmentations and observations on some of the scans were carried out by C. Salcido, S. Holpin and C. Clark. We thank H. Francischini (UFRGS) and Silvia Tachestto (MPDC) for granting us access to *Brasilodon* and *Riograndia* fossils. Specimen photos were skilfully taken by L. Flávio Lopes (UFRGS). We thank F. Abdala (CONICET-Instituto Miguel Lillo) for providing comparative photos of *Pachygenelus* specimens. This work was funded by NERC (Natural Environment Research Council) grant NE/K01496X/1 to E.J.R. and P.G.G. J.R.G.R. is supported by the Biotechnology and Biological Sciences Research Council-funded South West Biosciences Doctoral Training Partnership (grant BB/T008741/1). The Conselho Nacional de Desenvolvimento Científico e Tecnológico (CNPq, Brazil, grants 307938/2019-0; 311251/2021-8; 406902/2022-4; 308515/2023-4) and Fundação de Amparo à Pesquisa do Estado do Rio Grande do Sul (FAPERGS 19/2551-0000719-1) supported M.B.S., C.L.S. and A.G.M. The Fundação Carlos Chagas Filho de Amparo à Pesquisa do Estado do Rio de Janeiro (FAPERJ) supported M.B.S. (grant E-26/201.066/2021). A.G.M. was awarded the Benjamin Meaker Visiting Professorship of the University of Bristol and is also supported by CONICET-PIBAA 1137, PICT 2020-SERIEA-01498 and MILENIO NCN2023-025.

**Author contributions** The study was formulated by A.G.M., P.G.G. and E.J.R. μCT scanning was organized by A.G.M., P.G.G., M.B.S., C.L.S. and E.J.R. and segmentation and interpretation of the scans was done by J.R.G.R. Phylogenetic analyses were performed by A.G.M. and J.R.G.R. The manuscript was written by J.R.G.R., A.G.M., P.G.G and E.J.R. with input from all of the authors.

**Competing interests** The authors declare no competing interests.

**Additional information**
**Correspondence and requests for materials** should be addressed to James R. G. Rawson, Agustín G. Martinelli or Emily J. Rayfield.

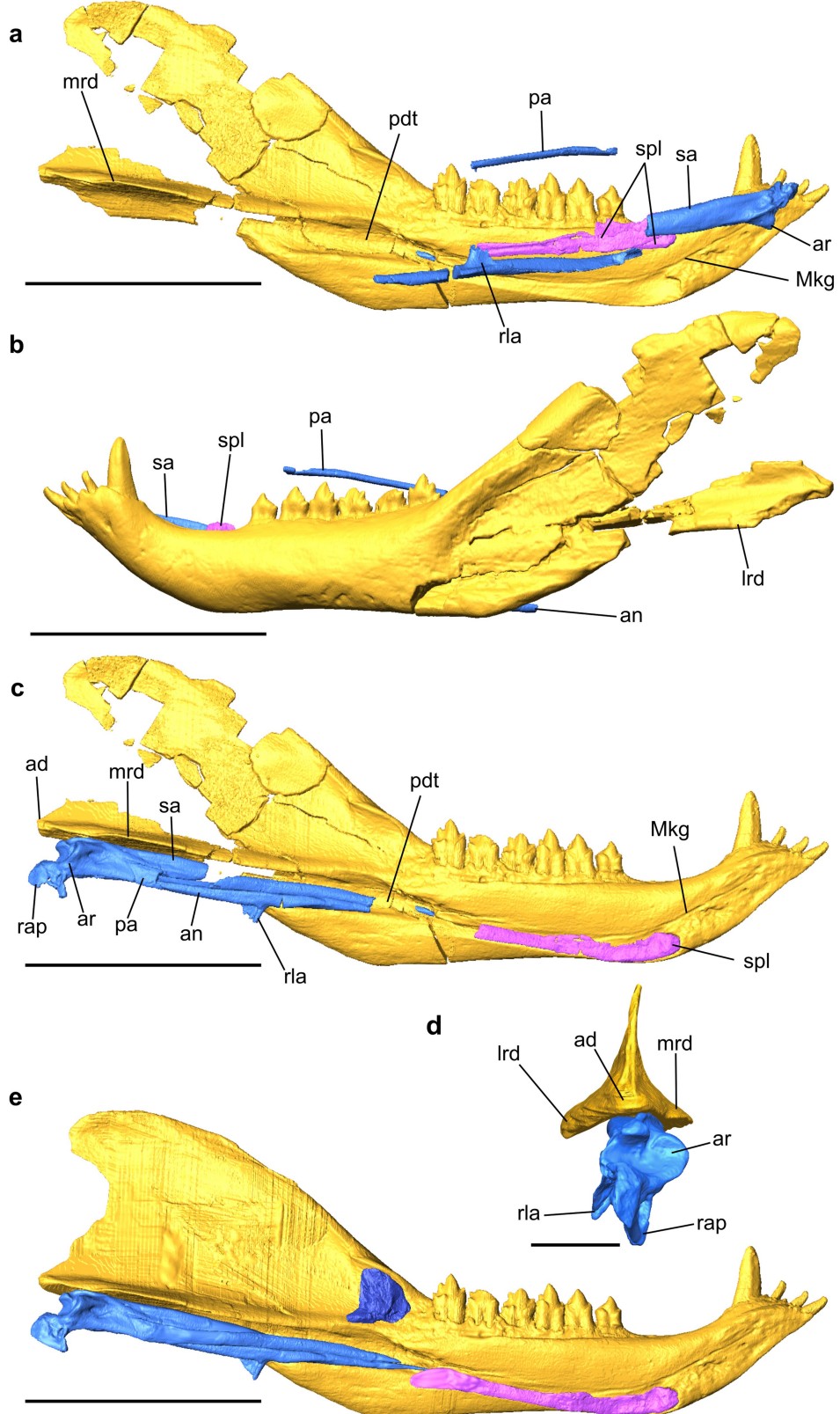

**Extended Data Fig. 1 | Reconstruction of the jaw of *Brasilodon quadrangularis*.** **a**, Medial view of the postdentary bones and left dentary of UFRGS-PV-1043-T, as preserved; **b**, lateral view of same; **c**, medial view of same showing postdentary elements moved into hypothesized life position, including mirrored elements; **d**, fully reconstructed posterior view of the articular process of the dentary and postdentaries; **e**, medial view of fully restored lower jaw including all repairs (see methods) and scaled coronoid from UFRGS-PV-628-T. Abbreviations as in main text plus: spl, splenial. Scale bars represent 10 mm or 2 mm (C).

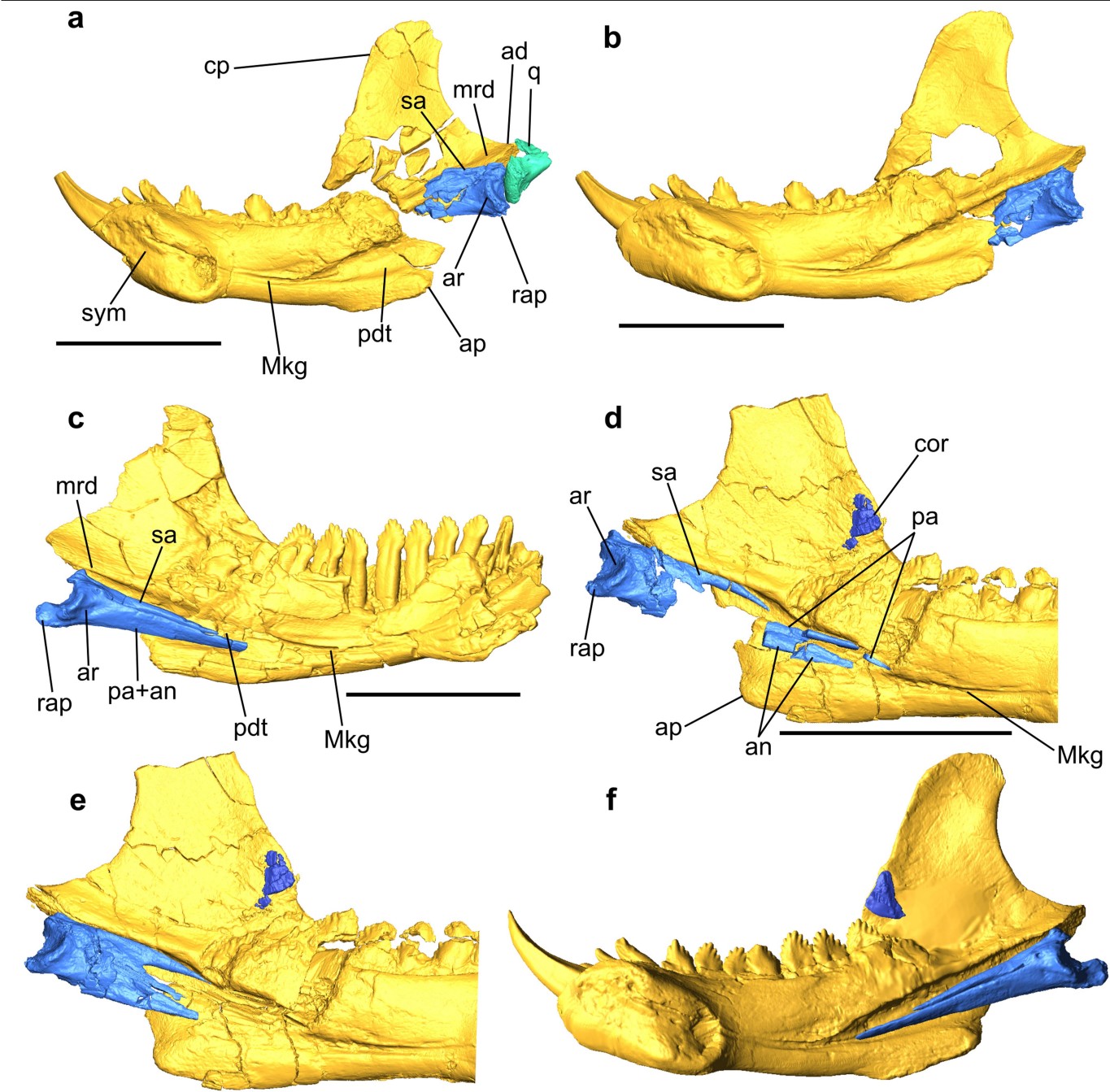

**Extended Data Fig. 2 | Reconstruction of the jaw of *Riograndia guaibensis*.**
**a**, Medial view of the right dentary and postdentary bones of UFRGS-PV-596-T, as preserved; **b**, medial view of same showing fragments of dentary and postdentary elements moved into hypothesized life position; **c**, medial view of the left jaw of UFRGS-PV-833-T, postdentary elements moved into hypothesized life position; **d**, posterior left jaw of UFRGS-PV-624-T in medial view, as preserved; **e**, medial view of same with coronoid fragments and postdentary bone fragments restored to hypothesized life position; **f**, medial view of fully restored right jaw including all repairs (see methods). All scale bars represent 10 mm.

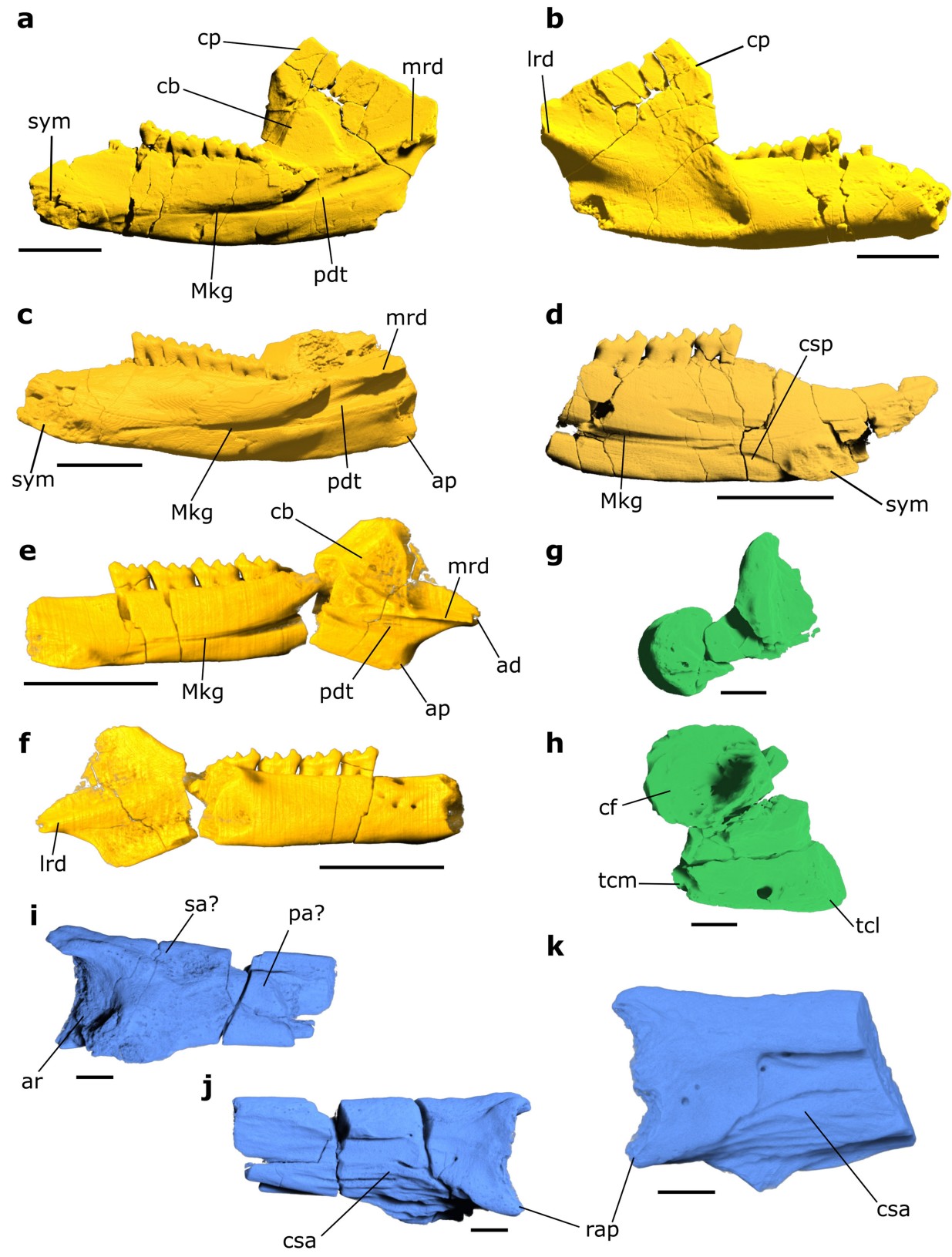

**Extended Data Fig. 3 | Surface models of specimens of *Oligokyphus major* scanned in this study. a**, Medial view of NHMUK-PV-R7119, a right dentary; **b**, lateral view of same; **c**, medial view of NHMUK-PV-R7121, a partial right dentary; **d**, medial view of NHMUK-PV-R7373, a partial left dentary; **e**, medial view of NHMUK-PV-R7204, a right dentary; **f**, lateral view of same; **g**, medial view of NHMUK-PV-R7196, a right quadrate; **h**, posterior view of same; **i**, medial view of NHMUK-PV-R7189, left posterior postdentaries; **j**, lateral view of same; **k**, lateral view of NHMUK-PV-R1790, right posterior postdentaries. Anatomical abbreviations as in main text plus: **cf**, contact facet; **cb**, coronoid boss; **csa**, contact surface for the angular; **csp**, contact surface for the splenial; **tcl**, lateral trochlear condyle; **tcm**, medial trochlear condyle Scale bars represent 10 mm (**a**–**f**) or 1 mm (**g**–**k**).

# Reporting Summary

Please do not complete any field with "not applicable" or n/a. Refer to the help text for what text to use if an item is not relevant to your study.
For final submission: please carefully check your responses for accuracy; you will not be able to make changes later.

## Statistics

For all statistical analyses, confirm that the following items are present in the figure legend, table legend, main text, or Methods section.

| n/a | Confirmed | |
|---|---|---|
| ☒ | ☐ | The exact sample size (*n*) for each experimental group/condition, given as a discrete number and unit of measurement |
| ☒ | ☐ | A statement on whether measurements were taken from distinct samples or whether the same sample was measured repeatedly |
| ☒ | ☐ | The statistical test(s) used AND whether they are one- or two-sided *Only common tests should be described solely by name; describe more complex techniques in the Methods section.* |
| ☒ | ☐ | A description of all covariates tested |
| ☒ | ☐ | A description of any assumptions or corrections, such as tests of normality and adjustment for multiple comparisons |
| ☒ | ☐ | A full description of the statistical parameters including central tendency (e.g. means) or other basic estimates (e.g. regression coefficient) AND variation (e.g. standard deviation) or associated estimates of uncertainty (e.g. confidence intervals) |
| ☒ | ☐ | For null hypothesis testing, the test statistic (e.g. *F*, *t*, *r*) with confidence intervals, effect sizes, degrees of freedom and *P* value noted *Give P values as exact values whenever suitable.* |
| ☒ | ☐ | For Bayesian analysis, information on the choice of priors and Markov chain Monte Carlo settings |
| ☒ | ☐ | For hierarchical and complex designs, identification of the appropriate level for tests and full reporting of outcomes |
| ☒ | ☐ | Estimates of effect sizes (e.g. Cohen's *d*, Pearson's *r*), indicating how they were calculated |

*Our web collection on statistics for biologists contains articles on many of the points above.*

## Software and code

Policy information about availability of computer code

| Data collection | Avizo 3D 2021.1 (Thermo Fisher Scientific, Waltham, Massachusetts, U.S.A.); Dragonfly 2022.2 (Object Research Systems, Montréal, Québec, Canada) |
|---|---|
| Data analysis | Blender 3.4; TNT v.1.5; MrBayes v. 3.2.7; R version 4.3.1; paleotree package version 3.4.5 |

For manuscripts utilizing custom algorithms or software that are central to the research but not yet described in published literature, software must be made available to editors and reviewers. We strongly encourage code deposition in a community repository (e.g. GitHub). See the Nature Portfolio guidelines for submitting code & software for further information.

## Data

Policy information about availability of data

All manuscripts must include a data availability statement. This statement should provide the following information, where applicable:
- Accession codes, unique identifiers, or web links for publicly available datasets
- A description of any restrictions on data availability
- For clinical datasets or third party data, please ensure that the statement adheres to our policy

The data matrix used in the phylogenetic analysis is provided in separate files in Mesquite and TNT formats. Videos showing rotating views of the 3D surface models of UFRGS-PV-628-T (Brasilodon), UFRGS-PV-929-T (Brasilodon), UFRGS-PV-1030-T (Brasilodon), UFRGS-PV-1043-T (Brasilodon), MPDC-1B1 (Riograndia), UFRGS-PV-596-T (Riograndia), UFRGS-PV-833-T (Riograndia) and UFRGS-PV-1319-T (Riograndia), STL files of figured specimens, and jaw reconstructions for Brasilodon and Riograndia shown in Fig. 4 (method for reconstruction described above), are available at the University of Bristol data repository, data.bris, at https://doi.org/10.5523/bris.2ie8neiry701e23iayx9gdsytv.

# Research involving human participants, their data, or biological material

Policy information about studies with human participants or human data. See also policy information about sex, gender (identity/presentation), and sexual orientation and race, ethnicity and racism.

| | |
|---|---|
| Reporting on sex and gender | N/A |
| Reporting on race, ethnicity, or other socially relevant groupings | N/A |
| Population characteristics | N/A |
| Recruitment | N/A |
| Ethics oversight | N/A |

Note that full information on the approval of the study protocol must also be provided in the manuscript.

# Field-specific reporting

Please select the one below that is the best fit for your research. If you are not sure, read the appropriate sections before making your selection.

☐ Life sciences ☐ Behavioural & social sciences ☒ Ecological, evolutionary & environmental sciences

For a reference copy of the document with all sections, see nature.com/documents/nr-reporting-summary-flat.pdf

# Life sciences study design

All studies must disclose on these points even when the disclosure is negative.

| | |
|---|---|
| Sample size | N/A |
| Data exclusions | N/A |
| Replication | N/A |
| Randomization | N/A |
| Blinding | N/A |

# Behavioural & social sciences study design

All studies must disclose on these points even when the disclosure is negative.

| | |
|---|---|
| Study description | N/A |
| Research sample | N/A |
| Sampling strategy | N/A |
| Data collection | N/A |
| Timing | N/A |
| Data exclusions | N/A |
| Non-participation | N/A |
| Randomization | N/A |

# Ecological, evolutionary & environmental sciences study design

All studies must disclose on these points even when the disclosure is negative.

| | |
|---|---|
| Study description | Micro-CT investigation into the anatomy of three non-mammaliaform cynodonts |
| Research sample | Scans of 27 non-mammaliaform cynodont specimens |
| Sampling strategy | Sampled accoring to those specimens that best preserve relevant anatomy |
| Data collection | Details of each micro-CT scan are available in manuscript |
| Timing and spatial scale | N/A |
| Data exclusions | N/A |
| Reproducibility | N/A |
| Randomization | N/A |
| Blinding | N/A |

Did the study involve field work? ☐ Yes ☒ No

## Field work, collection and transport

| | |
|---|---|
| Field conditions | N/A |
| Location | N/A |
| Access & import/export | N/A |
| Disturbance | N/A |

# Reporting for specific materials, systems and methods

We require information from authors about some types of materials, experimental systems and methods used in many studies. Here, indicate whether each material, system or method listed is relevant to your study. If you are not sure if a list item applies to your research, read the appropriate section before selecting a response.

### Materials & experimental systems

| n/a | Involved in the study |
|---|---|
| ☒ | ☐ Antibodies |
| ☒ | ☐ Eukaryotic cell lines |
| ☐ | ☒ Palaeontology and archaeology |
| ☒ | ☐ Animals and other organisms |
| ☒ | ☐ Clinical data |
| ☒ | ☐ Dual use research of concern |
| ☒ | ☐ Plants |

### Methods

| n/a | Involved in the study |
|---|---|
| ☒ | ☐ ChIP-seq |
| ☒ | ☐ Flow cytometry |
| ☒ | ☐ MRI-based neuroimaging |

## Antibodies

| | |
|---|---|
| Antibodies used | N/A |
| Validation | N/A |

# Eukaryotic cell lines

Policy information about cell lines and Sex and Gender in Research

| | |
|---|---|
| Cell line source(s) | N/A |
| Authentication | N/A |
| Mycoplasma contamination | N/A |
| Commonly misidentified lines (See ICLAC register) | N/A |

# Palaeontology and Archaeology

| | |
|---|---|
| Specimen provenance | Some specimens were collected from the Riograndia Assemblage Zone, Candelária Sequence, Rio Grande du sul, Brazil and others from Windsor Hilll quarry, Somerset, England |
| Specimen deposition | All specimens are accessioned in recognised museums. Brasilodon and Riograndia fossils are accessioned into the collections of the UFRGS (Universidad Federal do Rio Grande do Sul, Porto Alegre, Brazil). Here they have been studied extensively by our Brazilian and Argentinian co-authors. The Oligokyphus specimens are housed in the Natural History Museum, London. |
| Dating methods | Dates used for time-calibrating the phylogenetic tree were sourced from previously published literature (see methods) |

☒ Tick this box to confirm that the raw and calibrated dates are available in the paper or in Supplementary Information.

| | |
|---|---|
| Ethics oversight | |

Note that full information on the approval of the study protocol must also be provided in the manuscript.

# Animals and other research organisms

Policy information about studies involving animals; ARRIVE guidelines recommended for reporting animal research, and Sex and Gender in Research

| | |
|---|---|
| Laboratory animals | N/A |
| Wild animals | N/A |
| Reporting on sex | N/A |
| Field-collected samples | N/A |
| Ethics oversight | N/A |

Note that full information on the approval of the study protocol must also be provided in the manuscript.

# Clinical data

Policy information about clinical studies

All manuscripts should comply with the ICMJE guidelines for publication of clinical research and a completed CONSORT checklist must be included with all submissions.

| | |
|---|---|
| Clinical trial registration | N/A |
| Study protocol | N/A |
| Data collection | N/A |
| Outcomes | N/A |

# Dual use research of concern

Policy information about dual use research of concern

## Hazards

Could the accidental, deliberate or reckless misuse of agents or technologies generated in the work, or the application of information presented in the manuscript, pose a threat to:

|  | No | Yes |
|---|---|---|
| Public health | ☒ | ☐ |
| National security | ☒ | ☐ |
| Crops and/or livestock | ☒ | ☐ |
| Ecosystems | ☒ | ☐ |
| Any other significant area | ☒ | ☐ |

### Experiments of concern

Does the work involve any of these experiments of concern:

|  | No | Yes |
|---|---|---|
| Demonstrate how to render a vaccine ineffective | ☒ | ☐ |
| Confer resistance to therapeutically useful antibiotics or antiviral agents | ☒ | ☐ |
| Enhance the virulence of a pathogen or render a nonpathogen virulent | ☒ | ☐ |
| Increase transmissibility of a pathogen | ☒ | ☐ |
| Alter the host range of a pathogen | ☒ | ☐ |
| Enable evasion of diagnostic/detection modalities | ☒ | ☐ |
| Enable the weaponization of a biological agent or toxin | ☒ | ☐ |
| Any other potentially harmful combination of experiments and agents | ☒ | ☐ |

# Plants

| Seed stocks | N/A |
|---|---|
| Novel plant genotypes | N/A |
| Authentication | N/A |

# ChIP-seq

## Data deposition

☐ Confirm that both raw and final processed data have been deposited in a public database such as GEO.

☐ Confirm that you have deposited or provided access to graph files (e.g. BED files) for the called peaks.

| Data access links
May remain private before publication. | N/A |
|---|---|
| Files in database submission | N/A |
| Genome browser session
(e.g. UCSC) | N/A |

## Methodology

| Replicates | N/A |
|---|---|
| Sequencing depth | N/A |
| Antibodies | N/A |
| Peak calling parameters | N/A |
| Data quality | N/A |
| Software | N/A |

# Flow Cytometry

## Plots

Confirm that:

☐ The axis labels state the marker and fluorochrome used (e.g. CD4-FITC).

☐ The axis scales are clearly visible. Include numbers along axes only for bottom left plot of group (a 'group' is an analysis of identical markers).

☐ All plots are contour plots with outliers or pseudocolor plots.

☐ A numerical value for number of cells or percentage (with statistics) is provided.

## Methodology

| | |
|---|---|
| Sample preparation | N/A |
| Instrument | N/A |
| Software | N/A |
| Cell population abundance | N/A |
| Gating strategy | N/A |

☐ Tick this box to confirm that a figure exemplifying the gating strategy is provided in the Supplementary Information.

# Magnetic resonance imaging

## Experimental design

| | |
|---|---|
| Design type | N/A |
| Design specifications | N/A |
| Behavioral performance measures | N/A |

| | |
|---|---|
| Imaging type(s) | N/A |
| Field strength | N/A |
| Sequence & imaging parameters | N/A |
| Area of acquisition | N/A |

Diffusion MRI    ☐ Used    ☐ Not used

## Preprocessing

| | |
|---|---|
| Preprocessing software | N/A |
| Normalization | N/A |
| Normalization template | N/A |
| Noise and artifact removal | N/A |
| Volume censoring | N/A |

## Statistical modeling & inference

| | |
|---|---|
| Model type and settings | N/A |
| Effect(s) tested | N/A |

Specify type of analysis:    ☐ Whole brain    ☐ ROI-based    ☐ Both

| Statistic type for inference | N/A |
|---|---|

(See Eklund et al. 2016)

| Correction | N/A |
|---|---|

## Models & analysis

| n/a | Involved in the study |
|---|---|
| ☒ ☐ | Functional and/or effective connectivity |
| ☒ ☐ | Graph analysis |
| ☒ ☐ | Multivariate modeling or predictive analysis |

| Functional and/or effective connectivity | N/A |
|---|---|

| Graph analysis | N/A |
|---|---|

| Multivariate modeling and predictive analysis | N/A |
|---|---|

