## [Peer Review File · Nature]

Manuscript Title: Brazilian fossils reveal homoplasy in the oldest mammalian jaw joint

Reviewer Comments & Author Rebuttals

Reviewer Reports on the Initial Version:

Referees' comments:

Referee #1 (Remarks to the Author):

Comments on "South American stem mammals reveal homoplastic evolution of the oldest mammalian jaw joint" by Rawson et al. (2023-12-22267).

The report used micro-CT scanning to examine the 3D jaw joint anatomy of three non-mammaliaform cynodonts: *Brasilodon quadrangularis*, *Riograndia guaibensis*, and *Oligokyphus major*. This is another study with a focus on joint morphology and changes, particularly the formation of the secondary dentary-squamosal joint, that took place across the cynodont-mammaliaform transition; the dentary-squamosal joint has been commonly considered a key mammalian feature.

The main findings of the study include two aspects: 1. *Brasilodon quadrangularis*, often placed as a sister-taxon of Mammaliaformes, does not have the dentary-squamosal joint, contrary to the original report in which such a joint was claimed to be present. 2. A distinct dentary-squamosal articulation was found to be present in *Riograndia*, which is a derived probainognathian but older than some related taxa, such as *Pachygenelus* and *Diarthrognathus* that were known to have a dentary-squamosal joint and the dentary condyle, albeit small, is developed, particularly so in *Diarthrognathus*.

The authors concluded that at least two cynodont lineages were evolving a more 'mammal-like' jaw articulation, one is the clade that contains *Riograndia*, *Pachygenelus*, and *Diarthrognathus*; the other is Mammaliaformes. The dentary-squamosal joint in *Riograndia* is currently the oldest confirmed one in cynodonts. The authors thought the development of the dentary-squamosal jaw joint may be attributable to the miniaturization of the cynodont body sizes.

We agree with the authors' general view that multiple experimentations of jaw joints in derived probainognathians had happened and that these are partly driven by or associated with the miniaturization of the cynodont body sizes.

Comments on the evidence

We think the evidence that shows the absence of the dentary-squamosal joint in *Brasilodon quadrangularis* is convincing, based on the shape of the articular process of the dentary and the shape of the squamosal, as shown in Fig. 2.

The evidence for the presence of the dentary-squamosal joint in *Riograndia* is less convincing. There is "a clear indentation on the ventromedial surface" of the squamosal. The dentary, however, does not seem to show a clear articular area, likely convex, that should contact the indentation (or facet) in the squamosal. The right mandible is quite complete as shown in Fig. 3a and 3b (UFRGS-PV-596-T), but its distal end of the articular process of the dentary seems incomplete; the left mandible

(UFRGS-PV-833-T) has its articular process of the dentary even less complete. Carefully reviewing the Extended Data Fig. 3, the stl, and mpg files of UFRGS-PV-596-T (many thanks for providing these great images!), our concern remains, that is, the articular process of the dentary does not show a convincing articular area that should be in contact with the “indentation” of the squamosal to form a joint, or this area was damaged in the studied specimens. We fully understand, however, that it is difficult to have “perfect” fossils that could unequivocally show the jaw joint in the fossil records; we regard the evidence provided by the authors as reasonable, but partial, and we will continue commenting on the rest of the manuscript.

Comments on interpretations/conclusion.

1. The authors have shown an inconsistency in using the terminology on the jaw articulation in question. For instance, in the abstract, it is “The classic mammalian dentary-squamosal jaw hinge therefore evolved more than once and is more evolutionary labile than previously considered.”. In other places, “we show that this ‘mammalian’ jaw joint evolved independently from that of mammal precursors,” and “a more ‘mammal-like’ jaw articulation.”

The authors could use the dentary-squamosal joint (articulation, hinge) without associating it with mammals; otherwise, the implication that the typical mammalian jaw joint is found in the non-mammaliaform cynodonts would create a conflict meaning or confusion, particularly for those who are not familiar with the structure and phylogenetic context. It’s probably better to say something like: the dentary-squamosal jaw joint, traditionally considered a typical mammalian feature, evolved more than once.

2. The main conclusion of the work is that the dentary-squamosal jaw joint is not unique to mammaliaforms but probably evolved independently more than once. Although the conclusion is probably supported by evidence, it is not a new idea. This parallel evolution of the dentary-squamosal joint was proposed at least since Crompton and Parkyn (1963).

Crompton and Parkyn (1963: 743, 707): stated: “It was shown that in *Diarthrognathus* and *Morganucodon* the dentary condyle develops at the posterior termination of this lateral ridge.” “Although the dentition and the proportions of the lower jaws in *Diarthrognathus* and *Morganucodon* are different, the structure of the articular processes of the dentary in both forms is strikingly similar. The structure of the postcanine teeth indicates that the two forms are not closely related so that this similarity was probably due to parallel evolution.”

We would view that the jaw joint of *Riograndia* provides additional evidence supporting the view that the dentary-squamosal joint evolved more than once in cynodonts; in particular, it shows an earlier and probably more primitive condition than that in *Diarthrognathus*.

3. The statement (sentence) at the end of the next last paragraph (before the Conclusion): “one that appears to prioritise jaw joint robustness even at small body sizes.” appears to be problematic at least to us. The term “prioritise” may not be appropriate; this is not an intelligent design; it is by natural selection. More importantly, why did these animals prioritise jaw joint robustness even at small body sizes? It seems that you concluded something that you didn’t discuss in the article.

4. The current text could be considerably condensed. In general, we think the main text elaborates a bit too much on what has been known. It’s necessary to provide the context and background for the subject discussed in the manuscript, but there is no need to re-state much of what has been said or keep it at a minimum. Focus on what is the new evidence you have and what the new evidence tells us.

5. Figures 1 and 4 can be combined; much of the information and jaw figures are redundant. Fig. 4

displays the mandible shapes in a simplified phylogenetic frame and emphasizes the two independent occurrences of the dentary-squamosal joint. The same information can be displayed in Fig. 1 by marking the corresponding branches in green, orange, and blue, as in Fig. 4. The jaw figure bearing terminology (*Thrinaxodon*) at the lower left corner is redundant, which can be deleted (using the upper right one to show the terms, if that is necessary). Place the lower jaws of *Riograndia* (above) and *Brasilodon* (below) in the lower left area above the geological scale. This figure could be placed at the end to show the conclusion (morphology, age, independent evolutions, etc.).

Minor comments:

1. In the abstract, "have much to offer on this debate." Which is the debate? You mentioned a "question" but not a debate. Do you mean the question you mentioned above?
2. In "...predating that seen in Mammaliaformes by approximately 20 million years 7,8", Double check ref 7 is the right one to cite here.
3. The subtitle "Main" appears awkward – could this be Introduction? This section can be greatly condensed.
4. In "the origin of these features is a highly significant event in mammalian evolution 4,5,18,19." It's probably better to put Allin 1975 before A&H 1992 in the citation sequence - a trivial thing, but a historical fact.
5. In "The definitive mammalian middle ear evolved within crown mammals during the Mesozoic," It's probably safer to say "mammaliaformes" than "crown mammals" because of the unstable placement of allotherians within the mammaliaforms. In some analyses, allotherians, particularly haramiyidans, that have the DMME were placed outside of the crown mammals.
6. In "serve as the point of articulation between the skull and lower jaw." Better replace skull with cranium.
7. The section "Jaw articulation in *Oligokyphus major*" may not be necessary to include in the main text; it provides very little if any, new information.
8. In "...yet it does appear that the later forms were approaching a condition closer to that seen in *Morganucodon*." It is probably better "the later forms of tritheledontids" to be clear.
9. In discussing the reorganisation of muscles: "This raises the question of why derived probainognathians show..., allowing reduction of the postdentary bones and promoting the evolution of secondary articulations 5,27,67." It's better to cite Crompton & Parkyn, 1963 because they already explored this issue in describing the lower jaw of *Diarthrognathus*.
10. Put the year (1963) for Crompton & Parkyn (ref. 42) in the reference.
11. In the figure 1 legend, does the sentence "Topology is based on 31 and the phylogenetic analysis within this study" mean the topology is based on two phylogenetic results? How this was done should be explained in Phylogenetic analysis in Methods.
12. It would be better if some measurements were provided (jaw length, postdentary unit size, if possible) so that the size and relative size could be better appreciated. These numbers may also be useful to echo the discussion on miniaturisation.

Referee #2 (Remarks to the Author):

Thank you for the opportunity to review the manuscript 2023-12-22267 by Rawson et al., which I found of great interest. In this manuscript, the authors utilized micro-CT scanning to provide further information on the jaw joint morphology for three probainognathian cynodont taxa: *Brasilodon quadrangularis*, *Riograndia guaibensis*, and *Oligokyphus major*. The original description of *Brasilodon quadrangularis* suggested incipient contact between the dentary and squamosal. Here, they present a new interpretation that *Brasilodon* lacks an incipient dentary condyle and squamosal glenoid, and the jaws articulate solely via the quadrate-articular joint, more similar to that of tritylodontids. Based on the new information, they concluded that there is homoplastic evolution of the dentary-squamosal jaw joint within probainognathian cynodonts.

While this manuscript provides new insights into the origin of the dentary-squamosal joint in mammals, it falls short in presenting consistent and substantial results, requiring major revisions before it can be considered for publication.

Major Comments

1. Phylogenetic Analysis: I am surprised that the tree topology presented in Figures 1 and 4 is based on only one of the parsimonious trees. The optimization of dentary-squamosal articulation in Figure 4 cannot be validated, as the clade 'ictidosaur' collapsed into a polytomy along with the clade composed of tritylodontids, *Botucaraitherium/Brasilodon*, plus *Mammaliaformes* in the strict consensus tree or could differ from that in other parsimonious trees. I would recommend that the authors consider running a Bayesian analysis, as Bayesian analysis has been demonstrated to be more likely to recover a true phylogeny than Parsimony analysis (Wright and Hillis, 2014, 'Bayesian Analysis Using a Simple Likelihood Model Outperforms Parsimony for Estimation of Phylogeny from Discrete Morphological Data')

<http://journals.plos.org/plosone/article?id=10.1371/journal.pone.0109210>).

2. I appreciate that the authors provided the STL files of *Brasilodon quadrangularis*. The 3D model does show a gap between the squamosal and the lateral ridge of the dentary. However, the possibility of incipient contact between the dentary and squamosa in *Brasilodon* cannot be ruled out, considering the slight compression the skull has undergone in preservation. Additionally, the articular process extends posteriorly beyond the postdentary bones, similar to what is observed in other taxa like *Probainognathus*. It would be better to mention this aspect for a more comprehensive interpretation.

3. I suspect that many readers may feel confused by the usage of the term 'ictidosaur' for *Riograndia*. The clade 'ictidosaur' is paraphyletic in the strict consensus tree and several other phylogenetic analyses. I would suggest using 'basal tritheledontids' instead.

4. Because the dentary-squamosal contact evolved independently in the probainognathian cynodonts and *Mammaliaformes*, its use as the criterion for determining what is a mammal has a drawback. I would like to see a brief discussion included in the main text.

Minor Comments

L25: "probainognathian cynodont" instead of "non-mammaliaform probainognathian cynodont"

L30: Please reword this sentence. By the Late Triassic, a dentary-squamosal joint had already formed in at least two mammalian groups, including Haramiyavia.

Line 38: “ in the Late Triassic” instead of “ in the Early Jurassic”

Line 70: I would recommend using 'detached middle ear' instead of 'definitive mammalian middle ear' as the terminology does not accurately reflect homology.

Line 117: Delete “(AZ)”

Line 152: “ μ CT data” instead of “ 3D μ CT data”

Line 260: “ reveal” instead of “reveals”

Line 589: “ventral view of same” instead of “ventral view same”

Line 600: Delete “(1973)”

Extended Data Figure 2: “Brasilodon” in italic.

Appropriate use of statistic

This is not applicable here.

Reference

Barghusen, H. R., & Hopson, J. A. (1970). Dentary-squamosal joint and the origin of mammals. *Science*, 168(3931), 573-575.

Bonaparte, J. F., Martinelli, A. G., & Schultz, C. L. (2005). New information on Brasilodon and Brasilitherium (Cynodontia, Probainognathia) from the late Triassic of southern Brazil. *Revista brasileira de paleontologia*. Rio de Janeiro, RJ. Vol. 8, n. 1 (abr. 2005), p. 25-46.

Norton, L. A., Abdala, F., & Benoit, J. (2023). Craniodental anatomy in Permian–Jurassic Cynodontia and Mammaliaformes (Synapsida, Therapsida) as a gateway to defining mammalian soft tissue and behavioural traits. *Philosophical Transactions of the Royal Society B*, 378(1880), 20220084.

Luo, Z. X., Gatesy, S. M., Jenkins Jr, F. A., Amaral, W. W., & Shubin, N. H. (2015). Mandibular and dental characteristics of Late Triassic mammaliaform Haramiyavia and their ramifications for basal mammal evolution. *Proceedings of the National Academy of Sciences*, 112(51), E7101-E7109.

Referee #3 (Remarks to the Author):

Review by Zhe-Xi Luo (UChicago) Jan 2024

Overall Assessment

The novel cranial mandibular joint (CMJ) formed by dentary condyle and squamosal glenoid is a hall mark character of mammaliaforms. The development of this derived feature is a major evolutionary transformation in the rise of mammaliaforms from nonmammalian cynodonts, which has a long study history, and attracts a wide interest.

The new work by James Rawson, Agustin Martinelli and colleagues is very important in revealing significant new information on the precursor states to the dentary condyle in *Brasilodon* and *Ictidosauria* - two important sister clades to Mammaliaformes that are diagnosed by the dentary condyle.

I have much admiration for the Rawson et al. team for the impressive quality of original morphological data visualized from excellent CT scans of these important specimens. Furthermore and just as interesting, the newly revealed intermediate character states (“precursor character states”) have turned out to be homoplastic among the sister clades to mammaliaforms, indicative of iterative evolutionary experimentation in sister clades straddling the transition to the fully formed condyle-glenoid CMJ of mammaliaforms.

The apomorphic jaw hinge of dentary condyle and squamosal glenoid of mammaliaforms is capable functionally of bearing load from the force of jaw muscles that are integrated with complex teeth for mammalian chewing. The consensus among experts is that the dentary condyle of mammaliaforms is formed by a further enlargement and a flaring of the pre-existing dentary lateral ridge already present in cynodonts closest to mammaliaforms. The new study by Rawson et al. is the first to accurately visualize the 3D morphologies of these structures in rare fossils of *Brasilodon* and *Riograndia*.

These fossils discovered more than 20 years ago by Brazilian and Argentinian colleagues had been well-prepared in traditional fossil preparation. The features exposed on exterior of these fossils are informative as much as can be seen under microscope, and photographed and illustrated in several widely cited studies and in reviews. However, the traditional fossil preparation has its limitation and several aspects of the true 3D anatomy inside the fossil blocks are still unknown, until now. The new μ CT scans and visualization by Rawson et al are revelatory, and represent a major progress.

This paper has a wealth of new information about in the structure of craniomandibular joint in these sister clades of mammaliaforms - the Stl files submitted for review are abundant and easy for referees to explore and verify. The new insight from the authors on homoplastic evolution of precursor characters in closest clades to mammaliaforms are adding to a growing and new understanding of the evolutionary building-up of the fully developed dentary condyle. These are strong merits and I support this paper for publication in Nature.

However, two aspects of this paper still need significant improvement before the paper can be published: (1) supplementary data organization/presentation, and (2) several interrelated interpretations, for which anatomical vocabulary should be fine-tuned, and interpretation can be better contextualized.

Major Revision: Supplementary Information

First, I wish to point out that the STL files are superb and the videos from CT visualization look great – Rawson et al did a great job in segmentation work. Nonetheless, I would suggest some improvement, to make the paper even better. My issues with Supplementary Information are in organization and presentation, which should be improved and will be easy to revise:

Requested change 1 – The authors need to write an introductory text of Supplementary Information and provide a full list of supplementary files (STL files for models, animation video to represent the segmentation, and phylogenetic files Nexus on matrix and TNT files).

Currently there is not a master list. So the authors should organize such a list of the Stl files, the CT video, and they should also provide a caption for each STL file and each video, especially which composite reconstruction from which constituent fossil specimens. This is to make each STL and video self-standing with its own caption.

It appears that, when uploading these supplementary files in submission, the authors accidentally scrambled the sequence of upload. After downloading these files in a zip folder generated for referees by Nature's website, it took me hours to piece together the jigsaw puzzle about which file belongs to which fossil taxa. However inadvertently and unintentionally on the part of the authors, this poor organization of files is not good for reviewers. If a reviewer or a reader of Nature does not already know what these fossils are like, it would be difficult for the reviewers/readers to retrace the interpretation of these fossils by authors. This could be just a website glitch. But the onus is on the authors to fix it – and to write an intro text and to list all files for the supplementary information.

The long character list for phylogenetic analysis should also go into the supplementary information text.

The supplementary information must be better organized, before this paper can be published.

Request 2 – Somewhere in the main text of the paper, the authors must provide a statement as to how and where readers of Nature can access the Stl files and Video of CT rendering.

I cannot find such a statement or a URL in the main text of the paper, and methods section (in Nature standard format). I presume that all STL and video files will be uploaded in a zip folder as Supplementary Information on Nature's website. Or separately the authors can publish these files in Data Dryad (<https://datadryad.org>) with a published DOI. The data submitted by authors for review are impressive. But the authors must also explain how data will be publicly available.

This must be fixed before this paper can be accepted by Nature.

Request 3 – RE: Brasilodon UFRGS-PV-1043-T – this specimen needs a video, and more illustrations in figures. The flaring posterior end of lateral ridge is preserved in proximity to the squamosal zygoma in this fossil – so it is important for authors to provide better 2D graphics and video of this specimen.

Currently this specimen is shown in a small crop-out of local area in Fig 2e (ventral view) and shown again extended data Figure 2 in medial view and posterior view. In revision, I like to ask authors to include a lateral view of this mandible in Extended Data Figure 2.

The main composite reconstruction of the mandible (especially its posterior end), as presented in the first STL file of Brasilodon, is likely to be based on the specimen of PV1043T. But I have to guess about this because there is not a caption or other explanatory text for each of the STL file. If authors can add a “image caption” to each STL file in a list, this will make it easier for readers to follow. More specifically for reconstruction of the posterior end of the dentary (most relevant to CMJ), such information can facilitate for the interested reader to verify how the authors have accomplished the composite reconstruction from well-preserved parts of individual specimens. This is just a matter of improving the communication.

Request 4 – Video for PV1030T. PV-1030-T is important because it is the only specimen in which the squamosal zygoma is still attached to the petrosal, and the posterior end of the lateral ridge of the dentary in close proximity of squamosal zygoma. Although not perfect, the right squamosal and right dentary of PV1030T are the best available for interpreting the original anatomy. On the left mandible, the postdentary elements and the quadrate are still intact.

In revision, I like to request for the authors to provide a new video in full (360 degree) roll-rotation plus a full (360 degree) yaw-rotation of the skull, with segmented basicranial and CMJ parts in color of the skull. This information would be crucial to support authors’ interpretation that there is an unfilled gap between the dentary lateral ridge flaring (precursor to condyle), and the zygomatic part of the squamosal – a key anatomical observation of this work. The small cropped local anatomy in Fig. 2e is not enough. A video of the colored elements (from segmentation) would help to better demonstrate this interpretation by the authors.

Request 5 – Supplementary Video S2 for Brasilodon PV628T. Currently this video only shows the PV628T in yaw (around Z-axis) rotation. In revision, I like to see the authors further add a “roll” rotation the specimen around X-axis (along length of rostrum - occipital condyle). Presenting the segmented skull in both yaw and roll rotations (two rotations orthogonal to each other) will allow readers to fully see the anatomy intuitively.

Request 6 – Video S3 for Riograndia PV596T. The Yaw (Z-axis) rotation is all good. But the further roll (X-axis along length of cranium) rotation was cut short to 270 degrees. Can the authors extend the roll-rotation more to a full rotation of 360.

Request 7 – Video S4 for Riograndia PV833T. The yaw (Z-axis) rotation is cut short. Can you further the yaw-rotation to a full 360 degree. Also, I suggest that the authors add another 360-degree roll-rotation (around the X-axis through the length of the skull), to facilitate viewers to gain an intuitive visual understanding of the morphology.

Major Suggestion 1: Not all dentary-squamosal contacts are the same. I suggest that Rawson and colleagues adopt more distinct descriptors of the character-states of simple physical contact of Dentary-Squamosal (pre-mammaliaform cynodonts) and differentiate this from the derived pattern of a synovial joint of dentary condyle and squamosal glenoid that are load-bearing (mammaliaforms). This helps to convey the differences of ictidosaur-tritheledontids and Brasilodon on the one hand and the Mammaliaformes on the other.

In the descriptive text, I would suggest to call the dentary/squamosal contact for Ictidosaur (Riograndia+Tritheledontids) (= D-SQ 'articulation' via lateral reinforcement). In the meanwhile, call the dentary condyle and squamosal glenoid joint for mammaliaforms. The D/SQ contact and the joint of D-condyle and SQ glenoid are morphological different, although we assume the former is a precursor condition to the latter, in the continuum of evolutionary build-up of the true mammalian dentary condyle with squamosal glenoid.

There is a functional nuance - the former term "dentary-squamosal contact" would be neutral if the contact had (or lacked) load-bearing from chewing muscle force. The latter with a dentary condyle is clearly for load-bearing and a squamosal glenoid for a synovial joint a wide range of joint movement.

The important input of the new work is on the precursor states of dentary/squamosal contact (by lateral reinforcement. This is where the analytic originality of Rawson et al.'s work is.

Starting with Crompton (1958), most early works on the cranio-mandibular joint mentioned "dentary-squamosal articulation" in a loose manner. But we now know much better the morphologies of CMJ. Now it is clear there is a range of configurations between Brasilodontids and ictidosaur, versus Mammaliaformes. If you need example how these descriptors are verbally differentiated – check out the related text in Luo et al. (2002) discussion and character list.

Major Request for revision #2 – I urge the authors revise their middle ear reconstruction of Probainognathus in Figs 1 and 4 – Dr. Edgar F. Allin made a landmark contribution with Allin (1975) and this work really changed the opinion on synapsid-therapsid mandibular middle ear. However, one aspect of Allin's reconstruction is not accurate (Allin 1975; 1986). Dr. Allin over-estimated the size of reflected lamina of cynodonts.

All of Allin's reconstructions of "a long hook-like" reflected lamina of ectotympanic for a range of cynodonts are over-interpretation. The reflected lamina in Allin's rendering is much too long (too mammal-like) than the short projection actually preserved in the fossils of Probainognathus (Luo's personal observation of several specimens of this taxon). For example, Hopson-Kitching's (2001: fig. 5) reconstruction of ectotympanic reflected lamina Probainognathus (as shown in diagram below) is closer to the original fossil and more accurate.

Hopson & Kitching (2001) fig 5 "ref lam" is the reflected lamina of the ectotympanic – it is short.

Major Request for revision 3 – Several specimens of *Pachygenelus* show a thickened flange on the posterior end of the lateral ridge of the dentary (I saw these under microscope).

This structure in specimens of *P. monus* is identical to that of *Diarthrognathus*, and to the lateral ridge flaring of *Riograndia* according visualization of Rawson and co-authors here. In other words, the lateral ridge is thickened and rounded in *Pachygenelus* in the same way (see Crompton and Luo 1993: fig 4.14) as this structure in *Diarthrognathus* (see Crompton and Parkyn 1963).

[REDACTED]

Regarding Figure 3, I am convinced that the entire ictidosaur clade should be interpreted as possessing the “Dentary-Squamosal via lateral reinforcement” (green diamonds on green lines). *Pachygenelus* should not be in dashed line.

I suspected that Rawson and colleagues relied on Gow’s (1981) abstract (ref #50). But they appear to be unaware that Gow (1981) had turned out to be an erroneous mis-interpretation - I can further explain below.

Gow (1981) is a short abstract – here he challenged Crompton’s (1958, 1963) interpretation of the double joint of the tritheledontid *Diarthrognathus*. Gow’s 1981 abstract made two points. First, Gow pointed out that Crompton mis-identified a part of squamosal as the quadrate for the specimen of *Diarthrognathus*. Gow was correct on this, and I fully endorse it. In response, Crompton (1972: Ref #2) corrected this error.

Second, Gow (1981) questioned Crompton’s interpretation of the D/SQ contact in *Diarthrognathus*, on the basis of Gow’s review of specimens (of *Pachygenelus*) available to him. Gow (1981) briefly stated that “However, once the quadrate is correctly positioned such a glenoid configuration is no longer tenable...” However, here as a reviewer of Rawson et al manuscript, I like to point out to Rawson and colleagues that Gow was wrong on the latter point.

Gow did not publish the actual information (photos or illustrate graphics) to back up his two sentences of Gow 1981 abstract. But I got to see the good photos of “Gow’s *Pachygenelus*” materials through some interesting circumstances.

I met Dr. Chris Gow in 1993 and we had an extensive conversation on this topic of our common interest. Thereafter we exchanged reprints in 1994, and I sent him the reprints of Crompton & Luo (1993), Luo (1994) and Luo & Crompton (1994), all of which touched on the interpretation of the dentary-squamosal contact in tritheledontids.

In response to these papers, and also likely a follow up with our discussion (1993), in 1997 Gow submitted a manuscript to *Palaeontologia Africana* to describe the full details with excellent stereophotographs of “his *Pachygenelus* material.” In this manuscript, Gow explained that, in his study specimen, the posterior end of the dentary would not have a contact with the squamosal because the quadrate would be positioned on top of the dentary lateral ridge and in between the dentary’s end and the squamosal. This is the key reason why Gow maintained that because there was not a D/SQ contact in his *Pachygenelus* fossil. In the discussion of this manuscript, Gow raised a question again (as in his 1981 abstract) about Crompton’s interpretation of a double joint in *Diarthrognathus*. Gow suggested to *Palaeontologia Africana* that the manuscript be given for me to review.

But after I got to see the purported key evidence in good stereophotographs from Gow, I disagreed with Gow on his interpretation. In my review, I pointed out that in Gow’s specimen, the quadrate/articular and the postdentary rod were displaced, post mortem, away from the postdentary trough. This dislocation of “middle ear elements” is self-evident from the stereo photos of this manuscript. Simply put, his initial interpretation in the Gow (1981) abstract was based on a post mortem distortion of the fossil in which the quadrate was displaced by some distance from its original position.

Because Gow’s (1981) abstract had just two sentences on this anatomy but no figure or photo, nobody else had seen “his fossils” (or photographs and illustrated graphics). With Gow’s 1997 manuscript to *Palaeontologia Africana* to describe this fossil material with good stereo photos for documentation, it was the first time when Gow showed his fossil to the peer-review community. Thus likely I was the first person to assess his evidence, and I had a chance as a referee to point this interpretative error to him. Likely this is also the first time Gow received an alternative opinion about what his fossils actually show with regards to the jaw hinge.

Although I criticized Gow’s manuscript in my review, I was anticipating that he would revise his manuscript and return it to *Palaeontologia Africana*, or he would rebuttal my review. But Chris Gow never followed up, and later he withdrew this manuscript from *Palaeontologia Africana* (the editors of *Palaeontologia Africana* told me about Gow’s withdrawal of his *Pachygenelus* paper). In late 1990s Gow also retired and moved away from his university.

But unfortunately, because the material under Chris Gow’s study never got into print publication through peer-review, the Gow (1981) abstract (and errors therein) would become the only information from him about his *Pachygenelus* materials. The erroneous interpretation still got cited by the un-suspecting colleagues in this case.

But this is not right.

In Figure 4 caption, Rawson and colleagues are saying “...Dotted line leading to *Pachygenelus* denotes uncertainty as to presence/absence of a distinct condyle”. But this is really not right. Here I offer an illustration by Crompton and Luo (1993: fig 4.14 attached in this review), *Pachygenelus* has a thickened and flaring posterior end of lateral ridge (precursor condition to the dentary condyle). Also see Luo and Crompton 1994 JVP fig. 11B - the squamosal zygoma of *Pachygenelus* has a concavity that would receive the dentary, on the basis of *Pachygenelus* specimens (SAM-K139).

If anything, Rawson and colleagues would be able to streamline the narrative of ictidosaur-tritheledontids with regards to the dentary-squamosal contact: *Riograndia-Pachygenelus*-

Diarthrognathus-Tritheledon are more or less the same - they all had some kind of D/SQ contact, despite the two-sentence statement by Gow (1981).

Suggestion for change #4 - Lines 62-63; Lines 366-367: Re: nocturnality. Maybe the authors should delete the mention of nocturnality. Rawson and colleagues mentioned nocturnality in addition to the elevated metabolism and endotherm for mammals. I understand they would need some ad hoc statement of important evolutionary adaptations of mammaliaforms.

In my view, elevated metabolism and endothermy, novel feeding adaptation by chewing (functionally related to CMJ), and miniaturization are certainly important adaptations. However, why nocturnality? With this casual mentioning of nocturnality, Rawson and co-authors would open the manuscript up for potential dispute. Angielczyk and Schmitz (2014) proposed that nocturnality arose convergently in nonmammaliaform synapsids: possibly in early synapsids Sphenacodon and Dimetrodon, and certainly in several therapsids. Angielczyk and Schmitz (2014) also suggested that Tritylodon would be nocturnal, as an example from cynodonts. Thus nocturnality may not be a unique mammaliaform evolutionary innovation.

Delete nocturnality, and stick to (1) endothermy, (2) novel feeding adaptation by chewing, and (3) miniaturization. This helps to streamline the narrative.

Line by Line Queries and Suggestion of Corrections

Lines 37-38 Re: “..., mammaliaforms attaining a fully independent dentary-squamosal contact with a conspicuous dentary condyle **later, in the Early Jurassic.**” This loose statement is not accurate because Haramiyavia (Late Triassic at 208 ma) is a mammaliaform, and it clearly has a dentary condyle (see Luo et al. 2015 reconstruction of Haramiyavia). If the authors prefer to keep this narrative, perhaps it can be modified to:

“..., mammaliaforms attaining a fully independent dentary squamosal glenoid joint with a conspicuous dentary condyle, as exemplified by phylogenetically more derived Haramiyavia in Late Triassic, and Sinoconodon and Morganucodon the Early Jurassic.”

Lines 43-44 “The quadrate and the articular, which form the jaw joint of most tetrapods, were greatly reduced and separated...”. This loose statement is a bit imprecise. The quadrate-articular jaw joint is a feature of all gnathostome vertebrates. The phrase “most tetrapods” can be misconstrued. You either say “gnathostome vertebrates” (which would be politically correct for hardcore cladists) or you say “non-mammalian cynodonts” (contextually relevant for this study). Change to “The quadrate and the articular, which form the jaw joint of gnathostome vertebrate, were greatly reduced and separated...”.

Line 48: Re: “there remains a question when, and how often, this jaw joint first evolved”. This sentence can be misconstrued, and should be modified. I would suggest a change to “... there remain a question where along cynodont-mammal phylogeny the precursor conditions evolved in the build-up of this load-bearing jaw joint.” Or something to this effect.

Lines 60: Change from “These results demonstrate that significant experimentation...” to “These results demonstrate that significant **evolutionary** experimentation...”

Lines 67-68: Re: “a unique middle ear containing three ossicles (malleus, incus, stapes) and a dentary-squamosal jaw articulation;” I strongly urge the authors to distinguish the two different character-states: contact of dentary and squamosal vs. joint of dentary condyle and squamosal glenoid. How about modify this sentence to “a unique middle ear containing three ossicles (malleus, incus, stapes) and a load-bearing jaw hinge of dentary condyle and squamosal glenoid;”

Line 74: change from “as in other tetrapods” to “as in other gnathostome vertebrates”

Lines 121-123. Re: “...Previous descriptions have indicated that Jurassic tritheledontids (or ictiosaurs) may possess a type of dentary-squamosal articulation 2-4,6,42, but its similarity to that of mammaliaforms such as Morganucodon is unclear. “

This sentence is not an accurate characterization of our current state of knowledge. To be fair and also to be accurate, I urge authors to consider a revised text – for example: “Previous descriptions have indicated that Jurassic tritheledontids (or ictiosaurs) may possess a dentary-squamosal contact 2-4,6,42, although this contact in tritheledontids lack the robust dentary condyle as seen in mammaliaforms such as Morganucodon.” OK?

Line 126. “They” is not an accurate word here. I suggest that this sentence be modified “but the anatomical information of these new fossils have yet to be integrated into the wider analyses of cynodont morphological evolution alongside more well-known species.”

Lines 133-135. “Hitherto unknown morphology visible in our dataset overturns previous descriptions of jaw joint and middle ear anatomy and demonstrates the extent of homoplastic evolution in the precursors of mammals.” This sentence borders on an overstatement. To be fair and be accurate, I would suggest a revision to “Hitherto unknown morphology newly revealed by our CT scans and incorporated into our phylogenetic dataset demonstrate a much greater extent of homoplastic evolution of the jaw hinge and some middle ear bones in the precursors of mammals.”

Lines 372-374. “The relatively larger postdentaries of Riograndia alongside a secondary dentary-squamosal articulation represents a different line of experimentation that occurred earlier than the adoption of a secondary jaw joint in mammaliaforms;...”.

I like this discussion. But I would suggest a tweak of this sentence, by changing it to: “The relatively larger postdentaries of Riograndia alongside a secondary dentary-squamosal **contact would** represent a different **evolutionary** experimentation that occurred earlier than the **acquisition** of a secondary jaw joint **for load bearing** in mammaliaforms;...”.

Miscellaneous corrections on references:

Ref 2. Crompton (1972) is a single-author paper in an edited volume. You mixed up the name of a single author with the names of the volume editors.

Ref 25. Mussett should not be “MUSSETT”

Ref 30. Oliveira should not be “OLIVEIRA”

Ref 51. Two errors in this cited reference. It should be "Hopson, J. A. and Barghusen, H. R. An analysis of therapsid relationships. in *The Ecology and Biology of Mammal-like Reptiles*, (eds. Hotton, N. MacLean, P. D., Roth, J. J., and E. C. Roth) 83-106 (Smithsonian Institution Press, 1986).

Not "Hopson et al." and not "(Ed Hotton N.)"

Ref 67. (eds. Hotton, N. MacLean, P. D., Roth, J. J., and E. C. Roth). Not "(ed Hotton, N.)"

Author Rebuttals to Initial Comments:

We thank the reviewers for their detailed review of our manuscript. In brief, in this revised version we have made adjustments to the manuscript, performed additional phylogenetic analyses and included further supplementary data. Our full responses to reviewer comments are outlined below.

Referee #1:

Comments on “South American stem mammals reveal homoplastic evolution of the oldest mammalian jaw joint” by Rawson et al. (2023-12-22267).

The report used micro-CT scanning to examine the 3D jaw joint anatomy of three non-mammaliaform cynodonts: *Brasilodon quadrangularis*, *Riograndia guaibensis*, and *Oligokyphus major*. This is another study with a focus on joint morphology and changes, particularly the formation of the secondary dentary-squamosal joint, that took place across the cynodont-mammaliaform transition; the dentary-squamosal joint has been commonly considered a key mammalian feature.

The main findings of the study include two aspects: 1. *Brasilodon quadrangularis*, often placed as a sister-taxon of Mammaliaformes, does not have the dentary-squamosal joint, contrary to the original report in which such a joint was claimed to be present. 2. A distinct dentary-squamosal articulation was found to be present in *Riograndia*, which is a derived probainognathian but older than some related taxa, such as *Pachygenelus* and *Diarthrognathus* that were known to have a dentary-squamosal joint and the dentary condyle, albeit small, is developed, particularly so in *Diarthrognathus*.

The authors concluded that at least two cynodont lineages were evolving a more ‘mammal-like’ jaw articulation, one is the clade that contains *Riograndia*, *Pachygenelus*, and *Diarthrognathus*; the other is Mammaliaformes. The dentary-squamosal joint in *Riograndia* is currently the oldest confirmed one in cynodonts. The authors thought the development of the dentary-squamosal jaw joint may be attributable to the miniaturization of the cynodont body sizes.

We agree with the authors’ general view that multiple experimentations of jaw joints in derived probainognathians had happened and that these are partly driven by or associated with the miniaturization of the cynodont body sizes.

We thank the review(s) for this detailed and informed review of our manuscript. We provide our response to their comments as follows.

Comments on the evidence

We think the evidence that shows the absence of the dentary-squamosal joint in *Brasilodon quadrangularis* is convincing, based on the shape of the articular process of the dentary and the shape of the squamosal, as shown in Fig. 2.

The evidence for the presence of the dentary-squamosal joint in *Riograndia* is less convincing. There is “a clear indentation on the ventromedial surface” of the squamosal. The dentary, however, does not seem to show a clear articular area, likely convex, that should

contact the indentation (or facet) in the squamosal. The right mandible is quite complete as shown in Fig. 3a and 3b (UFRGS-PV-596-T), but its distal end of the articular process of the dentary seems incomplete; the left mandible (UFRGS-PV-833-T) has its articular process of the dentary even less complete. Carefully reviewing the Extended Data Fig. 3, the stl, and mpg files of UFRGS-PV-596-T (many thanks for providing these great images!), our concern remains, that is, the articular process of the dentary does not show a convincing articular area that should be in contact with the “indentation” of the squamosal to form a joint, or this area was damaged in the studied specimens. We fully understand, however, that it is difficult to have “perfect” fossils that could unequivocally show the jaw joint in the fossil records; we regard the evidence provided by the authors as reasonable, but partial, and we will continue commenting on the rest of the manuscript.

In the revised manuscript we have now provided more detail on the articular process of the dentary to strengthen our argument. In addition to the *Riograndia* specimen UFRGS-PV-596-T, the articular process is best-preserved in UFRGS-PV-1319-T and MPDC-1B1, which do show the extension of the lateral ridge that contacts the squamosal. Though the condition is not as derived as that of the Jurassic tritheledontids and does not possess the thickened and upturned ridge seen in those taxa, the lateral ridge of the articular process is widened and, based on UFRGS-PV-596-T, clearly matches the shape of the adjacent indentation on the squamosal. Alongside the unusually widened and extended articular process, we consider the position of the convex articular process next to the indentation on the squamosal of UFRGS-PV-596-T to be strong evidence that a dentary-squamosal contact was present in *Riograndia*. We have now added posterior and lateral views of the articular process of MPDC-1B1 in Figure 3 to better show this morphology and added a comparable line drawing of the articular process of *Pachygenelus* from Crompton and Luo (1993) to highlight the more derived condition in Jurassic tritheledontids. We have also included STL files and rotating animations of UFRGS-PV-1319-T and MPDC-1B1 as Supplementary Information to provide readers with additional evidence to support our conclusions. This data is now available at the University of Bristol data repository, data.bris, at <https://doi.org/10.5523/bris.2ie8neiry701e23iayx9gdsytv>.

Comments on interpretations/conclusion.

1. The authors have shown an inconsistency in using the terminology on the jaw articulation in question. For instance, in the abstract, it is “The classic mammalian dentary-squamosal jaw hinge therefore evolved more than once and is more evolutionary labile than previously considered.”. In other places, “we show that this ‘mammalian’ jaw joint evolved independently from that of mammal precursors,” and “a more ‘mammal-like’ jaw articulation.”

The authors could use the dentary-squamosal joint (articulation, hinge) without associating it with mammals; otherwise, the implication that the typical mammalian jaw joint is found in the non-mammaliaform cynodonts would create a conflict meaning or confusion, particularly for those who are not familiar with the structure and phylogenetic context. It’s probably better to say something like: the dentary-squamosal jaw joint, traditionally considered a typical mammalian feature, evolved more than once.

We thank the reviewer for spotting this and have edited the text to address their concern and ensure our terminology correctly delineates between the independent evolutions of the dentary-squamosal jaw joint. We have changed the examples that the reviewer mentions above to “The dentary-squamosal jaw contact, traditionally considered a typical mammalian feature” (L39), “this jaw contact evolved independently from that of mammal precursors”

(L57), and “were evolving features of the jaw articulation that are typically associated with mammals” (L388), respectively.

2. The main conclusion of the work is that the dentary-squamosal jaw joint is not unique to mammaliaforms but probably evolved independently more than once. Although the conclusion is probably supported by evidence, it is not a new idea. This parallel evolution of the dentary-squamosal joint was proposed at least since Crompton and Parkyn (1963). Crompton and Parkyn (1963: 743, 707): stated: “It was shown that in *Diarthrognathus* and *Morganucodon* the dentary condyle develops at the posterior termination of this lateral ridge.” “Although the dentition and the proportions of the lower jaws in *Diarthrognathus* and *Morganucodon* are different, the structure of the articular processes of the dentary in both forms is strikingly similar. The structure of the postcanine teeth indicates that the two forms are not closely related so that this similarity was probably due to parallel evolution.” We would view that the jaw joint of *Riograndia* provides additional evidence supporting the view that the dentary-squamosal joint evolved more than once in cynodonts; in particular, it shows an earlier and probably more primitive condition than that in *Diarthrognathus*.

The work of Crompton and Parkyn lacked any formal phylogenetic analysis that tested the idea of convergence of the dentary-squamosal contact in a cladistic framework. Also, the idea of convergence has not gone uncontested since the work of Crompton and Parkyn, given that several phylogenetic analyses have recovered tritheledontids as the sister clade to mammaliaforms which has cast doubt on the idea of homoplastic evolution. Hence the question is still of central importance to mammalian origins and synapsid evolution. Our work tests past assertions of homoplasy versus convergence using new fossil evidence and our present understanding of cynodont evolution and computational phylogenetics. Unique data from *Brasilodon* and *Riograndia* allows us to describe this convergence and presents an abundance of anatomical data that has been absent in previous discussions of this topic. In particular, the publication of 3D data for the ictidosaur jaw joint is a major improvement, since relatively little actual data on *Pachygenelus* and *Diarthrognathus* has been published thus far. Our data and methods allow us to make our conclusions with a significant body of evidence that was lacking previously.

3. The statement (sentence) at the end of the next last paragraph (before the Conclusion): “one that appears to prioritise jaw joint robustness even at small body sizes.” appears to be problematic at least to us. The term “prioritise” may not be appropriate; this is not an intelligent design; it is by natural selection. More importantly, why did these animals prioritise jaw joint robustness even at small body sizes? It seems that you concluded something that you didn’t discuss in the article.

We understand the reviewer’s concern with the wording we have used here and agree that it should be changed to avoid any confusion. The revised version (L377) reads “one that appears to be characterised by greater jaw joint robustness, possibly to cope with a more mechanically demanding diet, even at small body sizes”. The use of the word “characterised” still draws the correlation between these forms and more robust combinations of primary and secondary jaw joints but lacks the implication of deliberate change that “prioritise” does. Regarding the reason for this adaptation, this sentence was a call back to our discussion of possible herbivory in *Riograndia* (L270), and the effect this may have had on jaw joint morphology. We have added the clause “possibly to cope with a more mechanically demanding diet” to the sentence flagged by the reviewer to clarify this link.

4. The current text could be considerably condensed. In general, we think the main text elaborates a bit too much on what has been known. It's necessary to provide the context and background for the subject discussed in the manuscript, but there is no need to re-state much of what has been said or keep it at a minimum. Focus on what is the new evidence you have and what the new evidence tells us.

We have moved the section on *Oligokyphus* to the Supplementary Information to streamline the main text, given that our investigations into this taxon generally corroborated previous descriptions. We have also cut down the introduction section (e.g. L76)

5. Figures 1 and 4 can be combined; much of the information and jaw figures are redundant. Fig. 4 displays the mandible shapes in a simplified phylogenetic frame and emphasizes the two independent occurrences of the dentary-squamosal joint. The same information can be displayed in Fig. 1 by marking the corresponding branches in green, orange, and blue, as in Fig. 4. The jaw figure bearing terminology (*Thrinaxodon*) at the lower left corner is redundant, which can be deleted (using the upper right one to show the terms, if that is necessary). Place the lower jaws of *Riograndia* (above) and *Brasilodon* (below) in the lower left area above the geological scale. This figure could be placed at the end to show the conclusion (morphology, age, independent evolutions, etc.).

We thank the reviewer for their suggestion and we agree there is some redundancy between the two figures. We debated whether to keep or remove one of the figures but decided that both figures are useful for framing the overall narrative of the paper, especially for those less familiar with the topic as may be the case in a broad impact journal such as *Nature*. Our rationale is that Figure 1 serves to introduce the taxa that we are discussing, their relative ages, and the anatomy of the jaws/postdentary bones. Figure 4 serves to summarise our findings and highlight the evolution of the dentary-squamosal joint in a phylogenetic context using the new data presented in the paper. If the figures were combined and moved to the end of the manuscript, we feel that readers, especially those who do not specialise in cynodont/mammal evolution, would lack key context about these taxa and where they sit relative to mammals, as well as a clear diagrammatic introduction to the postdentary bones/middle ear ossicles. Likewise, if the figures were combined at the beginning of the text, readers would be drawn to the highlighted components about dentary-squamosal joint evolution before these new findings have been introduced by the text. Therefore, we argue that having Figure 1 as an introduction to cynodont taxa and their anatomy and Figure 4 as a simplified summary of our findings and what they show is the best way to complement the text and maximise detail and readability.

Minor comments:

1. In the abstract, "have much to offer on this debate." Which is the debate? You mentioned a "question" but not a debate. Do you mean the question you mentioned above?

Changed wording to "much to offer to this discussion" (L23).

2. In "...predating that seen in Mammaliaformes by approximately 20 million years 7,8", Double check ref 7 is the right one to cite here.

This reference to Whiteside et al., (2016) documents the age of the fissure faunas of Bristol and South Wales, and thus provides an age for the mammaliaform *Morganucodon* that is here

directly compared to the age of *Riograndia*. In light of other reviewer comments, we have added a citation for Clemmensen et al., (2016) that provides an age for the Rhaetian mammaliaform *Haramiyavia* that also possesses a dentary-squamosal joint (L56).

3. The subtitle “Main” appears awkward – could this be Introduction? This section can be greatly condensed.

Changed to “Introduction”

4. In “the origin of these features is a highly significant event in mammalian evolution 4,5,18,19.” It’s probably better to put Allin 1975 before A&H 1992 in the citation sequence - a trivial thing, but a historical fact.

These references have now been switched around so that the work of Allin (1975) appears first.

5. In “The definitive mammalian middle ear evolved within crown mammals during the Mesozoic,” It’s probably safer to say “mammaliaformes” than “crown mammals” because of the unstable placement of allotherians within the mammaliaforms. In some analyses, allotherians, particularly haramiyidans, that have the DMME were placed outside of the crown mammals.

This is a good point, changed to “Mammaliaformes” (L72).

6. In “serve as the point of articulation between the skull and lower jaw.” Better replace skull with cranium.

This sentence has been cut in the revised version

7. The section “Jaw articulation in *Oligokyphus major*” may not be necessary to include in the main text; it provides very little if any, new information.

We have moved the section on *Oligokyphus* to the supplementary text (L910) to better streamline the main text of the manuscript. We have added a sentence to the methods (L700) section informing readers that more information is available in the supplementary information.

8. In ...”yet it does appear that the later forms were approaching a condition closer to that seen in *Morganucodon*.” It is probably better “the later forms of tritheledontids” to be clear.

This sentence has been edited in the revised version (L241).

9. In discussing the reorganisation of muscles: “This raises the question of why derived probainognathians show..., allowing reduction of the postdentary bones and promoting the evolution of secondary articulations 5,27,67.” It’s better to cite Crompton & Parkyn, 1963 because they already explored this issue in describing the lower jaw of *Diarthrognathus*.

Reference added.

10. Put the year (1963) for Crompton & Parkyn (ref. 42) in the reference.

Corrected

11. In the figure 1 legend, does the sentence “Topology is based on 31 and the phylogenetic analysis within this study” mean the topology is based on two phylogenetic results? How this was done should be explained in Phylogenetic analysis in Methods.

In the initial submission, the interrelationships of ictidosaurs were taken from a previous publication where they were better resolved (i.e., Martinelli *et al.*, 2016). We have chosen to amend this in the current submission to remove confusion and show the relationships of probainognathians recovered from our new Parsimony analysis with an updated character matrix. This new analysis recovers ictidosaurs as a monophyletic group in the strict consensus tree with the interrelationships within the clade collapsed into a polytomy, with the exception of a sister relationship between *Pachygenelus* and *Diarthrognathus*. Additionally, we carried out a Bayesian analysis (see text for methods and results) that also recovered Ictidosauria as a monophyletic group. These additional analyses further support our conclusions regarding the evolution of the dentary-squamosal contact.

12. It would be better if some measurements were provided (jaw length, postdentary unit size, if possible) so that the size and relative size could be better appreciated. These numbers may also be useful to echo the discussion on miniaturisation.

We have added a column of “Specimen length (mm)” to Extended Data Table 1 to provide this information.

Referee #2:

Thank you for the opportunity to review the manuscript 2023-12-22267 by Rawson *et al.*, which I found of great interest. In this manuscript, the authors utilized micro-CT scanning to provide further information on the jaw joint morphology for three probainognathian cynodont taxa: *Brasilodon quadrangularis*, *Riograndia guaibensis*, and *Oligokyphus major*. The original description of *Brasilodon quadrangularis* suggested incipient contact between the dentary and squamosal. Here, they present a new interpretation that *Brasilodon* lacks an incipient dentary condyle and squamosal glenoid, and the jaws articulate solely via the quadrate-articular joint, more similar to that of tritylodontids. Based on the new information, they concluded that there is homoplastic evolution of the dentary-squamosal jaw joint within probainognathian cynodonts.

While this manuscript provides new insights into the origin of the dentary-squamosal joint in mammals, it falls short in presenting consistent and substantial results, requiring major revisions before it can be considered for publication.

We thank the reviewer for their suggestions on how improve our manuscript. We have addressed their comments and provide responses as follows.

Major Comments

1. Phylogenetic Analysis: I am surprised that the tree topology presented in Figures 1 and 4 is based on only one of the parsimonious trees. The optimization of dentary-squamosal articulation in Figure 4 cannot be validated, as the clade ‘ictidosaur’ collapsed into a

polytomy along with the clade composed of tritylodontids, *Botucaraitherium*/*Brasilodon*, plus Mammaliaformes in the strict consensus tree or could differ from that in other parsimonious trees. I would recommend that the authors consider running a Bayesian analysis, as Bayesian analysis has been demonstrated to be more likely to recover a true phylogeny than Parsimony analysis (Wright and Hillis, 2014, 'Bayesian Analysis Using a Simple Likelihood Model Outperforms Parsimony for Estimation of Phylogeny from Discrete Morphological Data' <http://journals.plos.org/plosone/article?id=10.1371/journal.pone.0109210>).

1. Phylogenetic analysis. In this revised version, we have updated our morphological character matrix to reflect newly published data on probainognathian anatomy and carried out additional phylogenetic analyses. We document how these further analyses make our conclusions regarding homoplasy of the dentary-squamosal joint more robust. Amendments to the character matrix based on the newly published work of Kerber *et al.*, (2023) and Martinelli *et al.*, (2024) have resulted in Ictidosauria being recovered as a monophyletic clade by our Parsimony analysis, wherein they branch more basally than the clade composed of tritylodontids, *Brasilodon*/*Botucaraitherium*, and Mammaliaformes. The optimisation of the dentary-squamosal articulation in figure 4 can therefore be validated with this new topology. Additionally, we have now carried out a Bayesian analysis in MrBayes to provide additional phylogenetic evidence and added the methods (L715) and results (L308, L1303) to the main manuscript and the Supplementary Information. We ran four MCMC's for one million generations with sampling every 100 generations and a burn-in of 25%. We once again recovered Ictidosauria as a monophyletic group that (in addition to *Therioherpeton*) branches more basally than tritylodontids, *Brasilodon*/*Botucaraitherium* and Mammaliaformes, thereby supporting our conclusions.

Kerber, L., Preto, F. A., & Müller, R. T. (2023). New information on the mandibular anatomy of *Agudotherium gassenae*, a Late Triassic non-mammaliaform probainognathian from Brazil. *The Anatomical Record*, 1–9. <https://doi.org/10.1002/ar.25317>

Martinelli, A. G., Ezcurra, M. D., Fiorelli, L. E., Escobar, J., Hechenleitner, E. M., von Baczko, M. B., Taborda, J. R. A., & Desojo, J. B. (2024). A new early-diverging probainognathian cynodont and a revision of the occurrence of cf. *Aleodon* from the Chañares Formation, northwestern Argentina: New clues on the faunistic composition of the latest Middle–?earliest Late Triassic *Tarjadia* Assemblage Zone. *The Anatomical Record*, 1–33. <https://doi.org/10.1002/ar.25388>

2. I appreciate that the authors provided the STL files of *Brasilodon quadrangularis*. The 3D model does show a gap between the squamosal and the lateral ridge of the dentary. However, the possibility of incipient contact between the dentary and squamosal in *Brasilodon* cannot be ruled out, considering the slight compression the skull has undergone in preservation. Additionally, the articular process extends posteriorly beyond the postdentary bones, similar to what is observed in other taxa like *Probainognathus*. It would be better to mention this aspect for a more comprehensive interpretation.

2. *Brasilodon*. The left side of UFRGS-PV-1030-T and the right side of UFRGS-PV-1043-T have suffered little postmortem distortion and illustrate the best preservation of the jaw joint region of the skull. Digital manipulation of the jaws of UFRGS-PV-1043-T to attempt to move the mandible into articulation with the skull resulted in a similar anatomy to that of UFRGS-PV-1030-T, where the articular process cannot be made to contact the squamosal either posteriorly or laterally. While not perfectly preserved, we believe the anatomy of these

minimally distorted specimens, and evidence from other specimens, presents a robust case for separation between the dentary and squamosal in *Brasilodon*. We have now highlighted specific specimens more frequently in the descriptive text (e.g. L154, L160), and the skull models of UFRGS-PV-1043-T and UFRGS-PV-1030-T are now included in the Supplementary data as both STL files and rotating animations to further illustrate the anatomical configuration. This data is now available at the University of Bristol data repository, data.bris, at <https://doi.org/10.5523/bris.2ie8neiry701e23iayx9gdsytv>.

We have also made some amendments to the descriptive section on *Brasilodon* to highlight the posterior extent of the articular process (L186, L191). Though posterior extension of the articular process was a vital step in the evolution of a dentary-squamosal connection, we would argue that the posterior extension of the articular process (that does not yet reach the squamosal) does not necessarily mean this contact was present. Tritylodontids also possess a long articular process that extends posteriorly beyond the postdentary bones, and yet they unambiguously do not possess a dentary-squamosal contact. We now more clearly suggest in the text (L191) that the extension of the articular process would have braced the postdentary bones from above during feeding, thus leading to the evolution of a precursor condition to a dentary-squamosal contact.

3. I suspect that many readers may feel confused by the usage of the term 'ictidosaur' for Riograndia. The clade 'ictidosaur' is paraphyletic in the strict consensus tree and several other phylogenetic analyses. I would suggest using 'basal tritheledontids' instead.

3. Ictidosaur. We acknowledge the problematic nature of using the term 'ictidosaur' in the previous iteration of this manuscript due to their recovery as a paraphyletic clade in our strict consensus tree. However, as explained above, our revised manuscript includes amendments to the morphological character matrix and phylogenetic analyses, and we now recover Ictidosauria as a monophyletic clade containing *Irajatherium*, *Riograndia*, *Pachygenelus* and *Diarthrognathus* (i.e. Ictidosauria sensu Martinelli & Rougier 2007 - reference 37 in our manuscript). This clade is also shown in a revised figure 1, giving readers a clear reference for our usage of the term "ictidosaur".

4. Because the dentary-squamosal contact evolved independently in the probainognathian cynodonts and Mammaliaformes, its use as the criterion for determining what is a mammal has a drawback. I would like to see a brief discussion included in the main text.

4. Dentary-squamosal contact. We now address the problematic nature of using the dentary-squamosal contact in defining the unique anatomical features of mammals in the section on phylogenetic results (L313) and in the conclusion (L389).

Minor Comments

L25: "probainognathian cynodont" instead of "non-mammaliaform probainognathian cynodont"

Removed "non-mammaliaform" (L25).

L30: Please reword this sentence. By the Late Triassic, a dentary-squamosal joint had already formed in at least two mammalian groups, including Haramiyavia.

We have changed the wording in this sentence and several others throughout the main text (e.g. L208, L224), to acknowledge the presence of a dentary-squamosal joint in Late Triassic taxa such as *Haramiyavia* and added appropriate citations.

Line 38: “ in the Late Triassic” instead of “ in the Early Jurassic”
Changed to “Late Triassic”

Line 70: I would recommend using 'detached middle ear' instead of 'definitive mammalian middle ear' as the terminology does not accurately reflect homology.

We have altered this sentence to now read “The definitive mammalian middle ear (also called the detached middle ear^{21,22})” (L71). We opted to include both terms rather than replacing DMME with DME because both terms are used multiple times in the literature directly connected to and cited by this work. We felt that including both terms would improve readability for those who may wish to read around the relevant literature after viewing our paper.

Line 117: Delete “(AZ)”

Deleted “(AZ)”.

Line 152: “ μ CT data” instead of “ 3D μ CT data”

Deleted “3D”.

Line 260: “ reveal” instead of “reveals”

Changed to “reveal”.

Line 589: “ventral view of same” instead of “ventral view same”

Added “of”.

Line 600: Delete “(1973)”

Deleted “(1973)”

Extended Data Figure 2: “*Brasilodon*” in italic.

Corrected

References

Barghusen, H. R., & Hopson, J. A. (1970). Dentary-squamosal joint and the origin of mammals. *Science*, 168(3931), 573-575.

Bonaparte, J. F., Martinelli, A. G., & Schultz, C. L. (2005). New information on *Brasilodon* and *Brasilitherium* (Cynodontia, Probainognathia) from the late Triassic of southern Brazil. *Revista brasileira de paleontologia*. Rio de Janeiro, RJ. Vol. 8, n. 1 (abr. 2005), p. 25-46.

Norton, L. A., Abdala, F., & Benoit, J. (2023). Craniodental anatomy in Permian–Jurassic Cynodontia and Mammaliaformes (Synapsida, Therapsida) as a gateway to defining mammalian soft tissue and behavioural traits. *Philosophical Transactions of the Royal Society B*, 378(1880), 20220084.

Luo, Z. X., Gatesy, S. M., Jenkins Jr, F. A., Amaral, W. W., & Shubin, N. H. (2015). Mandibular and dental characteristics of Late Triassic mammaliaform Haramiyavia and their ramifications for basal mammal evolution. *Proceedings of the National Academy of Sciences*, 112(51), E7101-E7109.

Referee #3:

Overall Assessment

The novel cranial mandibular joint (CMJ) formed by dentary condyle and squamosal glenoid is a hall mark character of mammaliaforms. The development of this derived feature is a major evolutionary transformation in the rise of mammaliaforms from nonmammalian cynodonts, which has a long study history, and attracts a wide interest.

The new work by James Rawson, Agustin Martinelli and colleagues is very important in revealing significant new information on the precursor states to the dentary condyle in Brasilodon and Ictidosauria - two important sister clades to Mammaliaformes that are diagnosed by the dentary condyle.

I have much admiration for the Rawson et al. team for the impressive quality of original morphological data visualized from excellent CT scans of these important specimens. Furthermore and just as interesting, the newly revealed intermediate character states (“precursor character states”) have turned out to be homoplastic among the sister clades to mammaliaforms, indicative of iterative evolutionary experimentation in sister clades straddling the transition to the fully formed condyle-glenoid CMJ of mammaliaforms.

The apomorphic jaw hinge of dentary condyle and squamosal glenoid of mammaliaforms is capable functionally of bearing load from the force of jaw muscles that are integrated with complex teeth for mammalian chewing. The consensus among experts is that the dentary condyle of mammaliaforms is formed by a further enlargement and a flaring of the pre-existing dentary lateral ridge already present in cynodonts closest to mammaliaforms. The new study by Rawson et al. is the first to accurately visualize the 3D morphologies of these structures in rare fossils of Brasilodon and Riograndia.

These fossils discovered more than 20 years ago by Brazilian and Argentinian colleagues had been well-prepared in traditional fossil preparation. The features exposed on exterior of these fossils are informative as much as can be seen under microscope, and photographed and illustrated in several widely cited studies and in reviews. However, the traditional fossil preparation has its limitation and several aspects of the true 3D anatomy inside the fossil blocks are still unknown, until now. The new CT scans and visualization by Rawson et al are revelatory, and represent a major progress.

This paper has a wealth of new information about in the structure of craniomandibular joint in these sister clades of mammaliaforms - the Stl files submitted for review are abundant and

easy for referees to explore and verify. The new insight from the authors on homoplastic evolution of precursor characters in closest clades to mammaliaforms are adding to a growing and new understanding of the evolutionary building-up of the fully developed dentary condyle. These are strong merits and I support this paper for publication in *Nature*.

We thank the reviewer for their comments on the importance of this work and the suitability for publication in *Nature*. We address their comments in full below.

However, two aspects of this paper still need significant improvement before the paper can be published: (1) supplementary data organization/presentation, and (2) several interrelated interpretations, for which anatomical vocabulary should be fine-tuned, and interpretation can be better contextualized.

Major Revision: Supplementary Information

First, I wish to point out that the STL files are superb and the videos from CT visualization look great – Rawson et al did a great job in segmentation work. Nonetheless, I would suggest some improvement, to make the paper even better. My issues with Supplementary Information are in organization and presentation, which should be improved and will be easy to revise:

Requested change 1 – The authors need to write an introductory text of Supplementary Information and provide a full list of supplementary files (STL files for models, animation video to represent the segmentation, and phylogenetic files Nexus on matrix and TNT files). Currently there is not a master list. So the authors should organize such a list of the Stl files, the CT video, and they should also provide a caption for each STL file and each video, especially which composite reconstruction from which constituent fossil specimens. This is to make each STL and video self-standing with its own caption. It appears that, when uploading these supplementary files in submission, the authors accidentally scrambled the sequence of upload. After downloading these files in a zip folder generated for referees by Nature's website, it took me hours to piece together the jigsaw puzzle about which file belongs to which fossil taxa. However inadvertently and unintentionally on the part of the authors, this poor organization of files is not good for reviewers. If a reviewer or a reader of *Nature* does not already know what these fossils are like, it would be difficult for the reviewers/readers to retrace the interpretation of these fossils by authors. This could be just a website glitch. But the onus is on the authors to fix it – and to write an intro text and to list all files for the supplementary information. The long character list for phylogenetic analysis should also go into the supplementary information text. The supplementary information must be better organized, before this paper can be published.

Request 1. We have improved the organisation of the Supplementary Information in this revised manuscript as the reviewer requests. We have added a master list of Supplementary files provided at the start of the Supplementary information to give the readers more information about files now available in a public data depository (see below), including a caption for each file alongside the file name (L783). For the reconstructed jaw STL files, the caption includes a short description of which specimens have been used in the reconstructions and how they were assembled. We have also moved the phylogenetic character list into the supplementary text (L930).

Regarding the order and naming of the files, in our initial submission each uploaded file had a filename that was descriptive of its contents, e.g. “*Brasilodon* UFRGS-PV929-T animation.mpg”, as they do in this revised submission. It seems that the file names were changed when they were uploaded to the *Nature* website, thereby making it difficult to identify the contents of the files for the reviewer. We have amended this by uploading the complete set of supplementary files to the University of Bristol public data repository, data.bris, at <https://doi.org/10.5523/bris.2ie8neiry701e23iayx9gdsytv>.

Request 2 – Somewhere in the main text of the paper, the authors must provide a statement as to how and where readers of *Nature* can access the STL files and Video of CT rendering.

I cannot find such a statement or a URL in the main text of the paper, and methods section (in *Nature* standard format). I presume that all STL and video files will be uploaded in a zip folder as Supplementary Information on *Nature*'s website. Or separately the authors can publish these files in Data Dryad (<https://datadryad.org>) with a published DOI. The data submitted by authors for review are impressive. But the authors must also explain how data will be publicly available.

This must be fixed before this paper can be accepted by *Nature*.

Request 2. The reviewer is correct in saying that our intention was to publish this data on a published depository. All supplementary files are now available on the University of Bristol public data repository, data.bris, at <https://doi.org/10.5523/bris.2ie8neiry701e23iayx9gdsytv>. to upload these files as supplementary information to the article in a zipped folder. We have expanded the “Data Availability” section of the methods to ensure we mention all the files that will be included in this depository, including the videos of the rotating surface models. We are happy to expand this section further if the editor and reviewers require any further changes.

Request 3 – RE: *Brasilodon* UFRGS-PV-1043-T – this specimen needs a video, and more illustrations in figures. The flaring posterior end of lateral ridge is preserved in proximity to the squamosal zygoma in this fossil – so it is important for authors to provide better 2D graphics and video of this specimen. Currently this specimen is shown in a small crop-out of local area in Fig 2e (ventral view) and shown again extended data Figure 2 in medial view and posterior view. In revision, I like to ask authors to include a lateral view of this mandible in Extended Data Figure 2. The main composite reconstruction of the mandible (especially its posterior end), as presented in the first STL file of *Brasilodon*, is likely to be based on the specimen of PV1043T. But I have to guess about this because there is not a caption or other explanatory text for each of the STL file. If authors can add a “image caption” to each STL file in a list, this will make it easier for readers to follow. More specifically for reconstruction of the posterior end of the dentary (most relevant to CMJ), such information can facilitate for the interested reader to verify how the authors have accomplished the composite reconstruction from well-preserved parts of individual specimens. This is just a matter of improving the communication.

Request 3. The skull of UFRGS-PV1043-T is separated from the lower jaw and postdentaries, and they were scanned and segmented separately. We have now included an stl file of the lower jaws and postdentaries, as well as a supplementary video showing two full roll and yaw rotations. Another stl file of the complete skull, and a video showing two full roll and yaw rotations, have also been included. A lateral view of the left dentary of this specimen, as preserved, has been added to Extended Data Figure 2. A full caption has also

been provided in the Supplementary Information for each STL file including a description of which specimens were used to make the reconstructed jaw for *Brasilodon* and *Riograndia*. This data provides additional information regarding the anatomy of this specimen to both reviewers and readers.

Request 4 – Video for PV1030T. PV-1030-T is important because it is the only specimen in which the squamosal zygoma is still attached to the petrosal, and the posterior end of the lateral ridge of the dentary in close proximity of squamosal zygoma. Although not perfect, the right squamosal and right dentary of PV1030T are the best available for interpreting the original anatomy. On the left mandible, the postdentary elements and the quadrate are still intact. In revision, I like to request for the authors to provide a new video in full (360 degree) roll-rotation plus a full (360 degree) yaw-rotation of the skull, with segmented basicranial and CMJ parts in color of the skull. This information would be crucial to support authors' interpretation that there is an un-filled gap between the dentary lateral ridge flaring (precursor to condyle), and the zygomatic part of the squamosal – a key anatomical observation of this work. The small cropped local anatomy in Fig. 2e is not enough. A video of the colored elements (from segmentation) would help to better demonstrate this interpretation by the authors.

Request 4. We have now included a supplementary video of PV1030T showing two full roll rotations and two full yaw rotations to better show the anatomy of this key specimen.

Request 5 – Supplementary Video S2 for *Brasilodon* PV628T. Currently this video only shows the PV628T in yaw (around Z-axis) rotation. In revision, I like to see the authors further add a “roll” rotation the specimen around X-axis (along length of rostrum - occipital condyle). Presenting the segmented skull in both yaw and roll rotations (two rotations orthogonal to each other) will allow readers to fully see the anatomy intuitively.

Request 5. We have updated this supplementary video to cycle through two yaw rotations and two roll rotations to better show the anatomy of the jaw in isolation. We have also now included a video that includes both the skull and left jaw as preserved together, showing two full yaw and two full roll rotations. The STL file of this specimen has also been updated to include the skull.

Request 6 – Video S3 for *Riograndia* PV596T. The Yaw (Z-axis) rotation is all good. But the further roll (X-axis along length of cranium) rotation was cut short to 270 degrees. Can the authors extend the roll-rotation more to a full rotation of 360.

Request 6. This is odd, the roll animation in this specimen should finish at the 360-degree mark, thereby doing one full rotation. We have added a second roll rotation in the revised version to ensure all the angles are visible.

Request 7 – Video S4 for *Riograndia* PV833T. The yaw (Z-axis) rotation is cut short. Can you further the yaw-rotation to a full 360 degree. Also, I suggest that the authors add another 360-degree roll-rotation (around the X-axis through the length of the skull), to facilitate viewers to gain an intuitive visual understanding of the morphology.

Request 7. We have added a second yaw rotation to the animation as well as two 360-degree roll rotations to better show off the anatomy in this specimen.

Revision on Anatomical Interpretation

Major Suggestion 1: Not all dentary-squamosal contacts are the same. I suggest that Rawson and colleagues adopt more distinct descriptors of the character-states of simple physical contact of Dentary-Squamosal (pre-mammaliaform cynodonts) and differentiate this from the derived pattern of a synovial joint of dentary condyle and squamosal glenoid that are load-bearing (mammaliaforms). This helps to convey the differences of ictidosaurs-tritheledontids and Brasilodon on the one hand and the Mammaliaformes on the other.

In the descriptive text, I would suggest to call the dentary/squamosal contact for Ictidosaurs (Riograndia+Tritheledontids) (= D-SQ ‘articulation’ via lateral reinforcement). In the meanwhile, call the dentary condyle and squamosal glenoid joint for mammaliaforms. The D/SQ contact and the joint of D-condyle and SQ glenoid are morphological different, although we assume the former is a precursor condition to the latter, in the continuum of evolutionary build-up of the true mammalian dentary condyle with squamosal glenoid.

There is a functional nuance - the former term “dentary-squamosal contact” would be neutral if the contact had (or lacked) load-bearing from chewing muscle force. The latter with a dentary condyle is clearly for load-bearing and a squamosal glenoid for a synovial joint a wide range of joint movement.

The important input of the new work is on the precursor states of dentary/squamosal contact (by lateral reinforcement. This is where the analytic originality of Rawson et al.’s work is.

Starting with Crompton (1958), most early works on the cranio-mandibular joint mentioned “dentary-squamosal articulation” in a loose manner. But we now know much better the morphologies of CMJ. Now it is clear there is a range of configurations between Brasilodontids and ictidosaurs, versus Mammaliaformes. If you need example how these descriptors are verbally differentiated – check out the related text in Luo et al. (2002) discussion and character list.

Request 1. As requested by the reviewer, we have made changes to the main text to better differentiate the two character states present in non-mammaliaform cynodonts and mammaliaforms respectively. We now consistently refer to the condition in *Riograndia* as a ‘contact’, while describing the condyle + glenoid joint of mammaliaforms as a ‘load-bearing joint’ or ‘articulation’, highlighting the differences between the two morphologies. We have also updated the legend of Fig. 4 to read “Dentary-squamosal contact via lateral reinforcement” and “Dentary-squamosal articulation via a condyle and glenoid” for the two conditions.

Major Request for revision #2 – I urge the authors to revise their middle ear reconstruction of Probainognathus in Figs 1 and 4 – Dr. Edgar F. Allin made a landmark contribution with Allin (1975) and this work really changed the opinion on synapsid-therapsid mandibular middle ear. However, one aspect of Allin’s reconstruction is not accurate (Allin 1975; 1986). Dr. Allin over-estimated the size of reflected lamina of cynodonts. All of Allin’s reconstructions of “a long hook-like” reflected lamina of ectotympanic for a range of cynodonts are over-interpretation. The reflected lamina in Allin’s rendering is much too long (too mammal-like) than the short projection actually preserved in the fossils of Probainognathus (Luo’s personal observation of several specimens of this taxon). For example, Hopson-Kitching’s (2001: fig. 5) reconstruction of ectotympanic reflected lamina

Probainognathus (as shown in diagram below) is closer to the original fossil and more accurate.

Hopson & Kitching (2001) fig 5 “ref lam” is the reflected lamina of the ectotympanic – it is short.

Request 2. Our reconstruction of *Probainognathus* was taken from the Allin (1975) reconstruction because it was the most recent reconstruction available showing the full set of postdentary bones in medial view. However, the reviewer rightfully points out the inaccuracy of the reflected lamina of the angular in this model, which we agree is too large based on more recent knowledge of cynodont anatomy. As requested by the reviewer, we have therefore amended our reconstructions in figures 1 and 4 to include a smaller reflected lamina more similar to that shown in Hopson & Kitching (2001).

Major Request for revision 3 – Several specimens of *Pachygenelus* show a thickened flange on the posterior end of the lateral ridge of the dentary (I saw these under microscope).

This structure in specimens of *P. monus* is identical to that of *Diarthrognathus*, and to the lateral ridge flaring of *Riograndia* according visualization of Rawson and co-authors here. In other words, the lateral ridge is thickened and rounded in *Pachygenelus* in the same way (see Crompton and Luo 1993: fig 4.14) as this structure in *Diarthrognathus* (see Crompton and Parkyn 1963).

Regarding Figure 3, I am convinced that the entire ictidosaur clade should be interpreted as possessing the “Dentary-Squamosal via lateral reinforcement” (green diamonds on green lines). *Pachygenelus* should not be in dashed line.

I suspected that Rawson and colleagues relied on Gow’s (1981) abstract (ref #50). But they appear to be unaware that Gow (1981) had turned out to be an erroneous mis-interpretation - I can further explain below.

Gow (1981) is a short abstract – here he challenged Crompton’s (1958, 1963) interpretation of the double joint of the tritheledontid *Diarthrognathus*. Gow’s 1981 abstract made two points. First, Gow pointed out that Crompton mis-identified a part of squamosal as the quadrate for the specimen of *Diarthrognathus*. Gow was correct on this, and I fully endorse it. In response, Crompton (1972: Ref #2) corrected this error.

Second, Gow (1981) questioned Crompton’s interpretation of the D/SQ contact in *Diarthrognathus*, on the basis of Gow’s review of specimens (of *Pachygenelus*) available to him. Gow (1981) briefly stated that “However, once the quadrate is correctly positioned such a glenoid configuration is no longer tenable....” However, here as a reviewer of Rawson et al manuscript, I like to point out to Rawson and colleagues that Gow was wrong on the latter point.

Gow did not publish the actual information (photos or illustrate graphics) to back up his two sentences of Gow 1981 abstract. But I got to see the good photos of “Gow’s *Pachygenelus*” materials through some interesting circumstances.

I met Dr. Chris Gow in 1993 and we had an extensive conversation on this topic of our common interest. Thereafter we exchanged reprints in 1994, and I sent him the reprints of

Crompton & Luo (1993), Luo (1994) and Luo & Crompton (1994), all of which touched on the interpretation of the dentary-squamosal contact in tritheledontids.

In response to these papers, and also likely a follow up with our discussion (1993), in 1997 Gow submitted a manuscript to *Palaeontologia Africana* to describe the full details with excellent stereophotographs of “his *Pachygenelus* material.” In this manuscript, Gow explained that, in his study specimen, the posterior end of the dentary would not have a contact with the squamosal because the quadrate would be positioned on top of dentary lateral ridge and in between dentary’s end and squamosal. This is the key reason why Gow maintained that because there was not a D/SQ contact in his *Pachygenelus* fossil. In the discussion of this manuscript, Gow raised question again (as in his 1981 abstract) about Crompton’s interpretation of a double joint in *Diarthrognathus*. Gow suggested to *Palaeontologia Africana* that the manuscript be given for me to review.

But after I got to see the purported key evidence in good stereophotographs from Gow, I disagreed with Gow on his interpretation. In my review, I pointed out that in Gow’s specimen, the quadrate/articular and the postdentary rod were displaced, post mortem, away from the postdentary trough. This dislocation of “middle ear elements” is self-evident from the stereo photos of this manuscript. Simply put, his initial interpretation in the Gow (1981) abstract was based on a post mortem distortion of the fossil in which the quadrate was displaced by some distance from its original position.

Because Gow’s (1981) abstract had just two sentences on this anatomy but no figure or photo, nobody else had seen “his fossils” (or photographs and illustrated graphics). With Gow’s 1997 manuscript to *Palaeontologia Africana* to describe of this fossil material with good stereo photos for documentation, it was the first time when Gow showed his fossil to the peer-review community. Thus likely I was the first person to assess his evidence, and I had a chance as a referee to point this interpretative error to him. Likely this is also the first time Gow received an alternative opinion about what his fossils actually shows with regards to the jaw hinge.

Although I criticized Gow’s manuscript in my review, I was anticipating that he would revise his manuscript and return it to *Palaeontologia Africana*, or he would rebuttal my review. But Chris Gow never followed up, and later he withdrew this manuscript from *Palaeontologia Africana* (the editors of *Palaeontologia Africana* told me about Gow’s withdrawal of his *Pachygenelus* paper). In late 1990s Gow also retired and moved away from his university.

But unfortunately, because the material under Chris Gow’s study never got into print publication through peer-review, the Gow (1981) abstract (and errors therein) would become the only information from him about his *Pachygenelus* materials. The erroneous interpretation still got cited by the un-suspecting colleagues in this case.

But this is not right.

In Figure 4 caption, Rawson and colleagues are saying “...Dotted line leading to *Pachygenelus* denotes uncertainty as to presence/absence of a distinct condyle”. But this is really not right. Here I offer an illustration by Crompton and Luo (1993: fig 4.14 attached in this review), *Pachygenelus* has a thickened and flaring posterior end of lateral ridge (precursor condition to the dentary condyle). Also see Luo and Crompton 1994 JVP fig. 11B - the

squamosal zygoma of *Pachygenelus* has a concavity that would receive the dentary, on the basis of *Pachygenelus* specimens (SAM-K139).

If anything, Rawson and colleagues would be able to streamline the narrative of ictidosaur-tritheledontids with regards to the dentary-squamosal contact: *Riograndia*-*Pachygenelus*-*Diarthrognathus*-*Tritheledon* are more or less the same - they all had some kind of D/SQ contact, despite the two-sentence statement by Gow (1981).

Request 3. We thank the reviewer for their valuable information and advice concerning the anatomy of *Pachygenelus*, and for their insight on the history of the published (and unpublished) descriptive works on tritheledontids. Little in the way of photographs and images are published of either *Pachygenelus* or *Diarthrognathus*, and our description was based on the information we found in the literature including the work of Gow (1981). Following the reviewer's comments and helpful inclusion of the line drawing from Crompton and Luo (1993), we were able to procure some images of two *Pachygenelus* specimens (one is BP/1/5691, the other lacks an accession number in the photographs) that confirmed the accuracy of the anatomy illustrated in the line drawing (thickened, flaring posterior ridge on the articular process). We agree with the reviewer that the morphology of *Riograndia* and *Pachygenelus*/*Diarthrognathus* can all be accurately described as "dentary-squamosal contact via lateral reinforcement" and have followed the reviewer's suggestion to change the labelling of the entire ictidosaur clade in Figure 4 to green diamonds on green lines and altered a few lines of the text that described the condition in the Jurassic tritheledontids. This work really highlights the need for an updated description of both *Pachygenelus* and *Diarthrognathus* that includes more fossil images and ideally 3D data similar to that presented here for *Brasilodon* and *Riograndia*, so that this vital anatomical information can be added to the published literature.

However, we would still argue that the anatomy of the articular process of *Riograndia* does show a slightly different, presumably more basal, condition than either *Pachygenelus* or *Diarthrognathus*, based on the images and descriptions currently available. The thickened, rounded posterior rim present on *Pachygenelus* and *Diarthrognathus* that forms the posterolateral surface of the articular process is absent, or at least much less developed, in *Riograndia*. This is now shown more clearly in our revised Figure 3 that includes posterior and lateral views of the articular process of MPDC-1B1 in comparison to the line drawing of *Pachygenelus* from Crompton and Luo (1993). The *Riograndia* specimens UFRGS-PV-596-T, UFRGS-PV-1319-T and especially MPDC-1B1 preserve the articular process in its complete form, with the latter two preserving the process within the matrix and therefore only fully visible from our CT scan data. We therefore believe that, between these specimens, we are seeing the true complete anatomy of the articular process of *Riograndia* and can therefore be confident that the thickened posterior ridge with a shallow concavity at its anterior border is absent in this taxon. The posterolateral surface may indeed show some slight flaring as a precursor condition, but it is definitely less developed than the condition in *Diarthrognathus* and *Pachygenelus*. Both the simple, more basal convex process in *Riograndia* and the thickened ridge of *Pachygenelus* and *Diarthrognathus* are reinforcing lateral contacts and are distinctly different from a dentary-squamosal joint involving a dentary condyle and squamosal glenoid (and are represented as such in the revised Figure 4), but we feel that it is useful to point out in the text that the Jurassic taxa seem to show a more derived condition with a more developed lateral posterior contact surface. We have now included STL files and rotating animations of UFRGS-PV-1319-T and MPDC-1B1 to provide additional evidence for our interpretations.

Suggestion for change #4 - Lines 62-63; Lines 366-367: Re: nocturnality. Maybe the authors should delete the mention of nocturnality. Rawson and colleagues mentioned nocturnality in addition to the elevated metabolism and endotherm for mammals. I understand they would need some ad hoc statement of important evolutionary adaptations of mammaliaforms.

In my view, elevated metabolism and endothermy, novel feeding adaptation by chewing (functionally related to CMJ), and miniaturization are certainly important adaptations. However, why nocturnality? With this casual mentioning of nocturnality, Rawson and co-authors would open the manuscript up for potential dispute. Angielczyk and Schmitz (2014) proposed that nocturnality arose convergently in nonmammaliaform synapsids: possibly in early synapsids *Sphenacodon* and *Dimetrodon*, and certainly in several therapsids. Angielczyk and Schmitz (2014) also suggested that *Tritylodon* would be nocturnal, as an example from cynodonts. Thus nocturnality may not be a unique mammaliaform evolutionary innovation.

Delete nocturnality, and stick to (1) endothermy, (2) novel feeding adaptation by chewing, and (3) miniaturization. This helps to streamline the narrative.

Request 4. We take this on board and as suggested have deleted the mention of nocturnality from these sections to streamline the narrative of the paper.

Line by Line Queries and Suggestion of Corrections

Lines 37-38 Re: “...., mammaliaforms attaining a fully independent dentary-squamosal contact with a conspicuous dentary condyle later, in the Early Jurassic.” This loose statement is not accurate because *Haramiyavia* (Late Triassic at 208 ma) is a mammaliaform, and it clearly has a dentary condyle (see Luo et al. 2015 reconstruction of *Haramiyavia*). If the authors prefer to keep this narrative, perhaps it can be modified to: “...., mammaliaforms attaining a fully independent dentary squamosal glenoid joint with a conspicuous dentary condyle, as exemplified by phylogenetically more derived *Haramiyavia* in Late Triassic, and *Sinoconodon* and *Morganucodon* the Early Jurassic.”

We thank the reviewer for pointing this out. We have now updated the text to acknowledge the evolution of the dentary-squamosal joint in the Late Triassic mammaliaform *Haramiyavia* and provided appropriate citations where needed (e.g. L208, L224). We have changed the wording in the abstract to “mammaliaforms attaining a fully independent dentary-squamosal contact with a conspicuous dentary condyle later, in the Late Triassic” (L37) to keep the abstract concise.

Lines 43-44 “The quadrate and the articular, which form the jaw joint of most tetrapods, were greatly reduced and separated...”. This loose statement is a bit imprecise. The quadrate-articular jaw joint is a feature of all gnathostome vertebrates. The phrase “most tetrapods” can be mis-constructed. You either say “gnathostome vertebrates” (which would be politically correct for hardcore cladists) or you say “non-mammalian cynodonts” (contextually relevant for this study). Change to “The quadrate and the articular, which form the jaw joint of gnathostome vertebrate, were greatly reduced and separated...”.

Changed to “form the jaw joint of gnathostome vertebrates”.

Line 48: Re: “there remains a question when, and how often, this jaw joint first evolved”. This sentence can be misconstrued, and should be modified. I would suggest a change to “... there remain a question where along cynodont-mammal phylogeny the precursor conditions evolved in the build-up of this load-bearing jaw joint.” Or something to this effect.

We have changed this part of the sentence to “there remains a question of where, and how often, along the cynodont-mammal phylogeny the precursor conditions to this load-bearing jaw joint first evolved”.

Lines 60: Change from “These results demonstrate that significant experimentation...” to “These results demonstrate that significant evolutionary experimentation...”

Added “evolutionary”.

Lines 67-68: Re: “a unique middle ear containing three ossicles (malleus, incus, stapes) and a dentary-squamosal jaw articulation;” I strongly urge the authors to distinguish the two different character-states: contact of dentary and squamosal vs. joint of dentary condyle and squamosal glenoid. How about modify this sentence to “a unique middle ear containing three ossicles (malleus, incus, stapes) and a load-bearing jaw hinge of dentary condyle and squamosal glenoid;”

Changed to “a unique middle ear containing three ossicles (malleus, incus, stapes) and a load-bearing jaw hinge comprised of a dentary condyle and squamosal glenoid;” We have also modified similar statements throughout the manuscript to further distinguish the different forms of dentary-squamosal contact.

Line 74: change from “as in other tetrapods” to “as in other gnathostome vertebrates”

Changed to “as in other gnathostome vertebrates”.

Lines 121-123. Re: “...Previous descriptions have indicated that Jurassic tritheledontids (or ictidosaur) may possess a type of dentary-squamosal articulation 2-4,6,42, but its similarity to that of mammaliaforms such as Morganucodon is unclear”. This sentence is not an accurate characterization of our current state of knowledge. To be fair and also to be accurate, I urge authors to consider a revised text – for example: “Previous descriptions have indicated that Jurassic tritheledontids (or ictidosaur) may possess a dentary-squamosal contact 2-4,6,42, although this contact in tritheledontids lack the robust dentary condyle as seen in mammaliaforms such as Morganucodon.” OK?

Changed to the above wording.

Line 126. “They” is not an accurate word here. I suggest that this sentence be modified “but the anatomical information of these new fossils have yet to be integrated into the wider analyses of cynodont morphological evolution alongside more well-known species.”

Changed the wording here to “but the anatomical information of these new fossils has yet to be integrated into the wider picture of cynodont morphological evolution alongside more well-known species.”

Lines 133-135. “Hitherto unknown morphology visible in our dataset overturns previous descriptions of jaw joint and middle ear anatomy and demonstrates the extent of homoplastic evolution in the precursors of mammals.” This sentence borders on an overstatement. To be fair and be accurate, I would suggest a revision to “Hitherto unknown morphology newly revealed by our CT scans and incorporated into our phylogenetic dataset demonstrate a much greater extent of homoplastic evolution of the jaw hinge and some middle ear bones in the precursors of mammals.”

Changed this section to “newly revealed by our μ CT scans and incorporated into our phylogenetic dataset demonstrate a much greater extent of homoplastic evolution of the jaw hinge and middle ear bones in the precursors of mammals” (L134).

Lines 372-374. “The relatively larger postdentaries of Riograndia alongside a secondary dentary-squamosal articulation represents a different line of experimentation that occurred earlier than the adoption of a secondary jaw joint in mammaliaforms;...”. I like this discussion. But I would suggest a tweak of this sentence, by changing it to: “The relatively larger postdentaries of Riograndia alongside a secondary dentary-squamosal contact would represent a different evolutionary experimentation that occurred earlier than the acquisition of a secondary jaw joint for load bearing in mammaliaforms;...”.

Changed the wording in this section to “The relatively larger postdentaries of *Riograndia* alongside a secondary dentary-squamosal contact represent a different instance of evolutionary experimentation that occurred earlier than the acquisition of a secondary jaw joint for load bearing in mammaliaforms;” (L374).

Miscellaneous corrections on references:

Ref 2. Crompton (1972) is a single-author paper in an edited volume. You mixed up the name of a single author with the names of the volume editors.

Ref 25. Mussett should not be “MUSSETT”

Ref 30. Oliveira should not be “OLIVEIRA”

Ref 51. Two errors in this cited reference. It should be “Hopson, J. A. and Barghusen, H. R. An analysis of therapsid relationships. in *The Ecology and Biology of Mammal-like Reptiles*, (eds. Hotton, N. MacLean, P. D., Roth, J. J., and E. C. Roth) 83-106 (Smithsonian Institution Press, 1986).

Not “Hopson et al.” and not “(Ed Hotton N.)”

Ref 67. (eds. Hotton, N. MacLean, P. D., Roth, J. J., and E. C. Roth). Not “(ed Hotton, N.)”

All corrections have been made.

Additional changes made

We have cropped out some of the visible white tack used to hold the fossils in place in parts 2a and 3h (formally 3e).

Reviewer Reports on the First Revision:

Referees' comments:

Referee #1 (Remarks to the Author):

Comments on "South American stem mammals reveal homoplastic evolution of the oldest mammalian jaw joint" (2023-12-22267A)

Thanks to the authors for responding to our comments and making necessary changes accordingly, e.g., providing more detail on the articular process of the dentary of Riograndia, which is key evidence to us for this research. In general, the manuscript has been improved.

We have three points for the authors to consider:

1. We still feel the introduction is unnecessarily lengthy – some material, such as the assemblage zone/age of the fossils and the general background knowledge of the mammalian middle ear evolution, could be deleted or placed in the supplementary information. The focused subject is the evolution of the dentary-squamosal joint (contact). In this regard, the authors may consider using the limited space to discuss the purported dentary-squamosal contact in the tritylodontid *Tritylodontoideus maximus* (see Fourie, 1968) in the paper "The jaw articulation of *Tritylodontoideus maximus*", which echoes Simpson's view on *Tritylodontia*). Although Fourie's view is not popular, it is not a fully settled issue, and it is certainly relevant to the focused subject of the study; it, if true, may not be consistent with the interpretation of the authors that the jaw-joint of *Brasilodon* is more similar to that of tritylodontids than to mammaliaforms.
2. In response to one of our comments, the authors correctly pointed out that "The work of Crompton and Parkyn lacked any formal phylogenetic analysis that tested the idea of convergence of the dentary-squamosal contact in a cladistic framework." The authors' work provided new fossil evidence and phylogeny that corroborate, not reject, Crompton and Parkyn's conclusion. We think this is the progress made in this study.
3. Some of the interpretations/speculations reached may not be necessary or may be troublesome for the authors' reasoning in the work. For instance, in interpreting various jaw joint forms of derived probainognathian cynodonts the authors claimed (line 366): "Jaw joint stress in *Brasilodon* was apparently reduced to the point where a reinforcing secondary contact was no longer needed, and articulation could take place solely through the small quadrate-articular joint." There seems no discussion or evidence in the text that supports the view that the jaw joint stress of *Brasilodon* was apparently reduced. In addition, the phrase "where a reinforcing secondary contact was no longer needed" suggests that the jaw joint of *Brasilodon* possibly represents a primitive reversal to a pre-Riograndia condition; this is not impossible but certainly not the preferred interpretation advocated by the authors.

In short, this study provides needed morphological evidence from key basal mammaliaform taxa to support the mosaic evolutionary pattern of the mammalian jaw joint and middle ear.

Referee #2 (Remarks to the Author):

The authors have performed the phylogenetic analyses and addressed the comments I raised in my initial review. I now recommend that the manuscript be accepted for publication. My comments below pertain to the supplementary information:

1. The supplementary information is difficult to follow. I advise the authors to add a table of contents and provide detailed morphological descriptions of the postdentary bones and jaws of *Brasilodon* and *Riograndia*.
2. I suggest the authors convert the phylogenetic analysis trees into extended figures, as they are central to the discussion and figures 1 and 4.

Referee #3 (Remarks to the Author):

by Zhe-Xi Luo (UChicago) May 2024

Merits

I re-emphasize that this work by James Rawson, Agustin Martinelli and colleagues is an important contribution about *Brasilodon* and *Riograndia* - two taxa of major sister clades to the Mammaliaformes. These important fossils (*Brasilodon* and *Riograndia*) were found by the UFRGS team some two decades ago, but it is only now and with advent of CT-based anatomical work with superb visualization by Rawson et al. that we get to learn about the true extent of their mandibles and the craniomandibular joint.

The new information of the craniomandibular joint (CMJ) of these fossils by CT visualization is significant in revealing the precursor character states to the “true” mammaliaform CMJ that is made up of a full condyle of the dentary and a full squamosal glenoid. The mammaliaform CMJ is capable of load-bearing for the significantly great jaw adductor muscle force. This joint represents an apomorphic structure with novel function that made mammals uniquely mammalian. A major new insight here is that the precursor character states to the true mammaliaform CMJ apomorphies occur by iterative evolution in separate clades of near-relatives to mammaliaforms, by homoplastic evolutionary experimentation. I fully endorse this proposition.

The high-quality CT visualization of these well-preserved specimens is all good work of segmentation. This volume of data will provide a good stepping stone for down-stream studies by others to further explore the biomechanical evolution of the craniomandibular joint – an important structure with so many biomechanical and evolutionary ramifications and attracted a wide and unabated attention from vertebrate biologists and palaeontologists for decades. Again, the work by Rawson and Martinelli et al. is significant.

Improvement on Presentation and Data Availability Statement

My previous requests in first round review were two-fold: (1) the authors must archive the segmented datasets (STL and CT movies) in public (third-party) repository; and (2) they ought to provide better rotation movies in addition to the STL models. These requests are meant for referees to have better ways to evaluate the anatomical interpretation of this work. Public archive of the STL and movies will facilitate for other researcher to use of these data upon the publication of the paper.

To my request 1, I am pleased with the concrete solution by authors to place the CT movies and STL's in University of Bristol Data Repository with a publication DOI. This is excellent. To my request 2, the revised movies for each specimen now has full rotations in both roll (long axis) and yaw (vertical axis). It is very helpful to see the morphologies from both perspectives.

Further Requests for Changes and Edits

The revised manuscript still has numerous smaller issues of imprecise statements, convoluted sentences and imprecise or incorrect attribution of references. These problems are mostly small, but obviously undesirable. These must be addressed and can be revised relatively easily.

I would point out that, even after the revision and improvement, the manuscript still has some wrong listing of references, a lost reference by accident. In several places, the authors made imprecise attribution of reference and their comparative context. In a couple of places of Rawson et al. text, the imprecise referral of past literature somewhat conflicted with the authors' own discussion. I hasten to add these are relatively easy to correct and should be further revised.

Wrong listings of References

(1) In previous version of Rawson et al., the first two refs are:

1. Luo & Manley (2020) review

2. Crompton's (1972) re-study of Diarthrognathus, Probainognathus and their comparative morphologies of CMJ. This is a single authored paper.

But the authors mistakenly listed Ref 2 (Crompton 1972) as a paper by 3-authors (Crompton, Joysey and Kemp 1972). This paper was Crompton (1972 = single author) published in the festschrift book co-edited by Ken Joysey and Tom Kemp (two co-editors).

For previous revision, I asked that Rawson et al fix this error on Ref 2. But the authors left erroneous listing of Crompton ("triple-authors") intact. Instead, regrettably, the authors wrongly changed Ref 1 Luo and Manley (2020) a correct and appropriate reference, replaced it with Crompton (1972 – single-author).

This careless error creates a new error: Crompton (1972) was listed *twice*: Ref - Crompton (1972) and Ref 2 Crompton, Joysey & Kemp (1972) are exactly the same paper, only that the latter was formatted in a wrong way (the 2 co-editors Joysey and Kemp were mistaken as additional co-authors but they are not).

Rawson et al. should go back to:

"1 - Luo & Manley (2020)"

"2 – Crompton (1972)" (but Rawson needs to re-format this to be a single author paper)

(2) References 60 and 61 is Luo (1994) got listed twice. You should list it only once.

Imprecise referral or attribution to references

The Rawson et al team seemed to have really struggled to finesse the nuances about who said/did what in the past literature. In several locations, the referral to earlier papers is out of the place, or inappropriate.

Changes in the craniomandibular joint through cynodont-mammal transition are complex. There were also debates: Crompton (1972) reinterpreted Romer (1969) work on Probainognathus CMJ (no dentary-squamosal contact as Romer initially claimed); while Barghusen-Hopson (1970) disagreed with Romer's (1969) emphasis of Probainognathus's squamosal contact as "singular evolutionary event". Later Kemp (1983) argued tritylodontids are not related to diademodontids, an idea supported by Crompton (1974) and Hopson and Barghusen (1986). There had been a series of vis-à-vis argument and counter-argument between Kemp and Hopson how to place other therapsids. I am sympathetic for Rawson and co-workers in dealing with this complexity. But the onus is on the authors that they must get this right.

Out of my intent and well wish to help the authors to fine-tune their presentation, I have again identified a long list of small errors that should be corrected, or places of imprecision that should be improved. Many of these are related to nuance of the reference citation, or choice of vocabulary, or use simpler and shorter

sentences, instead of long, convoluted sentences. Here are line-by-line changes and suggestions for change (line # based on the PDF version of the manuscript – I hope that the authors can relocate these).

Abstract Line 34 – Re: “Our findings indicate that derived non-mammaliaform probainognathian cynodonts show elevated jaw joint innovation compared to other cynodont groups, and independently acquire ‘double’ craniomandibular contacts...” How about use some simpler work, and change to

“We postulate that the jaw joint structures underwent faster evolutionary changes in probainognathian cynodonts, than in other cynodonts phylogenetically more distant from mammaliaforms. Some probainognathian clades independently acquired ‘double’ craniomandibular contacts...”

Summary paragraph Lines 42-43. Re: “Mammals possess a unique secondarily evolved jaw joint between the dentary and squamosal bones Ref. 1, Ref.2”. Ref. 1 is Crompton 1972; again Ref. 2 is repeating Crompton 1972 (still incorrectly formatted at a “3-authors” paper). This is an obvious mistake. In the previous version. Ref 1 was Luo and Manley 2020 review; Ref 2 was “Crompton, Kemp and Joysey” paper, both of these would make sense here. Just to change back to the Refs 1 and 2 as in the first draft and to correct the Crompton (1972) as single-authored paper, then it will be all OK.

Summary Paragraph Lines 46-47. Change from “...during the evolution of the first mammals and their earlier relatives, the non-mammaliaform cynodonts.” To “...during the evolution of the **living** mammals from **Mesozoic fossil** relatives, the non-mammaliaform cynodonts Ref 1 (Luo and Manley 2020) to Ref 5.”

Luo’s Note: please bear in mind that the first crown mammals had fully ossified Meckel’s cartilage that connects the dentary to the malleus and incus. The disconnection of the Meckel’s element from the malleus and incus happened independently in living therians and then in monotremes. The original sentence by authors here could be misconstrued. However, revising this sentence as suggested will help the authors to bypass this conundrum.

Lines 88-89. Re: “Understanding the **?mode?** and timing of evolution of the dentary-squamosal contact in cynodonts is therefore crucial to **understanding** the **formation** of this unique mammalian **body plan**.” How about change to:

“Understanding **the phylogenetic path** and timing of evolution of the dentary-squamosal contact **through the cynodont-mammaliaform transition** is therefore crucial to **interpretation of the origins** of this unique mammalian **structure**.”

Lines 97-98. Change “suggesting **morphological** experimentations among some, but not all, groups.” To “suggesting **evolutionary** experimentation among some, but not all, groups.”

Lines 121. “Jurassic tritheledontids (or ictidosaurs) may possess a type of dentary-squamosal contact (1, 4-6, 46)” Ref 48 (Luo and Crompton 1994) actually said this, Ref 48 should be added to this location.

Line 128. Change from “alongside more well-known species” to “alongside better-known species”

Lines 133-134. Re: “using an unprecedented 3D dataset of cynodont fossils obtained by microCT (μ CT) scanning (Extended Data Table 1).” Now that the authors already deposited the visualized CT scanning results, the URL of the Repository should be listed in this location of the text. This should be further

changed to: “using an unprecedented 3D dataset of cynodont fossils obtained by microCT (μ CT) scanning (Extended Data Table 1 and with data.Bris: <https://doi.org/10.5523/bris.2ie8neiry701e23iayx9gdsytv>)”

Lines 134-136. Re: “Hitherto unknown morphology newly revealed by our μ CT scans and incorporated into our phylogenetic dataset demonstrate a much greater extent of homoplastic evolution of the jaw hinge and middle ear bones in the precursors of mammals.” This is an awkward sentence. Also “morphology... demonstrate...” contains a grammatical error.

How about change it to “The new morphology revealed by our μ CT scanning is now incorporated into the dataset for our new phylogenetic analysis (Fig 4 and Supplementary Information), which demonstrates a much greater extent of homoplastic evolution of the jaw hinge and middle ear characters in phylogenetic predecessors of mammals.”

Lines 137-139. Change from “..., whereas Riograndia possesses a clear dentary-squamosal contact that predates the mammaliaform condyle-glenoid joint by approximately 17 ma...”. To

“..., whereas Riograndia possesses a clear dentary-squamosal contact similar to that of the closely related tritheledontids. And Riograndia predates the earliest-known mammaliaform preserved with a condyle-glenoid joint by approximately 17 ma....”

Lines 139-141. This sentence is unnecessarily complicated, bordering on being self-conflicting. I think it was garbled up by repetitive uses of words “ictidoaurs... in derived Jurassic ictidosaur (i.e, tritheledontids), with unintentional error by placing “jaw joints of mammaliaforms later evolving...” in between.

I propose that the authors should simplify this, to “This demonstrates that the ictidosaur-tritheledontid clade (Fig 4) independently acquired the dentary-squamosal contact, which bears resemblance to the jaw hinge of mammaliaforms.”

Lines 141-144. The words “... among cynodonts are concentrated crownwards,” are losing context and logical continuity, because these got separated too far apart from “derived probaingnathians.” This should be rearranged, and simplified to “Evolutionary experimentation with jaw hinge occurred in derived cynodonts, with some precursor characters evolved iteratively in probainognathians, and then in mammaliaforms.”

Lines 163-164. Re: “found in many other eucynodonts 26”. I propose that both Crompton 1972 (Ref 2) and Luo 2007 (Ref 26) should be cited at this location. Crompton 1972 actually illustrates this (e.g., Crompton 1972: fig. 9 on Probainognathus).

Line 166-167. Re: “Morganucodon 11” – I think Luo and Crompton (1994) quadrate paper (=ref 48) should be also be added at this location. The character of “twisted neck of the quadrate” in Morganucodon and tritheledontids was first recognized as a derived character, by Luo and Crompton (1994).

Lines 167-169. Re: “ μ CT scans allow description of the proximal contact surface that articulates with the squamosal, which is strongly concave is similar manner to Pachygenelus and especially Morganucodon48.” this sentence would be easier for others to follow if it can be revised slightly. I suggest a change to: “ μ CT scans show that the quadrate surface for contacting the squamosal is strongly concave, in a similar manner to Pachygenelus and especially Morganucodon (48)”

Lines 174-175. Re: “generally resemble those of cynodonts like Probainognathus and Pachygenelus in size and shape (3 – Allin 1975; 5 – Luo 2011; 47 Crompton 1995)” I don’t think you should cite Allin (1975) here in this context. Allin (1975) over-reconstructed the reflected lamina as a long rod – this is not correct any more. The specimens of Probainognathus all show the “reflected lamina” to be a much shorter structure. See Crompton 1995 (fig. 4.8 and fig. 4.9a). Citing Luo (2011) (ref 5) and Crompton (1995) (ref 47) is OK here.

Lines 187-188. Re: “Posterior extension of the articular process does not necessarily denote a dentary-squamosal articulation.” This is an important sentence, and I totally endorse it. But I would love for the authors to fine-tune it a bit more. How about change it to

“Posterior extension of the articular process is a prerequisite condition for establishing a dentary-squamosal contact, but this by itself does not denote a dentary-squamosal articulation.

Lines 210-211. “the Early Jurassic (Sinemurian) Sinoconodon 53 (=Luo and Wu 1994 faunal review).” Can you also add Crompton and Luo (1993) (ref 27) to this location.

Lines 217-221. “This contact surface has been described in the Jurassic tritheledontids Pachygenelus and Diarthrognathus (1,46,55) but examination of the 3D morphology, particularly that of UFRGS-PV-596-T, allows us to firmly state that distinct interaction between the dentary and squamosal was also present in the older Riograndia, recovered by several phylogenetic analyses (34,56) as being a more basal ictidosaur or possibly an outgroup to the clade (Fig. 1).” This sentence is convoluted and would be better if can be rearranged. Also what do you mean by “distinct interaction”?

It would be better if this long and complicated statement can be simplified to make the same claim in a more straight-forward way. How about a change to:

“This contact surface is well described in the Jurassic tritheledontids Pachygenelus and Diarthrognathus (1,27, 46, 55), which belong to the more inclusive ictidosaur clade including Riograndia (Fig 1) (refs 34, 35). The 3D morphology based on CT scans, particularly that of UFRGS-PV-596-T, allows us to firmly state that a similar contact between the dentary and squamosal was also present in Riograndia (Fig. 3).”

Lines 227-229. Although I clearly understand what the authors meant to convey here, but I think this comparative anatomical statement could be misconstrued. I suggest a change to “Overall, the squamosal configuration in Riograndia bears resemblance to the surangular-squamosal articulation present in some other eucynodonts such as Probainognathus (Crompton 1972 = ref 1 or ref 2). But the flaring lateral ridge of the dentary in Riograndia excludes the surangular from directly contacting the squamosal.”

Lines 235 -236. Re: “Pachygenelus, another derived tritheledontid from South Africa, likely also possesses this condition (Fig. 3g) 1,5,27,46.” Reference 48 (Luo & Crompton 1994) should be added to this location – ref 48 has illustrated this structure.

Lines 236-241. Re: “It appears then that Riograndia and tritheledontids (ictidosaur *sensu* 37) show variation in the construction of their dentary-squamosal contact, shifting from a simple convex articular process forming the dentary-squamosal contact in the older, likely more basal Riograndia to a condition more closely resembling the dentary condyle in mammaliaforms, though not as well-developed, in the derived Jurassic Diarthrognathus and Pachygenelus.”

This sentence is not clear because it stretches too long and is “running” to string together several comparative statements together, plus an unnecessary qualifying phrase “though not as well-developed” in the middle. Although I can decipher what the authors meant to convey, but it is not easy to follow, and must be changed. I suggest to change it to:

“It appears then that **ictidosaurs (including Riograndia and tritheledontids) (ref 37)** show some structural variation of the dentary-squamosal contact: the phylogenetically basal Riograndia shows a simple convex articular process forming the dentary-squamosal contact, whereas the more derived Pachygenelus and Diarthrognathus show more (albeit still incomplete) resemblance to the dentary condyle and squamosal glenoid in mammaliaforms.”

Lines 249-252. Change from “The postdentaries of Riograndia are described in Soares et al. (2011) 6, and details such as the fusion of the angular, articular and prearticular, the posteriorly projected retroarticular process and the lack of a preserved reflected lamina on the angular are confirmed by our μ CT scans.” To

“**Our μ CT scans confirm the key features of postdentaries of Riograndia as described by Soares et al. (2011) 6:** the fusion of the angular, articular and prearticular, and a posteriorly projected retroarticular process of the articular, and the reflected lamina of the angular is absent (or alternatively not ossified).”

Lines 252-254. Re: “Scans also confirm that the quadrate of Riograndia, though more slender, more closely resembles basal probainognathians like Probainognathus than Brasilodon and Morganucodon 6,48, particularly in the lack of a stapedial process.” This is all word salad! I suggest you change to:

“Riograndia has a slender quadrate, and the new scans confirm it is more similar to that of Probainognathus in the absence of stapedial process (a plesiomorphic character) than to the quadrates of Brasilodon and Morganucodon 6,48, “

Line 270. Change from “such as chisel-shaped lower incisors” to “such as enlarged and procumbent incisors”

Line 286. Re: “Both tritheledontids 27,58-62” – The authors made a mistake of counting Luo (1994) twice. Ref 60 = Luo (1994) and then Ref 61 = Luo (1994) again. Delete the redundancy.

Lines 384-385. Re: “Though only Mammaliaformes **evolved a fully developed** dentary-squamosal joint with a distinct dentary condyle and squamosal glenoid.” This can be shortened to “Though only Mammaliaformes **developed a joint of full dentary condyle** and squamosal glenoid.”

Figures 1 and 4. The tritylodontid (Kayentatherium) restoration was based on reconstruction by Luo and Manley (2020) (the only such figure published for Kayentatherium). Luo and Manley (2020) was accidentally deleted in the revised draft. This reference should be restored in the next revision.

Figure 1 and 4 caption. I propose that the authors add a sentence to this effect: “Details of phylogenetics in supplementary information.”

References

Lines 400-403. The authors accidentally deleted Luo and Manley (2020) review paper. This should be restored. In the meantime, the authors also listed Crompton (1972) paper twice, by accident. Also the

second redundant entry of Crompton 1972 also mistakenly list K A Joysey and T S Kemp as co-authors – wrong! These compounding errors must be corrected together.

Line 516. Ref 51. Jenkins et al. **Osbaeck, F. 1994.** “Osbaeck F” should be deleted. (1994) moved to the end of the reference, following Nature’s format.

Lines 566-567. Luo, Z-X, Kielan-Jaworowska, Z. & **Cifelli, R. L.** (2002) “**Cifell, R. L.**” should be added to the authorship. Pages 1-78. Pagination was not complete.

Lines 578 -579 Ref 77. Barghusen, H. R. (1968) Pages **1-56** (pages are missing)

Lines 587-589 Ref 82. Grossnickle et al. ref. Trends in **E**cology & **E**volution.

Author Rebuttals to First Revision:

We thank the reviewers for their further comments and insights regarding our manuscript. We have accommodated all requested changes and addressed all the reviewer comments. Please find details below.

Referee #1:

Comments on “South American stem mammals reveal homoplastic evolution of the oldest mammalian jaw joint” (2023-12-22267A)

Thanks to the authors for responding to our comments and making necessary changes accordingly, e.g., providing more detail on the articular process of the dentary of *Riograndia*, which is key evidence to us for this research. In general, the manuscript has been improved.

We have three points for the authors to consider:

1. We still feel the introduction is unnecessarily lengthy – some material, such as the assemblage zone/age of the fossils and the general background knowledge of the mammalian middle ear evolution, could be deleted or placed in the supplementary information. The focused subject is the evolution of the dentary-squamosal joint (contact). In this regard, the authors may consider using the limited space to discuss the purported dentary-squamosal contact in the tritylodontid *Tritylodoideus maximus* (see Fourie, 1968) in the paper “The jaw articulation of *Tritylodoideus maximus*”, which echoes Simpson’s view on Tritylodontia). Although Fourie’s view is not popular, it is not a fully settled issue, and it is certainly relevant to the focused subject of the study; it, if true, may not be consistent with the interpretation of the authors that the jaw-joint of *Brasilodon* is more similar to that of tritylodontids than to mammaliaforms.

We thank their reviewer for their suggestion to include more detail on the Fourie paper. While this anatomical interpretation is certainly relevant to our work here, we are sceptical of its conclusion. The specimen of *Tritylodoideus* (considered by Hopson and Kitching 1972 to be a junior synonym of *Tritylodon*) was heavily distorted by taphonomic processes and was preserved as negative space in the sandstone that was then used to produce a positive cast. Given the poor preservation of the specimen, we do not consider this specimen sufficient evidence to support the presence of a dentary-squamosal joint in *Tritylodon*. Though a comprehensive, up to date description of *Tritylodon* is currently lacking, other studies (e.g. Jasinowski & Chinsamy, 2012) that have examined better-preserved specimens also support the absence of a dentary-squamosal joint. Our personal examination of *Tritylodon* specimens at the Natural History Museum, London corroborates this conclusion. We also argue that, were Fourie’s interpretation found to be correct, our overall conclusions drawn in this paper would be unlikely to change. This is because the derived position of *Tritylodon* within tritylodontids would suggest that a dentary-squamosal contact would be an autapomorphy of this genus rather than an ancestral condition of tritylodontids, which would not affect our conclusion that the dentary-squamosal contact evolved independently in ictidosaur and Mammaliaformes. This would, of course, require phylogenetic analysis to verify, but we argue that additional undistorted specimens would be needed before such analysis should take place.

This being said, we agree with the reviewer that this paper should be acknowledged and discussed in our work here. We have added discussion pertaining to this paper to the extended text (L1042) where we discuss the jaw joint morphology of *Oligokyphus*, since the interpretation of tritylodontid jaw joints is most relevant to this section. Given that we have no additional anatomical data on *Tritylodon* to present in this work, we believe that this discussion of the paper and its interpretations is the extent that we can contribute to this discussion at this time.

Regarding the introduction, we argue that the context of mammalian middle ear evolution is vital for understanding the findings and significance of this research. Given that *Nature* is read by a wide variety of audiences, we felt that it was important to introduce these concepts thoroughly in the introduction. Should the article require cutting down before publication, we will follow the reviewer's or editor's recommendation to reduce the length of these sections.

2. In response to one of our comments, the authors correctly pointed out that “The work of Crompton and Parkyn lacked any formal phylogenetic analysis that tested the idea of convergence of the dentary-squamosal contact in a cladistic framework.” The authors' work provided new fossil evidence and phylogeny that corroborate, not reject, Crompton and Parkyn's conclusion. We think this is the progress made in this study.

We thank the reviewer for their comments. We agree that our work provides further corroboration to the work of Crompton and Parkyn, and we hope that further data on ictidosaur jaw joint anatomy will be published in the future, particularly on *Pachygenelus* and *Diarthrognathus*.

3. Some of the interpretations/speculations reached may not be necessary or may be troublesome for the authors' reasoning in the work. For instance, in interpreting various jaw joint forms of derived probainognathian cynodonts the authors claimed (line 366): “Jaw joint stress in *Brasilodon* was apparently reduced to the point where a reinforcing secondary contact was no longer needed, and articulation could take place solely through the small quadrate-articular joint.” There seems no discussion or evidence in the text that supports the view that the jaw joint stress of *Brasilodon* was apparently reduced. In addition, the phrase “where a reinforcing secondary contact was no longer needed” suggests that the jaw joint of *Brasilodon* possibly represents a primitive reversal to a pre-Riograndia condition; this is not impossible but certainly not the preferred interpretation advocated by the authors.

We thank the reviewer for highlighting this sentence and we acknowledge that our choice of wording here could be improved to avoid unsubstantiated claims. We have changed this sentence (L369) to “The jaw joint of *Brasilodon* was apparently capable of functioning without a reinforcing secondary contact, and articulation took place solely through the small quadrate-articular joint.” This avoids the implication of any evolutionary trends or reversals, and merely highlights the anatomical condition as evidenced by our data.

In short, this study provides needed morphological evidence from key basal mammalian taxa to support the mosaic evolutionary pattern of the mammalian jaw joint and middle ear.

Referee #2:

The authors have performed the phylogenetic analyses and addressed the comments I raised in my initial review. I now recommend that the manuscript be accepted for publication. My comments below pertain to the supplementary information:

1. The supplementary information is difficult to follow. I advise the authors to add a table of contents and provide detailed morphological descriptions of the postdentary bones and jaws of *Brasilodon* and *Riograndia*.

We have now added a contents page at the start of the Extended Data section (L791) to make the Supplementary information easier to follow. We have also added more detailed morphological descriptions for the lower jaws of *Brasilodon* and *Riograndia* to the Supplementary Material as extended text (L941).

2. I suggest the authors convert the phylogenetic analysis trees into extended figures, as they are central to the discussion and figures 1 and 4.

We have now included the trees as extended figures with captions (L1438, L1444) and referenced them directly within the main text.

Referee #3:

*** Please See the PDF version of this review that preserves font colors***

2nd Round Review of Nature manuscript 2023-12-22267A

“South American stem mammals reveal homoplastic evolution of the oldest mammalian jaw joint” (R1)

By James R. G. Rawson et al (Revised in April 2024)

by Zhe-Xi Luo (UChicago) May 2024

Merits

I re-emphasize that this work by James Rawson, Agustin Martinelli and colleagues is an important contribution about *Brasilodon* and *Riograndia* - two taxa of major sister clades to the Mammaliaformes. These important fossils (*Brasilodon* and *Riograndia*) were found by the UFRGS team some two decades ago, but it is only now and with advent of CT-based anatomical work with superb visualization by Rawson et al. that we get to learn about the true extent of their mandibles and the craniomandibular joint.

The new information of the craniomandibular joint (CMJ) of these fossils by CT visualization is significant in revealing the precursor character states to the “true” mammaliaform CMJ that is made up of a full condyle of the dentary and a full squamosal glenoid. The mammaliaform CMJ is capable of load-bearing for the significantly great jaw adductor muscle force. This joint represents an apomorphic structure with novel function that made mammals uniquely mammalian. A major new insight here is that the precursor character states to the true mammaliaform CMJ apomorphies occur by iterative evolution in separate clades of near-relatives to mammaliaforms, by homoplastic evolutionary experimentation. I fully endorse this proposition.

The high-quality CT visualization of these well-preserved specimens is all good work of segmentation. This volume of data will provide a good stepping stone for down-stream studies by others to further explore the biomechanical evolution of the craniomandibular joint – an important structure with so many biomechanical and evolutionary ramifications and attracted a

wide and an unabated attention from vertebrate biologists and palaeontologists for decades. Again, the work by Rawson and Martinelli et al. is significant.

Improvement on Presentation and Data Availability Statement

My previous requests in first round review were two-fold: (1) the authors must archive the segmented datasets (STL and CT movies) in public (third-party) repository; and (2) they ought to provide better rotation movies in addition to the STL models. These requests are meant for referees to have better ways to evaluate the anatomical interpretation of this work. Public archive of the STL and movies will facilitate for other researcher to use of these data upon the publication of the paper.

To my request 1, I am pleased with the concrete solution by authors to place the CT movies and STL's in University of Bristol Data Repository with a publication DOI. This is excellent. To my request 2, the revised movies for each specimen now has full rotations in both roll (long axis) and yaw (vertical axis). It is very helpful to see the morphologies from both perspectives.

We thank the reviewer for their kind comments and appreciate their detailed report.

Further Requests for Changes and Edits

The revised manuscript still has numerous smaller issues of imprecise statements, convoluted sentences and imprecise or incorrect attribution of references. These problems are mostly small, but obviously undesirable. These must be addressed and can be revised relatively easily.

I would point out that, even after the revision and improvement, the manuscript still has some wrong listing of references, a lost reference by accident. In several places, the authors made imprecise attribution of reference and their comparative context. In a couple of places of Rawson et al. text, the imprecise referral of past literature somewhat conflicted with the authors' own discussion. I hasten to add these are relatively easy to correct and should be further revised.

Wrong listings of References

(1) In previous version of Rawson et al., the first two refs are:

1. Luo & Manley (2020) review
2. Crompton's (1972) re-study of Diarthrognathus, Probainognathus and their comparative morphologies of CMJ. This is a single authored paper.

But the authors mistakenly listed Ref 2 (Crompton 1972) as a paper by 3-authors (Crompton, Joysey and Kemp 1972). This paper was Crompton (1972 = single author) published in the festschrift book co-edited by Ken Joysey and Tom Kemp (two co-editors).

For previous revision, I asked that Rawson et al fix this error on Ref 2. But the authors left erroneous listing of Crompton ("triple-authors") intact. Instead, regrettably, the authors wrongly changed Ref 1 Luo and Manley (2020) a correct and appropriate reference, replaced it with Crompton (1972 – single-author).

This careless error creates a new error: Crompton (1972) was listed *twice*: Ref - Crompton (1972) and Ref 2 Crompton, Joysey & Kemp (1972) are exactly the same paper, only that the latter was formatted in a wrong way (the 2 co-editors Joysey and Kemp were mistaken as additional co-authors but they are not).

Rawson et al. should go back to:

"1 - Luo & Manley (2020)"

“2 – Crompton (1972)” (but Rawson needs to re-format this to be a single author paper)

We have corrected this error in the revised version of the manuscript.

(2) References 60 and 61 is Luo (1994) got listed twice. You should list it only once.

We have corrected this error in the revised version of the manuscript and renumbered the reference list accordingly.

Imprecise referral or attribution to references

The Rawson et al team seemed to have really struggled to finesse the nuances about who said/did what in the past literature. In several locations, the referral to earlier papers is out of the place, or inappropriate.

Changes in the craniomandibular joint through cynodont-mammal transition are complex. There were also debates: Crompton (1972) reinterpreted Romer (1969) work on *Probainognathus* CMJ (no dentary-squamosal contact as Romer initially claimed); while Barghusen-Hopson (1970) disagreed with Romer’s (1969) emphasis of *Probainognathus*’s squamosal contact as “singular evolutionary event”. Later Kemp (1983) argued tritylodontids are not related to diademodontids, an idea supported by Crompton (1974) and Hopson and Barghusen (1986). There had been a series of vis-à-vis argument and counter-argument between Kemp and Hopson how to place other therapsids. I am sympathetic for Rawson and co-workers in dealing with this complexity. But the onus is on the authors that they must get this right.

Out of my intent and well wish to help the authors to fine-tune their presentation, I have again identified a long list of small errors that should be corrected, or places of imprecision that should be improved. Many of these are related to nuance of the reference citation, or choice of vocabulary, or use simpler and shorter sentences, instead of long, convoluted sentences. Here are line-by-line changes and suggestions for change (line # based on the PDF version of the manuscript – I hope that the authors can relocate these).

We thank the reviewer for their detailed examination of our manuscript, and we appreciate all the knowledgeable and helpful suggestions to improve accuracy and readability.

Abstract Line 34 – Re: “Our findings indicate that derived non-mammaliaform probainognathian cynodonts show elevated jaw joint innovation compared to other cynodont groups, and independently acquire ‘double’ craniomandibular contacts...”How about use some simpler work, and change to

“We postulate that the jaw joint structures underwent faster evolutionary changes in probainognathian cynodonts, than in other cynodonts phylogenetically more distant from mammaliaforms. Some probainognathian clades independently acquired ‘double’ craniomandibular contacts...”

We have made the suggested change.

Summary paragraph Lines 42-43. Re: “Mammals possess a unique secondarily evolved jaw joint between the dentary and squamosal bones Ref. 1, Ref.2”. Ref. 1 is Crompton 1972; again Ref. 2 is repeating Crompton 1972 (still incorrectly formatted at a “3-authors” paper). This is an obvious mistake. In the previous version. Ref 1 was Luo and Manley 2020 review; Ref 2 was “Crompton, Kemp and Joysey” paper, both of these would make sense here. Just to change back to the Refs 1 and 2 as in the first draft and to correct the Crompton (1972) as single-authored paper, then it will be all OK.

Correction made.

Summary Paragraph Lines 46-47. Change from "...during the evolution of the first mammals and their earlier relatives, the non-mammaliaform cynodonts." To "...during the evolution of the living mammals from Mesozoic fossil relatives, the non-mammaliaform cynodonts Ref 1 (Luo and Manley 2020) to Ref 5."

Correction made.

Luo's Note: please bear in mind that the first crown mammals had fully ossified Meckel's cartilage that connects the dentary to the malleus and incus. The disconnection of the Meckel's element from the malleus and incus happened independently in living therians and then in monotremes. The original sentence by authors here could be misconstrued. However, revising this sentence as suggested will help the authors to bypass this conundrum.

We thank the reviewer for helping us to clarify this point.

Lines 88-89. Re: "Understanding the ?mode? and timing of evolution of the dentary-squamosal contact in cynodonts is therefore crucial to understanding the formation of this unique mammalian body plan." How about change to:

"Understanding the phylogenetic path and timing of evolution of the dentary-squamosal contact through the cynodont-mammaliaform transition is therefore crucial to interpretation of the origins of this unique mammalian structure."

Correction Made.

Lines 97-98. Change "suggesting morphological experimentations among some, but not all, groups." To "suggesting evolutionary experimentation among some, but not all, groups."

Correction Made

Lines 121. "Jurassic tritheledontids (or ictosaurs) may possess a type of dentary-squamosal contact (1, 4-6, 46)" Ref 48 (Luo and Crompton 1994) actually said this, Ref 48 should be added to this location.

We have added a reference to Luo and Crompton 1994 here and swapped references 47 and 48 in the bibliography.

Line 128. Change from "alongside more well-known species" to "alongside better-known species"

Correction Made

Lines 133-134. Re: "using an unprecedented 3D dataset of cynodont fossils obtained by microCT (CT) scanning (Extended Data Table 1)." Now that the authors already deposited the visualized CT scanning results, the URL of the Repository should be listed in this location of the text. This should be further changed to: "using an unprecedented 3D dataset of cynodont fossils obtained by microCT (CT) scanning (Extended Data Table 1 and with data.Bris: <https://doi.org/10.5523/bris.2ie8neiry701e23iayx9gdsytv>)"

Correction Made

Lines 134-136. Re: “Hitherto unknown morphology newly revealed by our μ CT scans and incorporated into our phylogenetic dataset demonstrate a much greater extent of homoplastic evolution of the jaw hinge and middle ear bones in the precursors of mammals.” This is an awkward sentence. Also “morphology.... demonstrate...” contains a grammatical error.

How about change it to “The new morphology revealed by our μ CT scanning is now incorporated into the dataset for our new phylogenetic analysis (Fig 4 and Supplementary Information), which demonstrates a much greater extent of homoplastic evolution of the jaw hinge and middle ear characters in phylogenetic predecessors of mammals.”

Correction Made

Lines 137-139. Change from “..., whereas Riograndia possesses a clear dentary-squamosal contact that predates the mammaliaform condyle-glenoid joint by approximately 17 ma...”. To

“..., whereas Riograndia possesses a clear dentary-squamosal contact similar to that of the closely related tritheledontids. And Riograndia predates the earliest-known mammaliaform preserved with a condyle-glenoid joint by approximately 17 ma...”

Correction Made

Lines 139-141. This sentence is unnecessarily complicated, bordering on being self-conflicting. I think it was garbled up by repetitive uses of words “ictidoaurs... in derived Jurassic ictidosaurs (i.e, tritheledontids), with unintentional error by placing “jaw joints of mammaliaforms later evolving...” in between.

I propose that the authors should simplify this, to “This demonstrates that the ictidosaur-tritheledontid clade (Fig 4) independently acquired the dentary-squamosal contact, which bears resemblance to the jaw hinge of mammaliaforms.”

Correction Made

Lines 141-144. The words “... among cynodonts are concentrated crownwards,” are losing context and logical continuity, because these got separated too far apart from “derived probainognathians.” This should be rearranged, and simplified to “Evolutionary experimentation with jaw hinge occurred in derived cynodonts, with some precursor characters evolved iteratively in probainognathians, and then in mammaliaforms.”

Correction Made

Lines 163-164. Re: “found in many other eucynodonts 26”. I propose that both Crompton 1972 (Ref 2) and Luo 2007 (Ref 26) should be cited at this location. Crompton 1972 actually illustrates this (e.g., Crompton 1972: fig. 9 on Probainognathus).

Correction Made

Line 166-167. Re: “Morganucodon 11” – I think Luo and Crompton (1994) quadrate paper (=ref 48) should be also be added at this location. The character of “twisted neck of the quadrate” in Morganucodon and tritheledontids was first recognized as a derived character, by Luo and Crompton (1994).

Correction Made

Lines 167-169. Re: “CT scans allow description of the proximal contact surface that articulates with the squamosal, which is strongly concave in a similar manner to Pachygenelus and especially Morganucodon48.” this sentence would be easier for others to follow if it can be revised slightly. I suggest a change to: “CT scans show that the quadrate surface for contacting the squamosal is strongly concave, in a similar manner to Pachygenelus and especially Morganucodon (48)”

Correction Made

Lines 174-175. Re: “generally resemble those of cynodonts like Probainognathus and Pachygenelus in size and shape (3 – Allin 1975; 5 – Luo 2011; 47 Crompton 1995)” I don’t think you should cite Allin (1975) here in this context. Allin (1975) over-reconstructed the reflected lamina as a long rod – this is not correct any more. The specimens of Probainognathus all show the “reflected lamina” to be a much shorter structure. See Crompton 1995 (fig. 4.8 and fig. 4.9a). Citing Luo (2011) (ref 5) and Crompton (1995) (ref 47) is OK here.

Correction Made

Lines 187-188. Re: “Posterior extension of the articular process does not necessarily denote a dentary-squamosal articulation.” This is an important sentence, and I totally endorse it. But I would love for the authors to fine-tune it a bit more. How about change it to

“Posterior extension of the articular process is a prerequisite condition for establishing a dentary-squamosal contact, but this by itself does not denote a dentary-squamosal articulation.

Correction Made

Lines 210-211. “the Early Jurassic (Sinemurian) Sinoconodon 53 (=Luo and Wu 1994 faunal review).” Can you also add Crompton and Luo (1993) (ref 27) to this location.

Correction Made

Lines 217-221. ”This contact surface has been described in the Jurassic tritheledontids Pachygenelus and Diarthrognathus (1,46,55) but examination of the 3D morphology, particularly that of UFRGS-PV-596-T, allows us to firmly state that distinct interaction between the dentary and squamosal was also present in the older Riograndia, recovered by several phylogenetic analyses (34,56) as being a more basal ictidosaur or possibly an outgroup to the clade (Fig. 1).” This sentence is convoluted and would be better if it can be rearranged. Also what do you mean by “distinct interaction”?

It would be better if this long and complicated statement can be simplified to make the same claim in a more straight-forward way. How about a change to:

“This contact surface is well described in the Jurassic tritheledontids Pachygenelus and Diarthrognathus (1,27, 46, 55), which belong to the more inclusive ictidosaur clade including Riograndia (Fig 1) (refs 34, 35). The 3D morphology based on CT scans, particularly that of UFRGS-PV-596-T, allows us to firmly state that a similar contact between the dentary and squamosal was also present in Riograndia (Fig. 3).”

Correction Made

Lines 227-229. Although I clearly understand what the authors meant to convey here, but I think this comparative anatomical statement could be misconstrued. I suggest a change to “Overall, the

squamosal configuration in Riograndia bears resemblance to the surangular-squamosal articulation present in some other eucynodonts such as Probainognathus (Crompton 1972 = ref 1 or ref 2). But the flaring lateral ridge of the dentary in Riograndia excludes the surangular from directly contacting the squamosal.”

Correction Made

Lines 235 -236. Re: “Pachygenelus, another derived tritheledontid from South Africa, likely also possesses this condition (Fig. 3g) 1,5,27,46.” Reference 48 (Luo & Crompton 1994) should be added to this location – ref 48 has illustrated this structure.

Correction Made

Lines 236-241. Re: “It appears then that Riograndia and tritheledontids (ictidosaurs sensu 37) show variation in the construction of their dentary-squamosal contact, shifting from a simple convex articular process forming the dentary- squamosal contact in the older, likely more basal Riograndia to a condition more closely resembling the dentary condyle in mammaliaforms, though not as well-developed, in the derived Jurassic Diarthrognathus and Pachygenelus.”

This sentence is not clear because it stretches too long and is “running” to string together several comparative statements together, plus an unnecessary qualifying phrase “though not as well-developed” in the middle. Although I can decipher what the authors meant to convey, but it is not easy to follow, and must be changed. I suggest to change it to:

“It appears then that ictidosaurs (including Riograndia and tritheledontids) (ref 37) show some structural variation of the dentary-squamosal contact: the phylogenetically basal Riograndia shows a simple convex articular process forming the dentary-squamosal contact, whereas the more derived Pachygenelus and Diarthrognathus show more (albeit still incomplete) resemblance to the dentary condyle and squamosal glenoid in mammaliaforms.”

Correction Made

Lines 249-252. Change from “The postdentaries of Riograndia are described in Soares et al. (2011) 6, and details such as the fusion of the angular, articular and prearticular, the posteriorly projected retroarticular process and the lack of a preserved reflected lamina on the angular are confirmed by our μ CT scans.” To

“Our μ CT scans confirm the key features of postdentaries of Riograndia as described by Soares et al. (2011) 6: the fusion of the angular, articular and prearticular, and a posteriorly projected retroarticular process of the articular, and the reflected lamina of the angular is absent (or alternatively not ossified).”

Correction Made

Lines 252-254. Re: “Scans also confirm that the quadrate of Riograndia, though more slender, more closely resembles basal probainognathians like Probainognathus than Brasilodon and Morganucodon 6,48, particularly in the lack of a stapedial process.” This is all word salad! I suggest you change to:

“Riograndia has a slender quadrate, and the new scans confirm it is more similar to that of Probainognathus in the absence of stapedial process (a plesiomorphic character) than to the quadrates of Brasilodon and Morganucodon 6,48,

Correction Made

Line 270. Change from “such as chisel-shaped lower incisors” to “such as enlarged and procumbent incisors”

Correction Made

Line 286. Re: “Both tritheledontids 27,58-62” – The authors made a mistake of counting Luo (1994) twice. Ref 60 = Luo (1994) and then Ref 61 = Luo (1994) again. Delete the redundancy.

We have deleted the repeated reference and adjusted the citation numbers accordingly.

Lines 384-385. Re: “Though only Mammaliaformes evolved a fully developed dentary-squamosal joint with a distinct dentary condyle and squamosal glenoid.” This can be shortened to “Though only Mammaliaformes developed a joint of full dentary condyle and squamosal glenoid.”

Amended to “Though only Mammaliaformes evolved a fully developed dentary condyle and squamosal glenoid,”

Figures 1 and 4. The tritylodontid (Kayentatherium) restoration was based on reconstruction by Luo and Manley (2020) (the only such figure published for Kayentatherium). Luo and Manley (2020) was accidentally deleted in the revised draft. This reference should be restored in the next revision.

Correction Made

Figure 1 and 4 caption. I propose that the authors add a sentence to this effect: “Details of phylogenetics in supplementary information.”

Correction Made

References

Lines 400-403. The authors accidentally deleted Luo and Manley (2020) review paper. This should be restored. In the meantime, the authors also listed Crompton (1972) paper twice, by accident. Also the second redundant entry of Crompton 1972 also mistakenly list K A Joysey and T S Kemp as co-authors – wrong! These compounding errors must be corrected together.

Correction Made

Line 516. Ref 51. Jenkins et al. Osbaeck, F. 1994. “Osbaeck F” should be deleted. (1994) moved to the end of the reference, following Nature’s format.

Correction Made

Lines 566-567. Luo, Z-X, Kielan-Jaworowska, Z. & Cifelli, R. L. (2002) “Cifell, R. L.” should be added to the authorship. Pages 1-78. Pagination was not complete.

Correction Made

Lines 578 -579 Ref 77. Barghusen, H. R. (1968) Pages 1-56 (pages are missing)

Correction Made

Lines 587-589 Ref 82. Grossnickle et al. ref. Trends in Ecology & Evolution.

Journal name now correctly capitalised.

Reviewer Reports on the Second Revision:

Referees' comments:

Referee #1 (Remarks to the Author):

I think the authors made a reasonable case arguing for a relatively lengthy introduction, which I think is fine now if this is not a space issue for Nature.

I had the same feeling as reviewer #2: it's difficult to follow the supplementary information (I think an SI guide is now required by Nature – I may have missed it though). I echo the suggestion to add more detailed descriptions of the postdentary bones and jaws of *Brasilodon* and *Riograndia*, particularly because some of the authors' observations are contrary to previous descriptions. Ideally, if possible, some measurements of the bones in question would be useful to back up qualitative statements such as "relatively larger postdentaries of *Riograndia*", etc.

Referee #2 (Remarks to the Author):

This important manuscript describes the jaw joint anatomy of three key non-mammaliaform cynodont taxa, providing tremendous information for future studies. I support the publication of this manuscript.

I have only a few minor changes/corrections to the text.

Line 222: " The morphology of this contact differs in several important respects from a condylar dentary-squamosal jaw joint, as seen in mammaliaforms such as *Sinoconodon* and *Morganucodon* and *Haramiyavia*." should be "The morphology of this contact differs in several important respects from a condylar dentary-squamosal jaw joint, as seen in mammaliaforms such as *Sinoconodon*, *Morganucodon*, and *Haramiyavia*."

In Figure 2, I recommend using different colors to distinguish the angular and articular/prearticular for easier reader comprehension.

For professionalism, I recommend that the authors use italics for taxon names in Extended Data Figures 1 and 2.

Referee #3 (Remarks to the Author):

Third round review of Rawson, Martinelli et al. Nature Manuscript 2023-12-22267B.

The new CT scan visualization and comparative analyses by this research have added significant new fossil information on ictosaurs and brasilodontids, two sister-clades to the Mammaliaformes. From the first-hand anatomical data of the ictosaur-tritheledontid and Brasilodon, the authors demonstrated that the precursor character-states of the fully-formed mammaliaform jaw hinge are homoplastic, and they made a strong argument that through the cynodont-mammal transition, the jaw joint evolved disparate precursor configurations in the different sister-clades, analogous to alternative evolutionary experimentations, before the establishment of the full dentary-squamosal joint in modern mammals and their Mesozoic fossil relatives.

During the last round of revision, the authors fixed miscellaneous (although mostly small) issues of languages and imprecise statements (such attribution of references). I am satisfied with the revision and improvement, and appreciative of the authors' effort to address my requests for revision.

This research represents a real progress in a research issue of early evolution of mammals with long and wide interest. I support for it to be accepted by Nature.

Zhe-Xi Luo (UChicago)

Author Rebuttals to Second Revision:

We thank the reviewers for their further comments and insights regarding our manuscript. We have accommodated requested changes and addressed all the reviewer comments. Please find details below.

Referee #1:

I think the authors made a reasonable case arguing for a relatively lengthy introduction, which I think is fine now if this is not a space issue for Nature.

I had the same feeling as reviewer #2: it's difficult to follow the supplementary information (I think an SI guide is now required by Nature – I may have missed it though).

An SI guide has been included in this version of the submission

I echo the suggestion to add more detailed descriptions of the postdentary bones and jaws of *Brasilodon* and *Riograndia*, particularly because some of the authors' observations are contrary to previous descriptions.

Given that referee 2 did not request further description in this round of reviews, we presume this comment refers to the previous round. In which case, detailed descriptions of the postdentary bones of *Brasilodon* and *Riograndia* are now included as Supplementary Information (section 2).

Ideally, if possible, some measurements of the bones in question would be useful to back up qualitative statements such as “relatively larger postdentaries of *Riograndia*”, etc.

We have added a column to Table S1 containing measurements of the length (as preserved) and dorsoventral height (at the articular, if preserved) for all postdentary bones of *Brasilodon*, *Riograndia* and *Oligokyphus* segmented in this study.

Referee #2:

This important manuscript describes the jaw joint anatomy of three key non-mammaliaform cynodont taxa, providing tremendous information for future studies. I support the publication of this manuscript.

We thank their reviewer for their helpful feedback throughout the review process and the numerous suggestions that have improved the quality of the manuscript.

I have only a few minor changes/corrections to the text.

Line 222: “The morphology of this contact differs in several important respects from a condylar dentary-squamosal jaw joint, as seen in mammaliaforms such as *Sinoconodon* and *Morganucodon* and *Haramiyavia*.” should be “The morphology of this contact differs in several important respects from a condylar dentary-squamosal jaw joint, as seen in mammaliaforms such as *Sinoconodon*, *Morganucodon*, and *Haramiyavia*.”

Correction made.

In Figure 2, I recommend using different colors to distinguish the angular and articular/prearticular for easier reader comprehension.

We understand the reason behind the request to change the colour of the angular in Fig. 2., but we have opted not to do this for two reasons. Firstly, many of the postdentary bones figured across the main text and Supplementary Information are fused together and so we are unable to draw accurate boundaries between the individual elements. Though this is not the case for the angular in figure 2, we felt that consistency in colour across all our figures was important to minimise confusion and assist with cross comparison between the main figures and our Supplementary information. Secondly, these specimens appear in our Supplementary videos of surface models that have already been uploaded to an online repository and have been assigned a DOI. We felt that altering all these videos and contacting the repository for a potential reupload would be excessively time-consuming for the relatively minor gains in distinguishing the angular and articular/prearticular by colour. In this revised submission, we have altered the positions of the bone labels on Figure 2g to more clearly draw attention to the boundary between the angular and articular/prearticular. We hope that this change will accommodate the reviewers request for easier reader comprehension.

For professionalism, I recommend that the authors use italics for taxon names in Extended Data Figures 1 and 2.

We have now italicised all taxonomic names in figures S1 and S2.

Referee #3:

Third round review of Rawson, Martinelli et al. Nature Manuscript 2023-12-22267B.

The new CT scan visualization and comparative analyses by this research have added significant new fossil information on ictidosaurs and brasilodontids, two sister-clades to the Mammaliaformes. From the first-hand anatomical data of the ictidosaur-tritheledontid and Brasilodon, the authors demonstrated that the precursor character-states of the fully-formed mammaliaform jaw hinge are homoplastic, and they made a strong argument that through the cynodont-mammal transition, the jaw joint evolved disparate precursor configurations in the different sister-clades, analogous to alternative evolutionary experimentations, before the establishment of the full dentary-squamosal joint in modern mammals and their Mesozoic fossil relatives.

During the last round of revision, the authors fixed miscellaneous (although mostly small) issues of languages and imprecise statements (such attribution of references). I am satisfied with the revision and improvement, and appreciative of the authors' effort to address my requests for revision.

This research represents a real progress in a research issue of early evolution of mammals with long and wide interest. I support for it to be accepted by Nature.

Zhe-Xi Luo (UChicago)

We thank the reviewer for their kind words and helpful feedback throughout the review process. Our manuscript has been much improved due to their input.